# Meta-Learning with Neural Bandit Scheduler

**Yunzhe Qi** *
University of Illinois at Urbana-Champaign
Champaign, IL
yunzheq2@illinois.edu

**Yikun Ban**\*
University of Illinois at Urbana-Champaign
Champaign, IL
yikunb2@illinois.edu

**Tianxin Wei**
University of Illinois at Urbana-Champaign
Champaign, IL
twei10@illinois.edu

**Jiaru Zou**
University of Illinois at Urbana-Champaign
Champaign, IL
jiaruz2@illinois.edu

**Huaxiu Yao**
University of North Carolina at Chapel Hill
Chapel Hill, NC
huaxiu@cs.unc.edu

**Jingrui He**
University of Illinois at Urbana-Champaign
Champaign, IL
jingrui@illinois.edu

## Abstract

Meta-learning has been proven an effective learning paradigm for training machine learning models with good generalization ability. Apart from the common practice of uniformly sampling the meta-training tasks, existing methods working on task scheduling strategies are mainly based on pre-defined sampling protocols or the assumed task-model correlations, and greedily make scheduling decisions, which can lead to sub-optimal performance bottlenecks of the meta-model. In this paper, we propose a novel task scheduling framework under Contextual Bandits settings, named BASS, which directly optimizes the task scheduling strategy based on the status of the meta-model. By balancing the exploitation and exploration in meta-learning task scheduling, BASS can help tackle the challenge of limited knowledge about the task distribution during the early stage of meta-training, while simultaneously exploring potential benefits for forthcoming meta-training iterations through an adaptive exploration strategy. Theoretical analysis and extensive experiments are presented to show the effectiveness of our proposed framework.

## 1 Introduction

Meta-learning algorithms [19] have been receiving increased attention due to their strong generalization performance across a wide range of tasks [32, 20]. Most existing meta-learning methods often assume a uniform distribution when drawing meta-training tasks, treating each task as equally important [19]. However, this assumption can possibly fail in real-world scenarios. For example, during data collection, candidate training tasks can be subject to noise perturbation, leading to performance bottlenecks in the meta-model if noisy and clean tasks are treated on an equal footing [43, 45]. In addition, some tasks can pose greater challenges for the meta-model to adapt to, necessitating a more flexible allocation of computational resources during the meta-training process. Furthermore, the task distribution may be skewed, with "tail" tasks receiving inadequate attention under uniform sampling. As a result, the task scheduling methods [14, 51, 31, 24, 29] have been proposed for a refined meta-training strategy.

---

*Equal Contribution.

37th Conference on Neural Information Processing Systems (NeurIPS 2023).

Existing scheduling approaches mainly aim to improve meta-training strategies based on various pre-defined criteria and assumptions, including both non-adaptive and adaptive methods. Among them, non-adaptive methods working on the gradient that updates trainable parameters [31, 13] have proven their superiority over meta-training methods with uniform sampling. But they are unable to adjust the task scheduling strategy adaptively based on the status of the meta-model. On the other hand, adaptive methods aim to schedule the tasks based on Curriculum Learning [14, 52, 50] or adaptively sample the tasks based on the task adaptation difficulty (loss) [51]. However, the existing approaches are all greedy algorithms, which means that they tend to make locally optimal decisions based on the current knowledge (i.e., exploitation only). As the learner only has limited knowledge regarding the task and data distribution at the early stage of meta-training, the greedy

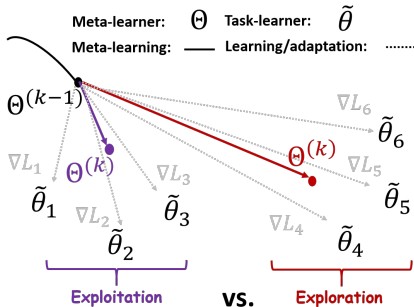

Figure 1: Exploitation and exploration in meta-training iteration $k \in [K]$. E.g., tasks 1, 2, 3 are "frequent" tasks in a skewed task distribution, and tasks 4, 5, 6 are from task distribution "tail". Exploring the "tail" tasks can help improve the meta-model generalization performance (Subsec. 6.2).

strategy can lead to the sub-optimal meta-model due to multiple reasons, such as being misled by the noisy tasks or affected by the skewness of the task distribution. Here, we use Figure 2 to illustrate an example where the greedy approach (only exploitation) may be trapped in sub-optimal solutions.

One intuitive solution for the aforementioned problems is tackling the exploitation-exploration dilemma [4] in meta-training process: balancing the exploitation of current knowledge of selected tasks, and the exploration of under-explored tasks for potential long-term benefits. Therefore, unlike existing approaches with greedy strategies, we propose a novel task scheduling framework under the Contextual Bandits settings [15, 28, 4, 1, 54, 53, 2], named **BA**ndit Ta**S**k **S**cheduler (BASS). In each meta-training iteration, we formulate each candidate meta-training task as an arm under the contextual bandit settings, and the corresponding arm reward will naturally be the meta-model generalization ability score after including this candidate task (arm) into the meta-training process. To achieve this objective, BASS directly learns the mapping from the meta-parameters to the meta-model generalization ability score, instead of depending on the hand-crafted criteria or assumptions. This design also enables us to update meta-parameters and BASS simultaneously within one round of meta-optimization. In particular, instead of greedily scheduling the meta-training tasks solely based on the current knowledge (i.e., exploitation), BASS leverages an additional adaptive exploration module with two different exploration objectives to explore for unrevealed benefits. Our main contributions can be summarized as follows:

- **[Problem and Proposed Framework]** We are the first to formulate the meta-learning task scheduling problem under the Contextual Bandits settings, where we optimize the meta-model w.r.t. chosen task batches in each meta-training iteration. To tackle this problem, we propose a novel bandit-based meta-learning task scheduling framework named BASS, which is model-agnostic and can be applied to various meta-learning frameworks. Different from existing works, BASS directly learns the relationship between the meta-model parameters and the meta-model generalization ability. In addition, instead of greedily exploiting the current knowledge as in existing works, BASS utilizes a novel exploration module to adaptively plan for the future meta-training iterations.

- **[Theoretical Analysis]** Under the general meta-learning settings as well as standard assumptions of over-parameterized neural networks and neural bandits, we derive the theoretical performance guarantee for the proposed BASS framework in terms of regret bound.

- **[Experiments]** To demonstrate the effectiveness of BASS, we compare our method against seven strong baselines on three real data sets, with different specifications. In addition, complementary experiments as well as a case study on Ensemble Inference are also provided to better understand the property and behavior of BASS.

## 2 Related Works

In this section, we briefly review the related works of our proposed BASS framework from the aspects of task scheduling in meta-learning and contextual Bandits.

**Task Scheduling in Meta-Learning.** By considering each meta-training task to be equally important, many existing works sample the training tasks uniformly from the given task distribution [19, 40]. Then, with fixed sampling strategies, [24, 35] propose to assign the task sampling probability based on the quantity of task information, and [31] utilizes a probabilistic sampling method based on class-pairs. Meanwhile, there are also works try to improve task-specific gradients for the randomly sampled tasks [13, 34]. On the other hand, Curriculum-based approaches [14, 52, 50] schedule the tasks based on the difficulty of task adaptation, and [51] propose to adaptively schedule the tasks based on the assumed correlation between the difficulty of meta-training tasks and the meta-model generalization ability. However, the existing works are all greedy approaches that solely focus on the instant benefits, which can lead to sub-optimal meta-models due to the potential performance bottleneck. Compared with existing works, our proposed BASS directly learns the mapping from the meta-parameters to the generalization loss with no pre-defined criteria. With the adaptive exploration strategy, our proposed BASS helps tackle the insufficient knowledge regarding the task distribution by balancing the exploitation and exploration, as well as focusing on the long-term effects.

**Contextual Bandits.** Contextual bandits algorithms aim to solve the sequential decision making problem under the online learning settings, such as online recommendation [28, 49, 6]. Assuming that the mapping from arm contexts to rewards is linear, linear upper confidence bound (UCB) algorithms [15, 28, 4] are proposed to solve the exploitation-exploration dilemma. After kernel-based methods [46, 17] being applied to deal with the non-linear setting where the reward mapping is assumed to be a function in Reproducing Kernel Hilbert Space (RKHS), neural bandit algorithms [54, 53, 7, 5, 39, 8] are proposed to utilize neural networks for the reward and confidence bound estimation. In particular, neural bandit algorithms have demonstrated their superior performance over linear and kernel-based algorithms [54], thanks to the representation power of neural networks. Moreover, instead of using non-negative UCB, EE-Net [9, 10] achieves adaptive exploration by adopting an additional neural network to estimate the potential gain of reward estimations. In this paper, we model the task scheduling problem under the Contextual Bandit settings to balance the exploitation and exploration dilemma regarding the meta-task scheduling.

## 3 Problem Definition and Learning Objective

Under the settings of few-shot supervised meta-learning [19], our goal is to train a meta-model $\mathcal{F}(\cdot; \boldsymbol{\Theta})$ with parameters $\boldsymbol{\Theta} \in \mathbb{R}^p$ that can generalize well to meta-testing tasks, where $p$ is the number of trainable meta-parameters. The meta-model parameters are initialized as $\boldsymbol{\Theta}^{(0)}$. Note that in this work, the trainable parameters are all represented by column vectors, for the simplicity of notation. Following similar settings in [19, 51], in each training iteration $k \in [K]$, we will receive a pool of candidate training tasks $\Omega_{\text{task}}^{(k)} = \{\mathcal{T}_{k,i}\}_{i \in \mathcal{N}}$ where its cardinality $|\Omega_{\text{task}}^{(k)}| = N_{\text{task}}$ and the index set $\mathcal{N} = \{1, \ldots, N_{\text{task}}\}$. Given a task distribution $\mathcal{P}(\mathcal{T})$, we draw each candidate task $\mathcal{T}_{k,i}$ from $\mathcal{P}(\mathcal{T})$ to form the candidate pool, i.e., $\mathcal{T}_{k,i} \sim \mathcal{P}(\mathcal{T})$. Each task $\mathcal{T}_{k,i}$ is also associated with a support data set $D_{k,i}^s$ and a query data set $D_{k,i}^q$, where the samples (including their labels) of $D_{k,i}^s, D_{k,i}^q$ are drawn from the corresponding task data distribution $\mathcal{D}_{\mathcal{T}_{k,i}}$ of task $\mathcal{T}_{k,i}$.

**Meta-training Process.** Then, we need to choose a batch of $B$ tasks $\Omega_k = \{\mathcal{T}_{k,\hat{i}}\}_{\hat{i} \in \widehat{\mathcal{N}}_k} \subset \Omega_{\text{task}}^{(k)}$ for each meta-training iteration $k \in [K]$, where $\widehat{\mathcal{N}}_k \subset \mathcal{N}$ refers to the indices of chosen tasks. With $\boldsymbol{\Theta}^{(k-1)}$ being the meta-model parameters after completing the $(k-1)$-th meta-training iteration, for each **chosen** meta-training task $\mathcal{T}_{k,\hat{i}}, \hat{i} \in \widehat{\mathcal{N}}_k$, we run Gradient Descent (GD) for $J$ steps on its support set $D_{k,\hat{i}}^s$ to obtain the task-specific parameters $\boldsymbol{\Theta}_{k,\hat{i}}^{(J)}$. This refers to the *inner-loop optimization*. In this work, we consider a loss function $\mathcal{L}(\cdot; \cdot)$ that maps the meta-model parameters and the input sample (or sample set) to the loss value, with the range $[0, 1]$ (e.g., the MSE loss on normalized user ratings [27]). Denoting $\boldsymbol{\Theta}_{k,\hat{i}}^{(0)} = \boldsymbol{\Theta}^{(k-1)}$ in meta-training iteration $k$, each GD step is represented as

$$\boldsymbol{\Theta}_{k,\hat{i}}^{(j)} = \boldsymbol{\Theta}_{k,\hat{i}}^{(j-1)} - \eta_1 \cdot \nabla_{\boldsymbol{\Theta}} \mathcal{L}(D_{k,\hat{i}}^s; \boldsymbol{\Theta}_{k,\hat{i}}^{(j-1)}), \quad \forall j \in [J] \tag{1}$$

where $\eta_1 \in \mathbb{R}^+$ is the learning rate of GD. For the simplicity of notation, we formulate the above $J$-iteration *inner-loop optimization* as an operator $\mathcal{I}(\cdot, \cdot) : \mathcal{T} \times \boldsymbol{\Theta} \mapsto \boldsymbol{\Theta}$, such that

$$\boldsymbol{\Theta}_{k,\hat{i}}^{(J)} = \mathcal{I}(\mathcal{T}_{k,\hat{i}}, \boldsymbol{\Theta}^{(k-1)}). \tag{2}$$

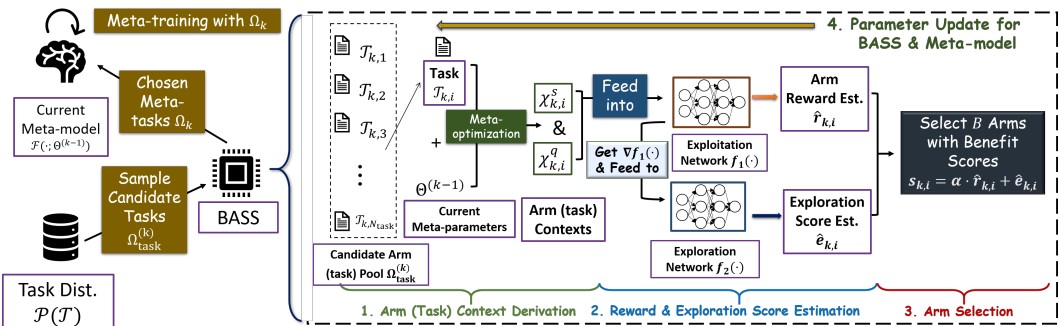

Figure 2: In meta-training iteration $k \in [K]$, the BASS framework overview. We only need one round of the optimization process (LHS of the figure) to update the meta-model and BASS.

Then, we optimize the meta-parameters through the *outer-loop optimization* with query sets

$$\boldsymbol{\Theta}^{(k)} = \boldsymbol{\Theta}^{(k-1)}[\Omega_k] = \boldsymbol{\Theta}^{(k-1)} - \eta_2 \cdot \nabla_{\boldsymbol{\Theta}} \Big( \frac{1}{|\Omega_k|} \sum_{\mathcal{T}_{k,\widehat{i}} \in \Omega_k} \mathcal{L}(D^q_{k,\widehat{i}}; \boldsymbol{\Theta}^{(J)}_{k,\widehat{i}}) \Big) \tag{3}$$

where $\eta_2 \in \mathbb{R}^+$ is the learning rate. Here, the trained meta-model parameters $\boldsymbol{\Theta}^{(K)}$ is expected to minimize the generalization loss $\mathbb{E}_{\mathcal{T} \sim \mathcal{P}(\mathcal{T}), \boldsymbol{x} \sim \mathcal{D}_{\mathcal{T}}} \big[ \mathcal{L}(\boldsymbol{x}; \mathcal{I}(\mathcal{T}, \boldsymbol{\Theta}^{(K)})) \big]$, where we let $\mathcal{D}_{\mathcal{T}}$ being the associated data distribution of task $\mathcal{T}$.

**Evaluating Task Batch Benefits.** Recall that in meta-training iteration $k \in [K]$, the meta-model will be updated based on the chosen batch of $B$ tasks (**Eq. 3**), and we will need to evaluate the benefit in terms of the whole task batch. By the task scheduling problem definition, the learner will *select one task batch* $\Omega_k \subset \Omega_{\text{task}}^{(k)}$ based on its strategy, and the chosen task batch $\Omega_k$ will be used for meta-training in this iteration $k$. Here, we define the reward of the corresponding task batch $\Omega_k$ as

$$h\big(\boldsymbol{\Theta}^{(k-1)}[\Omega_k]\big) = 1 - \mathbb{E}_{\mathcal{T} \sim \mathcal{P}(\mathcal{T}), \boldsymbol{x} \sim \mathcal{D}_{\mathcal{T}}} \big[ \mathcal{L}\big(\boldsymbol{x}; \mathcal{I}(\mathcal{T}, \boldsymbol{\Theta}^{(k-1)}[\Omega_k])\big) \big] \tag{4}$$

where $\boldsymbol{\Theta}^{(k-1)}[\Omega_k]$ refer to the meta-parameters after adapting $\boldsymbol{\Theta}^{(k-1)}$ to task batch $\Omega_k$ based on **Eq. 3**. For simplicity, we let $h : \mathbb{R}^p \mapsto \mathbb{R}$ be the mapping function conditioned on distribution $\mathcal{P}(\mathcal{T})$, which maps the trained meta-model parameters $\boldsymbol{\Theta}^{(k-1)}[\Omega_k]$ to task batch rewards.

**Learning Objective.** Up to meta-training iteration $k$, we have $\Omega^*(K) = \{\Omega_1^*, \ldots, \Omega_K^*\}$ being a series of unknown *optimal* task batches that minimizes the generalization loss. Each optimal batch is $\Omega_k^* = \{\mathcal{T}_{k,i^*}\}_{i^* \in \mathcal{N}_k^*} \subset \Omega_{\text{task}}^{(k)}, k \in [K]$ including a total of $|\Omega_k^*| = B$ tasks, with the indices $\mathcal{N}_k^* \subset \mathcal{N}$. The corresponding optimal meta-parameters are defined as $\boldsymbol{\Theta}^{(K),*} = \arg\inf_{\boldsymbol{\Theta} \in \mathcal{W}} \Big[ \mathbb{E}_{\mathcal{T} \sim \mathcal{P}(\mathcal{T}), \boldsymbol{x} \sim \mathcal{D}_{\mathcal{T}}} \big[ \mathcal{L}\big(\boldsymbol{x}; \mathcal{I}(\mathcal{T}, \boldsymbol{\Theta})\big) \big] \Big]$, where $\mathcal{W} \subset \mathbb{R}^p$ refers to the reachable parameter space, which contains the possible trained meta-parameter values, after observing the available candidate tasks $\{\Omega_{\text{task}}^{(k)}\}, k \in [K]$ and the randomly initialized meta-parameters $\boldsymbol{\Theta}^{(0)}$. Meanwhile, denoting our selection of the meta-training task batches as $\Omega(K) = \{\Omega_1, \ldots, \Omega_K\}$, we will have the corresponding trained meta-parameters $\boldsymbol{\Theta}^{(K)}$. Here, we can define the $K$-iteration regret as

$$R(K) = \mathbb{E}_{\mathcal{T} \sim \mathcal{P}(\mathcal{T}), \boldsymbol{x} \sim \mathcal{D}_{\mathcal{T}}} \Big[ \mathcal{L}\big(\boldsymbol{x}; \mathcal{I}(\mathcal{T}, \boldsymbol{\Theta}^{(K)})\big) - \mathcal{L}\big(\boldsymbol{x}; \mathcal{I}(\mathcal{T}, \boldsymbol{\Theta}^{(K),*})\big) \Big] \tag{5}$$

which measures the difference of the generalization ability between the trained meta-parameters $\boldsymbol{\Theta}^{(K)}$ and the optimal meta-parameters $\boldsymbol{\Theta}^{(K),*}$, after $K$ meta-training iterations.

# 4 Proposed Framework: BASS

In this section, we introduce our proposed BASS framework, which simultaneously optimizes the meta-model and the task scheduling strategy on the fly. The pseudo-code is presented in **Algo. 1**, and the illustration for each meta-training iteration $k \in [K]$ is shown in **Figure 2**.

---

**Algorithm 1 BAndit TaSk Scheduler (BASS)**

---

1: **Input:** Task distribution $\mathcal{P}(\mathcal{T})$. Iterations $K$. GD steps $J$. Number of chosen tasks $B$. Learning rates for meta-model $\eta_1, \eta_2$. Learning rates for BASS $\eta_1^{\boldsymbol{\theta}}, \eta_2^{\boldsymbol{\theta}}$. Exploration coefficient $\alpha \in (0, 1]$.

2: **Output:** Trained meta-parameters $\boldsymbol{\Theta}^{(K)}$.

3: **Initialization:** Randomly initialized meta-model parameters $\boldsymbol{\Theta}^{(0)}$, and BASS parameters $\boldsymbol{\theta}^{(0)}$.

4: **for** each meta-training iteration $k \in [K]$ **do**

5:      Sample a pool of $N_{\text{task}}$ candidate meta-training tasks $\Omega_{\text{task}}^{(k)}$ from the task distribution $\mathcal{P}(\mathcal{T})$.

        ▷ `------ Task Scheduling ------`

6:      **for** For each **candidate** task $\mathcal{T}_{k,i} \in \Omega_{\text{task}}^{(k)}$ **do**

7:         Derive two arm context vectors $\boldsymbol{\chi}_{k,i}^s, \boldsymbol{\chi}_{k,i}^q$ [**Eq. 6**].

8:         Calculate arm benefit score $\widehat{s}_{k,i} = \alpha \cdot f_1(\cdot; \boldsymbol{\theta}_1^{(k-1)}) + f_2(\cdot; \boldsymbol{\theta}_2^{(k-1)})$ with BASS. [**Eq. 7**]

9:      **end for**

10:      From $\Omega_{\text{task}}^{(k)}$, choose the top-$B$ tasks $\Omega_k \subset \Omega_{\text{task}}^{(k)}$ with the highest scores $\{\widehat{s}_{k,i}\}_{i \in \mathcal{N}}$.

        ▷ `------ Parameter Updating ------`

11:      **for** each **chosen** training task $\mathcal{T}_{k,\widehat{i}} \in \Omega_k$ **do**

12:         Derive the corresponding rewards $\widetilde{r}_{k,i}$ and exploration score $\widetilde{e}_{k,i}$. [**Eq. 8-9**]

13:      **end for**

14:      Update meta-parameters to $\boldsymbol{\Theta}^{(k)}$ based on **chosen arms** (tasks) $\mathcal{T}_{k,\widehat{i}} \in \Omega_k$. [**Remark** 2, **Eq.** 3]

15:      Update BASS parameters to $\boldsymbol{\theta}^{(k)}$ with **chosen arms**. [**Eq. 8-9**, **Subsec.** 4.4]

16: **end for**

---

**Remark 1** (Task Scheduling with Contextual Bandits). *By the problem definition, there are $N_{batch} = \binom{N_{task}}{B}$ candidate task batches in each meta-training iteration, which can be a large number. In this case, enumerating all possible task batches and estimating their rewards will be time consuming. Therefore, under the Contextual Bandits settings, BASS alternatively considers **each candidate task** $\mathcal{T}_{k,i} \in \Omega_{task}^{(k)}$ **as an arm**, and directly chooses $B$ arms as the meta-training tasks $\Omega_k \subset \Omega_{task}^{(k)}$. As a result, BASS can (1) reduce the arm space size from $\mathcal{O}(N_{batch})$ to $\mathcal{O}(N_{task})$, while (2) enjoy the performance guarantee (**Section** 5) in terms of regret bound (**Eq.** 5).*

**Section Outline.** Here, we will first present our definition of the **arm contexts** (**Subsec.** 4.1), whose formulation is challenging, because our settings are different from conventional Contextual Bandits where arm contexts are readily available from the environment. Then, applying two neural networks $f_1, f_2$ for exploitation and exploration respectively, we formulate the **arm benefit score** (**Subsec.** 4.2), which measures the benefit if we include the corresponding arm (task) into the current meta-training process. Next, we define the **arm rewards** and **exploration scores** as the labels for training $f_1, f_2$ respectively. In particular, to deal with the challenge of achieving exploration in task scheduling, we incorporate the information and the dynamics w.r.t. meta-optimizations, and formulate two separate exploration objectives for a refined exploration strategy (**Subsec.** 4.3). Finally, we update BASS with GD (**Subsec.** 4.4), and train the meta-model with chosen tasks.

## 4.1 Formulating Arm Contexts

To encode the information from both the task side and the meta-model side, for each candidate task (i.e., arm) $\mathcal{T}_{k,i} \in \Omega_{\text{task}}^{(k)}$ in meta-training iteration $k \in [K]$, we formulate its arm contexts as the meta-parameters after task adaptations, denoted by

$$\boldsymbol{\chi}_{k,i}^s := \boldsymbol{\Theta}_{k,i}^{(J)} = \mathcal{I}(\mathcal{T}_{k,i}, \boldsymbol{\Theta}^{(k-1)}); \quad \boldsymbol{\chi}_{k,i}^q := \boldsymbol{\Theta}^{(k-1)}[\mathcal{T}_{k,i}] = \boldsymbol{\Theta}^{(k-1)} - \eta_2 \nabla_{\boldsymbol{\Theta}} \mathcal{L}(D_{k,i}^q; \boldsymbol{\Theta}_{k,i}^{(J)}) \quad (6)$$

where $\boldsymbol{\Theta}_{k,i}^{(J)}$ are the task-specific parameters after adapting meta-parameters $\boldsymbol{\Theta}^{(k-1)}$ to task $\mathcal{T}_{k,i}$ with inner-loop optimization (**Eqs.** 1-2); while $\boldsymbol{\Theta}^{(k-1)}[\mathcal{T}_{k,i}]$ refer to the meta-parameters after adapting the current $\boldsymbol{\Theta}^{(k-1)}$ to task $\mathcal{T}_{k,i}$, with both inner-loop and outer-loop optimization (as in **Eq.** 3). In particular, we assign each arm $\mathcal{T}_{k,i}$ with two different arm contexts to model the dynamics of meta-parameters w.r.t. inner-loop optimization and outer-loop optimization respectively. For example, the variance of the corresponding data distribution $\mathcal{D}_{\mathcal{T}_{k,i}}$ can be high. In this case, the support set $D_{k,i}^s$ and the query set $D_{k,i}^q$ will be considerably different, which tends to make the corresponding

arm contexts $\chi_{k,i}^s$, $\chi_{k,i}^q$ divergent. As a result, the gradient vectors $\nabla_\theta f_1(\chi_{k,i}^s)$, $\nabla_\theta f_1(\chi_{k,i}^q)$ will likely be distinct from each other. Alternatively, if the support set $D_{k,i}^s$ and the query set $D_{k,i}^q$ are not significantly distinct (the distance between $\chi_{k,i}^s$ and $\chi_{k,i}^q$ is also likely to be relatively small), these two gradient vectors tend to change dramatically when adapting to $\mathcal{T}_{k,i}$. The reason is possibly that the exploitation model $f_1$ is not well adapted to this task $\mathcal{T}_{k,i}$. For both scenarios above, it can be beneficial to include more exploration for the task $\mathcal{T}_{k,i}$, and the target is helping $f_1$ better learn the reward for this task by actively acquiring the knowledge of it. And the two formulated arm contexts can provide important reference for our exploration module.

**Remark 2** (Recycling Arm Contexts). *In order to derive the arm contexts (**Eq. 6**), the gradients for the outer-loop optimization* $\nabla_\Theta \mathcal{L}(D_{k,\widehat{i}}^q; \Theta_{k,\widehat{i}}^{(J)}), \widehat{i} \in \widehat{\mathcal{N}}_k$ *of the chosen arms* $\mathcal{T}_{k,\widehat{i}} \in \Omega_k$ *are calculated. As a result, these gradients can be **recycled** to update the meta-model parameters based on **Eq. 3** (line 15, **Algo. 1**), which helps reduce the computational cost when updating the meta-model.*

## 4.2 Estimating Benefit Scores for Tasks

To determine which arms (tasks) should be included to the meta-training iteration $k$, we formulate the **arm benefit score** estimation for each candidate arm $\mathcal{T}_{k,i} \in \Omega_{\text{task}}^{(k)}$. The estimated benefit score consists of two parts: (1) the estimated *arm reward* of choosing this task based on existing knowledge (i.e., exploitation); (2) and the *exploration score* for the future potential benefit (i.e., exploration). Inspired by recent advances in neural bandits [9], we introduce two separate neural networks, $f_1(\cdot; \theta_1)$ and $f_2(\cdot; \theta_2)$, to estimate the *arm reward* and *exploration score* respectively. The exploitation network $f_1(\cdot; \theta_1)$ aims to learn the mapping $h(\cdot)$ from arm contexts (i.e., meta-parameters) to rewards, while the exploration network $f_2(\cdot; \theta_2)$ aims to learn the uncertainty of reward estimations as the exploration criterion. Different from conventional bandit models, e.g. [9], that works on static arm contexts given by the environment, our design alternatively leverages the evolving information from both the task (arm) side and meta-parameters side, across meta-training iterations. In addition, we consider the dynamics of meta-optimizations for a more comprehensive modeling of the exploration aspect, and the details will be introduced later. Here, given a candidate arm $\mathcal{T}_{k,i} \in \Omega_{\text{task}}^{(k)}$, its estimated benefit score $\widehat{s}_{k,i}$ is formulated as

$$\widehat{s}_{k,i} = \alpha \cdot \widehat{r}_{k,i} + \widehat{e}_{k,i} = \alpha \cdot f_1(\chi_{k,i}^q; \theta_1^{(k-1)}) + f_2\bigg([\nabla_\theta f_1(\chi_{k,i}^s); \ \nabla_\theta f_1(\chi_{k,i}^q)]; \theta_2^{(k-1)}\bigg) \quad (7)$$

where $\alpha \in (0,1]$ is the exploration coefficient to balance exploitation and exploration. Notice that $f_2(\cdot; \theta_2)$ will take the *concatenated gradient* of $f_1(\cdot; \theta_1)$ w.r.t. both arm contexts $\chi_{k,i}^s, \chi_{k,i}^q$ as the input, represented by $[\nabla_\theta f_1(\chi_{k,i}^s); \nabla_\theta f_1(\chi_{k,i}^q)]$. And the output will be the exploration score estimation $\widehat{e}_{k,i}$. To obtain $\nabla_\theta f_1(\chi_{k,i}^q)$, we also calculate $f_1(\chi_{k,i}^q; \theta_1)$ and run the back-propagation. Afterwards, we choose the top-$B$ arms with the highest estimated benefit scores $\widehat{s}_{k,i}, i \in \mathcal{N}$, as the chosen task batch $\Omega_k \subset \Omega_{\text{task}}^{(k)}$ (line 10, **Algo.** 1).

**Design Intuition.** First, recent advances of neural Contextual Bandits [54, 9, 38] have shown that the uncertainty of reward estimations is directly related to the gradients of the estimation model. Therefore, we leverage an exploration module $f_2(\cdot; \theta_2)$ to directly learn this unknown relationship. Second, since $D_{k,i}^s, D_{k,i}^q$ are from the same data distribution $\mathcal{D}_{\mathcal{T}_{k,i}}$, if these two gradients $\nabla_\theta f_1(\chi_{k,i}^s), \nabla_\theta f_1(\chi_{k,i}^q)$ are distinct, the reason can be: (1) the variance of the data distribution $\mathcal{D}_{\mathcal{T}_{k,i}}$ is high (due to the potentially noisy or difficult task); or (2) the gradients of $f_1(\cdot; \theta_1)$ tend to change significantly when adapting to task $\mathcal{T}_{k,i}$. In both cases, it can be harder for $f_1(\cdot; \theta_1)$ to accurately predict the arm reward $r_{k,i}$, and the meta-model can fail to properly adapt to the task $\mathcal{T}_{k,i}$. In this case, we apply the concatenated gradients w.r.t. both arm contexts as the input of $f_2(\cdot; \theta_2)$, in order to provide the information for $f_2(\cdot; \theta_2)$ to evaluate exploration scores.

**Network Architecture and Parameter Initialization.** Here, we consider $f_1(\cdot; \theta_1), f_2(\cdot; \theta_2)$ to be two $L$-layer fully-connected (FC) networks with network width $m$, while $\theta = \{\theta_1, \theta_2\}$ refer to their trainable parameters. For their randomly initialized parameters $\theta^{(0)} = \{\theta_1^{(0)}, \theta_2^{(0)}\}$, the weight matrix entries for the first $L-1$ layers are drawn from the Gaussian distribution $N(0, 2/m)$, while the entries of the last layer ($L$-th layer) are sampled from $N(0, 1/m)$.

**Remark 3** (Reducing Input Complexity). *The input of $f_1(\cdot; \theta_1)$ is the arm context $\chi_{k,i}^q$, whose dimensionality is the number of meta-parameters $p$. A similar situation also exists for the exploration*

network $f_2(\cdot; \boldsymbol{\theta}_2)$. *Inspired by the idea of learning dense low-dimensional representations with Convolutional Neural Networks (CNNs) (e.g., [44]), we apply the **average pooling** approach to approximate original inputs for reducing the running time and space complexity in practice. To show its effectiveness, we will apply this approach on BASS for all the experiments in Section 6.*

## 4.3 Formulating Arm Rewards and Exploration Scores

Different from the conventional neural bandit algorithms [53, 54, 9] where the reward is provided by the environment oracle, we need to carefully design the arm rewards to reflect the arm benefit in terms of the meta-model's generalization ability. Analogous to task batch rewards (**Eq.** 4), we formulate the single **arm reward** $r_{k,i} = h(\boldsymbol{\Theta}^{(k-1)}[\mathcal{T}_{k,i}]) = 1 - \mathbb{E}_{\mathcal{T} \sim \mathcal{P}(\mathcal{T}), \boldsymbol{x} \sim \mathcal{D}_{\mathcal{T}}}\left[\mathcal{L}(\boldsymbol{x}; \mathcal{I}(\mathcal{T}, \boldsymbol{\Theta}^{(k-1)}[\mathcal{T}_{k,i}]))\right]$ for arm $\mathcal{T}_{k,i}$. Since it is impractical to calculate the arm reward by enumerating over $\mathcal{P}(\mathcal{T})$, we sample a batch of validation tasks $\Omega_k^{\text{valid}}$ to derive the unbiased reward approximation, denoted by

$$\widetilde{r}_{k,i} = 1 - \frac{1}{|\Omega_k^{\text{valid}}|} \sum_{\mathcal{T}^{\text{valid}} \in \Omega_k^{\text{valid}}} \mathcal{L}\left(D_{\mathcal{T}^{\text{valid}}}^q; \mathcal{I}(\mathcal{T}^{\text{valid}}, \boldsymbol{\Theta}^{(k-1)}[\mathcal{T}_{k,i}])\right). \tag{8}$$

Here, we adopt the single-step inner-loop optimization [19, 40] to derive $\mathcal{I}(\mathcal{T}_{k,\widehat{i}}, \boldsymbol{\Theta}^{(k-1)}[\mathcal{T}_{k,i}])$ in **Eq.** 8, in order to save the computational cost in practice. Under the few-shot settings, the computation of arm rewards is efficient, since the support set is generally small for inner-loop optimization. The approximation error here can be bounded by the concentration inequality, as the validation tasks $\Omega_k^{\text{valid}}$ are sampled from $\mathcal{P}(\mathcal{T})$.

On the other hand, to formulate the **exploration score** $e_{k,i}$ (i.e., the label for $f_2(\cdot; \boldsymbol{\theta}_2)$), we consider two separate exploration objectives: (1) the prediction uncertainty for the exploitation module $f_1(\cdot; \boldsymbol{\theta}_1)$, which is $r_{k,i} - f_1(\boldsymbol{\chi}_{k,i}^q; \boldsymbol{\theta}_1)$; (2) the validation loss of the meta-model, which represents the difficulty of adapting to $\mathcal{T}_{k,i}$, inspired by the "task difficulty measurer" in Curriculum Learning [52, 50]. As a result, with $\mathcal{L}_{k,i} = \mathcal{L}(D_{k,i}^q; \boldsymbol{\Theta}_{k,i}^{(J)})$ being the validation loss of arm $\mathcal{T}_{k,i}$, we formulate the exploration score as $e_{k,i} = \alpha \cdot \left(r_{k,i} - f_1(\boldsymbol{\chi}_{k,i}^q; \boldsymbol{\theta}_1^{(k-1)})\right) + (1-\alpha) \cdot \mathcal{L}_{k,i}$. Analogously, with the approximated reward $\widetilde{r}_{k,i}$ (**Eq.** 8), we calculate the exploration score approximation by

$$\widetilde{e}_{k,i} = \alpha \cdot \left(\widetilde{r}_{k,i} - f_1(\boldsymbol{\chi}_{k,i}^q; \boldsymbol{\theta}_1^{(k-1)})\right) + (1-\alpha) \cdot \mathcal{L}_{k,i}. \tag{9}$$

Here, the exploration coefficient $\alpha \in (0, 1]$ (in **Eq.** 7) is also used to balance our two exploration objectives, which are (1) prediction uncertainty $r_{k,i} - f_1(\cdot; \boldsymbol{\theta}_1)$: if the exploitation model $f_1(\cdot; \boldsymbol{\theta}_1)$ is under-estimating the arm reward, leading to the positive residual $r_{k,i} - f_1(\cdot; \boldsymbol{\theta}_1)$, we will have a high exploration score to enhance the exploration for this arm; otherwise, when $r_{k,i} - f_1(\cdot; \boldsymbol{\theta}_1)$ is negative, it indicates an excessively high estimation, which will alternatively lower the exploration score to compensate for the over-estimation. With a higher $\alpha$ value, our exploration strategy will focus more on the behavior of the exploitation model $f_1(\cdot)$; (2) the difficulty of task adaptation (i.e., validation loss) $\mathcal{L}_{k,i}$: if the current meta-model does not generalize well to arm $\mathcal{T}_{k,i}$, the validation loss $\mathcal{L}_{k,i}$ will be high, which will also lead to a high exploration score. In this way, our formulation considers two different exploration objectives as well as the dynamics of meta-optimizations (base on concatenated network gradients), for a refined exploration strategy.

## 4.4 Updating Bandit Scheduler Parameters

After updating the meta-parameters (Line 14, **Algo.** 1, **Remark** 2) with tasks $\Omega_k = \{\mathcal{T}_{k,\widehat{i}}\}_{\widehat{i} \in \widehat{\mathcal{N}}_k}$, we proceed to update the parameters of BASS (Line 15, **Algo.** 1). Recall that $f_1(\cdot; \boldsymbol{\theta}_1)$ tries to learn the reward mapping function $h(\cdot)$, and $f_2(\cdot; \boldsymbol{\theta}_2)$ aims to learn the exploration score. Given the selected arms $\Omega_k$, with $\eta_1^{\boldsymbol{\theta}}, \eta_2^{\boldsymbol{\theta}} \in \mathbb{R}^+$ being the learning rates, we apply the GD and quadratic loss to update the parameters of BASS, denoted by $\boldsymbol{\theta}_1^{(k)} = \boldsymbol{\theta}_1^{(k-1)} - \eta_1^{\boldsymbol{\theta}} \cdot \nabla_{\boldsymbol{\theta}_1}\left(\frac{1}{B}\sum_{\mathcal{T}_{k,\widehat{i}} \in \Omega_k} \left|f_1(\boldsymbol{\chi}_{k,\widehat{i}}^q; \boldsymbol{\theta}_1^{(k-1)}) - \widetilde{r}_{k,\widehat{i}}\right|^2\right)$, $\boldsymbol{\theta}_2^{(k)} = \boldsymbol{\theta}_2^{(k-1)} - \eta_2^{\boldsymbol{\theta}} \cdot \nabla_{\boldsymbol{\theta}_2}\left(\frac{1}{B}\sum_{\mathcal{T}_{k,\widehat{i}} \in \Omega_k} \left|f_2([\nabla_{\boldsymbol{\theta}}f_1(\boldsymbol{\chi}_{k,\widehat{i}}^s); \nabla_{\boldsymbol{\theta}}f_1(\boldsymbol{\chi}_{k,\widehat{i}}^q)]; \boldsymbol{\theta}_2^{(k-1)}) - \widetilde{e}_{k,\widehat{i}}\right|^2\right)$. We refer to **Eqs.** 8-9 for calculating the approximated arm reward $\widetilde{r}_{k,\widehat{i}}$ and exploration score $\widetilde{e}_{k,\widehat{i}}$.

## 5 Theoretical Analysis

Recall that in each iteration $k \in [K]$, we receive candidate arms (tasks) $\Omega_{\text{task}}^{(k)} = \{\mathcal{T}_{k,i}\}_{i \in \mathcal{N}}$, and each arm $\mathcal{T}_{k,i}$ is associated with two context vectors $\boldsymbol{\chi}_{k,i}^s, \boldsymbol{\chi}_{k,i}^q$. For the sake of analysis, we normalize these

two contexts such that $\|\boldsymbol{\chi}_{k,i}^s\|_2 = \|\boldsymbol{\chi}_{k,i}^q\|_2 = 1$, and set the exploration coefficient $\alpha = 1$. Following the existing work [47, 48], we let the meta-model $\mathcal{F}(\cdot; \boldsymbol{\Theta})$ be a $L_{\mathcal{F}}$-layer FC network with Gaussian Initialization, with the network width $m_{\mathcal{F}}$. Note that our results can also be generalized to other network architectures, such as CNN and ResNet [22], based on the analysis of over-parameterized neural networks [11, 48, 3]. For the theoretical analysis, we adopt Sigmoid activation for $f_1$ and ReLU for $f_2$, in order to make $f_1$ Lipschitz smooth under over-parameterization settings. Then, we draw trained parameters of BASS with $\{\boldsymbol{\theta}_1^{(k)}, \boldsymbol{\theta}_2^{(k)}\} \sim \{\widetilde{\boldsymbol{\theta}}_1^{(\tau)}, \widetilde{\boldsymbol{\theta}}_2^{(\tau)}\}_{\tau \in [k]}$. Here, starting from the randomly initialized parameters $\{\boldsymbol{\theta}_1^{(0)}, \boldsymbol{\theta}_2^{(0)}\}$, each parameter pair $\{\widetilde{\boldsymbol{\theta}}_1^{(\tau)}, \widetilde{\boldsymbol{\theta}}_2^{(\tau)}\}, \tau \in [k]$ is separately trained on past arm rewards $\{r_{\tau', \widehat{i}}\}_{\tau' \in [\tau], \widehat{i} \in \widehat{\mathcal{N}}_{\tau'}}$ and exploration scores $\{e_{\tau', \widehat{i}}\}_{\tau' \in [\tau], \widehat{i} \in \widehat{\mathcal{N}}_{\tau'}}$ with $J_{\boldsymbol{\theta}}$-iteration GD. Next, similar to existing neural bandit works (e.g., [54, 9, 53]) and the works on meta-model convergence analysis (e.g., [47, 48]), we have the following separateness assumption.

**Assumption 5.1** ($\rho$-Separateness). *After $K$ meta-training iterations, for every pair of arm contexts $\boldsymbol{\chi}_{k,i}^q, \boldsymbol{\chi}_{k',i'}^q$ with $k, k' \in [K]$ such that the corresponding arms $\mathcal{T}_{k,i} \in \Omega_k \wedge \mathcal{T}_{k',i'} \in \Omega_{k'}$, if $(k, i) \neq (k', i')$, we have $\|\boldsymbol{\chi}_{k,i}^q - \boldsymbol{\chi}_{k',i'}^q\|_2 \geq \rho$ where $0 < \rho \leq \mathcal{O}(\frac{1}{L})$.*

The assumption above is mild because of two main reasons: (1) since $L$ is manually chosen (e.g., $L = 2$), we can easily satisfy the condition $0 < \rho \leq \mathcal{O}(\frac{1}{L})$ as long as no two arm contexts are identical; (2) since the meta-parameters $\boldsymbol{\Theta}^{(k)}, k \in [K]$ are constantly changing, the corresponding arm contexts will also be distinct across different meta-training iterations. Additional discussions on this assumption are in Appendix **Section** B. With standard settings of over-parameterized neural networks [47, 3, 11] and the definition of regret $R(K)$ in **Eq.** 5, we have the following **Theorem** 5.2.

**Theorem 5.2.** *Define* $\delta \in (0, 1)$, $0 < \xi_1, \xi_2 \leq \mathcal{O}(1/K)$, $\xi_f = \max\{\xi_1, \xi_2\}$, $0 < \rho \leq \mathcal{O}(1/L)$, $c_\xi > 0$, $\xi_L = (c_\xi)^L$. *Suppose the network width* $m \geq \Omega(Poly(K, L, N_{task}, \rho^{-1}) \log(1/\delta)); m_{\mathcal{F}} \geq \Omega(Poly(K, L_{\mathcal{F}}, N_{task}) \cdot \log(1/\delta))$. *Then, let the learning rates be* $\eta_1, \eta_2 = \Theta\big(\frac{m_{\mathcal{F}}^{-1}}{Poly(K, N_{task}, L_{\mathcal{F}})}\big)$; $\eta_{\boldsymbol{\theta}}^1, \eta_{\boldsymbol{\theta}}^2 = \Theta\big(\frac{\rho \cdot m^{-1}}{Poly(K, N_{task}, L)}\big)$. $J_{\boldsymbol{\theta}} = \Theta\big(\frac{Poly(K, N_{task}, L)}{\rho \cdot \delta^2} \cdot \log(\frac{1}{\xi_1})\big)$. *Following **Algo.** 1, with probability at least* $1 - \delta$, *the $K$-round $R(K)$ of BASS could be bounded by*

$$R(K) \leq \mathcal{O}\left(\frac{1}{\sqrt{K}}\big(\sqrt{2\xi_f} + \frac{3L}{\sqrt{2}} + (1 + 2\gamma_1)\sqrt{2 \log(K/\delta)}\big)\right) + \mathcal{O}\big(\frac{\xi_L^2 KJB\sqrt{L_{\mathcal{F}}}}{\sqrt{m_{\mathcal{F}}}}\big) + \gamma_m \quad (10)$$

*where* $\gamma_m = \mathcal{O}(\frac{1}{m^{1/c_\gamma}}), c_\gamma > 1$, *and* $\gamma_1 = \mathcal{O}(1)$ *with sufficient network width $m$ of BASS.*

The proof is presented in Appendix **Section** D. Here, the first term on the RHS is scaled by the $1/\sqrt{K}$ term, which means the regret bound will shrink along with more iterations $K$. The second term on the RHS is scaled by $1/\sqrt{m_{\mathcal{F}}}$, which makes it a diminutive term under the over-parameterization settings. Since the network depth $L$ of BASS is a small integer (we apply $L = 2$ for experiments in **Section** 6), $\xi_L$ will also be a relatively small constant. Meanwhile, $\gamma_m$ will also decrease significantly with increasingly large network width $m$ of BASS. In contrast, with a convex loss function (e.g., $L_2$ loss or cross-entropy loss) and the same over-parameterization settings, the regret upper bound of the *uniform sampling* strategy [19, 40] can possibly scale up to 1 for the worst-case scenario, and the upper bound will not decrease with more iterations $K$ (Appendix **Lemma** D.13). Alternatively, BASS works under the bandit settings, by directly measuring the meta-model performance difference w.r.t. the chosen task batch and the optimal one. With more iterations $K$, BASS tends to make more accurate scheduling decisions, which makes our regret bound possible. For the existing works, [51] prove that they can improve the optimization landscape with the assumed correlation of task difficulties and meta-model generalization ability. [13] show that their self-paced strategy can improve the model robustness when facing noisy training tasks. Different from previous works, we provide the performance guarantee for the proposed BASS under the neural bandit framework.

## 6 Experiments

In this section, we compare BASS against seven strong baselines, including: (1) Uniform Sampling; non-adaptive self-paced methods and task schedulers (2) SPL [26], (3) Focal-Loss (FOCAL) [30], (4) DAML [29], (5) GCP [31], (6) PAML [24]; and the adaptive task scheduler (7) ATS [51]. Since GCP is not originally compatible with our problem settings and can only work with classification

problems, we properly adapt it by choosing the tasks with the highest probabilities, and apply it on the classification data sets. ANIL [40] is adopted as the backbone meta-learning framework. Due to page limit, we include the complementary experiments (e.g., parameter study for $\alpha$, effects of different levels of task skewness), and the configurations to Appendix **Section** A.

## 6.1 Real Data Sets with Noisy Meta-Training Tasks

We adopt Drug [36], Mini-ImageNet (M-ImageNet) [42] and CIFAR-100 (CIFAR) [25] data sets under the few-shot learning scenario. Similar to [51], we apply classification accuracy as the evaluation criterion for the Mini-ImageNet and CIFAR-100 data sets, and consider the squared Pearson coefficient for the Drug data set. For each meta-learning iteration $k \in [K]$, the learner is given a candidate pool of 10 tasks (i.e., $|\Omega_{\text{task}}^{(k)}| = N_{\text{task}} = 10$), and it will need to choose a batch of $B = 2$ tasks as the training tasks $\Omega_k$ for this iteration. For the Mini-ImageNet and CIFAR-100 data sets, we consider half of the meta-training tasks are perturbed by the label flipping noise [21], where the chance of a label being flipped is $\epsilon \in [0, 1]$. As the Drug data set stands for a regression problem, we draw the label noise from the Gaussian distribution $N(0, \epsilon^2)$. The experimental settings are under 1-shot or 5-shot, 5-Way (for classification data sets) learning scenario with the noise level $\epsilon = 0.5$. The experiment results are shown in **Table** 1 and **Figure** 3. For the average ranking column, we exclude results from the M-ImageNet (1) setting, since BASS and the baselines tend to train a meta-model performing "random guessing". More discussions are in the next paragraph.

Table 1: Results on real data sets [data set (shot); results $\pm$ standard deviation]. *For 1-shot M-ImageNet, all the methods end up with an invalid meta-model performing "random guessing".*

| Algo. \ Data | Drug (1) | Drug (5) | M-ImageNet (1) | M-ImageNet (5) | CIFAR (1) | CIFAR (5) | Avg. Rank |
|---|---|---|---|---|---|---|---|
| Uniform | 0.210±0.013 | 0.220±0.001 | *0.201±0.002* | 0.301±0.025 | 0.234±0.029 | 0.526±0.011 | 4.4 |
| SPL | **0.244±0.008** | 0.236±0.004 | *0.203±0.002* | 0.240±0.018 | 0.200±0.002 | 0.367±0.039 | 5.0 |
| FOCAL | 0.222±0.024 | 0.223±0.003 | *0.200±0.000* | 0.316±0.029 | 0.231±0.024 | 0.485±0.006 | 4.4 |
| DAML | 0.146±0.009 | 0.177±0.003 | *0.201±0.001* | 0.310±0.016 | 0.247±0.003 | 0.414±0.025 | 5.4 |
| GCP | N/A | N/A | *0.201±0.001* | 0.282±0.016 | 0.243±0.007 | 0.508±0.009 | 6.0 |
| PAML | 0.192±0.020 | 0.205±0.009 | *0.199±0.001* | 0.218±0.013 | 0.199±0.004 | 0.316±0.022 | 7.2 |
| ATS | 0.230±0.002 | 0.237±0.014 | *0.201±0.001* | 0.334±0.053 | 0.257±0.048 | 0.515±0.015 | 2.4 |
| **BASS** | 0.242±0.012 | **0.245±0.006** | *0.198±0.004* | **0.351±0.012** | **0.272±0.025** | **0.553±0.008** | **1.2** |

Here, BASS can generally outperform the baselines by directly learning the mapping from the meta-parameters to the arm rewards, as well as balancing the exploitation and exploration. ATS also achieves good performance as it adaptively learns the correlation between the task adaptation difficulty and task scheduling, which proves that it is necessary to apply the adaptive scheduling strategy instead of staying with a fixed protocol. Meanwhile, BASS can generally train the meta-model more efficiently (**Figure** 3), leading to good performances at the early stage of

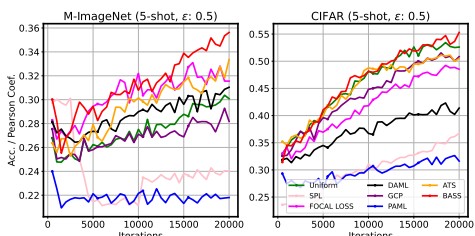

Figure 3: Accuracy results (5-shot, $\epsilon = 0.5$). BASS can achieve a good performance at early meta-training stage.

meta-training. In particular, for the Mini-ImageNet data set under the 1-shot, 5-way settings, all the algorithms fail to train an effective meta-model. Here, under the 5-way classification scenario, all the methods will likely generate a meta-model performing "random guessing" (around 20% accuracy). In this case, utilizing Ensemble Inference techniques [12, 16] can help alleviate this problem, and we include further discussion in the case study (**Subsec.** 6.3).

## 6.2 Effects of the Skewed Task Distribution

The skewed task distribution $\mathcal{P}(\mathcal{T})$ commonly exists in real-word cases. For instance, consider an animal image classification data set where each class (i.e., task) corresponds to one

Table 2: CIFAR-100 with skewed task distribution (5-shot, 5-way / 10-way).

| Data \ Algo. | Uniform | SPL | ATS | **BASS** |
|---|---|---|---|---|
| 5-way | 0.375±0.009 | 0.279±0.002 | 0.382±0.007 | **0.408±0.008** |
| 10-way | 0.264±0.047 | 0.165±0.007 | 0.283±0.039 | **0.320±0.021** |

kind of animals. In this case, felid classes can be considered as "frequent" tasks in the task distribution due to their large quantity and strong mutual correlations, compared with "tail" tasks like kangaroo classes. In this case, paying insufficient attention to the "tail" classes can impair the

generalization performance of the trained meta-model. Thus, to investigate the effects of when the task distribution $\mathcal{P}(\mathcal{T})$ is skewed, we randomly choose some tasks from $\mathcal{P}(\mathcal{T})$, and assign them with higher sampling probabilities (weights). This corresponds to the situation when $\mathcal{P}(\mathcal{T})$ is skewed, so that sampling from $\mathcal{P}(\mathcal{T})$ will likely lead to similar tasks. Here, with the CIFAR-100 data set, we sample 10 tasks and assign them with higher sampling probabilities (weights) (5 tasks with $10\%$, 5 tasks with $5\%$), while the rest of the tasks equally share the remaining $25\%$ probability.

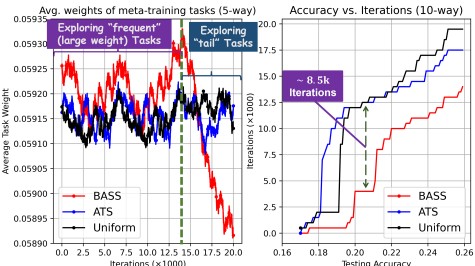

Figure 4: Average weights of chosen meta-training tasks. The testing accuracy vs. iterations needed. BASS can actively exploring for "tail" tasks, and requires much fewer iterations for the same performance (as few as $\sim 1/3$ of baselines' iterations).

From **Table** 2, our proposed BASS maintains the best performance due to its adaptive scheduling strategy and the ability of balancing exploitation and exploration. On the other hand, the baselines are unable to adjust their meta-training strategies towards exploration, when the meta-model has already well adapted to the "frequent" tasks. This can possibly lead to the sub-optimal performance of the meta-model. In particular, based on **Figure** 4, when facing a skewed task distribution, BASS can actively explore the "tail" tasks, after sufficiently exploring the "frequent" tasks. BASS is also able to achieve good performances with fewer meta-training iterations.

## 6.3 Case Study: BASS-aided Ensemble Inference

From **Figure** 3, we notice that BASS can train a meta-model that achieves good generalization performance at the early stage of meta-training. One application of this property is using BASS to assist meta-learning models under the Ensemble Inference settings, where separate models are combined to enhance the generalization ability. One renowned ensemble approach is the model-parameter ensemble [41, 33]. With a collection of $N_E$ individual models $\{\mathcal{F}(\cdot; \boldsymbol{\Theta}_i)\}_{i \in [N_E]}$ of the same architecture, the ensemble model will be $\mathcal{F}_E(\cdot; \boldsymbol{\Theta}_E)$, and its parameters $\boldsymbol{\Theta}_E = \frac{1}{N_E} \sum_{i \in [N_E]} \boldsymbol{\Theta}_i$ are the averaged parameters across individual models. Then, the ensemble model $\mathcal{F}_E(\cdot; \boldsymbol{\Theta}_E)$ will be applied as the inference model for downstream problems. Here, one natural way of obtaining the individual models $\mathcal{F}(\cdot; \boldsymbol{\Theta}_i), i \in [N_E]$ is deeming the models from different training iterations as the individual models for ensemble [12, 16]. Here, we conduct experiments using the ensemble techniques with individual models $\mathcal{F}(\cdot; \boldsymbol{\Theta}_i)$ trained by baselines and BASS. We choose the top $N_E = 10$ models with the smallest validation loss across different meta-training iterations as the individual models $\{\mathcal{F}(\cdot; \boldsymbol{\Theta}_i)\}_{i \in [N_E]}$ for ensemble. The results are shown in **Table** 3. We label the ensemble version of BASS as "BASS-E", and the non-ensemble version as "BASS-S".

Table 3: Ensemble case study [dataset (shot); results $\pm$ standard deviation].

| Algo. \ Data | M-ImageNet (1) | M-ImageNet (5) | CIFAR (1) | CIFAR (5) |
|---|---|---|---|---|
| Uniform | 0.231±0.014 | 0.313±0.027 | 0.270±0.014 | 0.534±0.012 |
| SPL | 0.218±0.006 | 0.298±0.004 | 0.219±0.005 | 0.363±0.038 |
| FOCAL | 0.204±0.005 | 0.347±0.030 | 0.235±0.015 | 0.499±0.003 |
| DAML | 0.222±0.011 | 0.326±0.031 | 0.261±0.008 | 0.432±0.019 |
| GCP | 0.226±0.006 | 0.297±0.011 | 0.268±0.019 | 0.512±0.015 |
| PAML | 0.213±0.024 | 0.232±0.009 | 0.223±0.009 | 0.336±0.029 |
| ATS | 0.202±0.002 | 0.334±0.052 | 0.313±0.081 | 0.517±0.017 |
| **BASS-S** | 0.198±0.004 | 0.351±0.012 | 0.272±0.025 | **0.553±0.008** |
| **BASS-E** | **0.242±0.004** | **0.366±0.003** | **0.327±0.010** | 0.551±0.004 |

Compared with the non-ensemble settings (**Table** 1), we see that the BASS-aided ensemble model can generally perform better. In particular, the ensemble model can improve the meta-model inference performance in significantly difficult cases, such as the Mini-ImageNet under the 1-shot setting (**Subsec.** 6.1). As a result, BASS can help generate high-quality ensemble model with the meta-models trained in different iterations. While the ensemble inference technique can also benefit the other baselines, BASS still maintains decent performances.

## 7 Conclusion

In this paper, we formulate the task scheduling problem in meta-learning under the Contextual Bandits settings, and propose a novel bandit-based task scheduling framework named BASS. It directly optimizes the task sampling strategy based on the status of the meta-model rather than applying fixed task scheduling protocols. Instead of greedily making decisions, BASS can help deal with the insufficient knowledge problem at the early stage of meta-training, as well as plan for the future meta-training iterations with the adaptive exploration strategy. We include both theoretical analyses and a comprehensive set of experiments to demonstrate the effectiveness of our proposed framework as well as its key properties.

## Acknowledgments and Disclosure of Funding

This work is supported by National Science Foundation under Award No. IIS-1947203, IIS-2117902, IIS-2137468, IIS-2002540, and Agriculture and Food Research Initiative (AFRI) grant no. 2020-67021-32799/project accession no.1024178 from the USDA National Institute of Food and Agriculture. The views and conclusions are those of the authors and should not be interpreted as representing the official policies of the funding agencies or the government.

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

# A  Appendix: Experiments (Continue)

## A.1  Further Details for the Experiment Settings

For the data partitioning, we have the Mini-ImageNet and CIFAR-100 data sets divided into the partitions $64 : 16 : 20$, which correspond to the training set, validation set and the testing set respectively. Each class is corresponding to a task. Then, for the Drug data set, we partition the tasks into $4100 : 76 : 100$ representing the training set, validation set and the testing set.

For our BASS, we apply two 2-layer FC networks for $f_1(\cdot; \boldsymbol{\theta}_1)$, $f_2(\cdot; \boldsymbol{\theta}_2)$ respectively, and set network width $m = 200$. For deriving approximated arm rewards, we let $|\Omega_k^{\text{valid}}| = 5$. Recall that we apply the approximation approach mentioned in **Remark** 3 to reduce the space complexity and time complexity in practice for the experiments. Here, we tune the pooling step such that the inputs of $f_1(\cdot; \boldsymbol{\theta}_1)$, $f_1(\cdot; \boldsymbol{\theta}_1)$ are approximately 50 and 20 respectively. For the learning rate, we find the learning rate for BASS with grid search from $\{0.01, 0.001, 0.0001\}$, and choose the learning rates for the meta-model $\eta_1 = 0.01, \eta_2 = 0.001$. The meta-model architecture as well as its learning rates will stay the same for all the baselines and our proposed BASS. For the CIFAR-100 and Mini-ImageNet data sets, we use the the meta-model with four convolutional blocks where the network width of each block is 32, followed by an FC layer as the output layer. For the Drug data set, we apply a meta-model with two FC layers, where the network width is 500. All the experiments are performed on a Linux machine with Intel Xeon CPU, 128GB RAM, and Tesla V100 GPU. Code will be made available at `https://github.com/yunzhe0306/Bandit_Task_Scheduler`.

## A.2  Effect of the Task Noise Magnitude

We conduct the experiments to show the effects of the noise magnitude factor $\epsilon$ on the Drug and CIFAR-100 data sets. The experiment results are shown in **Table** 4.

Table 4: Comparison with baselines with different noise magnitude [data set (noise magnitude $\epsilon$) ; final results $\pm$ standard deviation].

| Algo. \ Data | Drug (0.3) | Drug (0.5) | CIFAR100 (0.3) | CIFAR100 (0.5) |
|---|---|---|---|---|
| Uniform | 0.218±0.007 | 0.220±0.001 | 0.655±0.009 | 0.526±0.011 |
| SPL | 0.243±0.008 | 0.236±0.004 | 0.625±0.017 | 0.367±0.039 |
| FOCAL | 0.224±0.019 | 0.223±0.003 | 0.638±0.010 | 0.485±0.006 |
| DAML | 0.182±0.025 | 0.177±0.003 | 0.543±0.017 | 0.414±0.025 |
| GCP | N/A | N/A | 0.653±0.005 | 0.508±0.009 |
| PAML | 0.186±0.006 | 0.205±0.009 | 0.537±0.009 | 0.316±0.022 |
| ATS | 0.239±0.011 | 0.237±0.014 | 0.651±0.001 | 0.505±0.015 |
| **BASS (Ours)** | **0.258±0.003** | **0.245±0.006** | **0.657±0.005** | **0.553±0.008** |

With increasing noise magnitude $\epsilon$, the performances of the meta-model trained by baselines and our BASS tend to drop, which is intuitive. In particular, for the CIFAR-100 data set, when we increase $\epsilon$, the performance difference between BASS and the other baselines tends to increase. This can be the reason that the greedy baselines with no exploration strategies can be more susceptible to the task noise perturbation, which can lead to the sub-optimal performances of the meta-model.

Table 5: Experiment results of noise-free settings on three real data sets (5-way, 5-shot).

| Data \ Algo. | Uniform | SPL | FOCAL-LOSS | DAML | GCP | PAML | ATS | BASS |
|---|---|---|---|---|---|---|---|---|
| Drug | 0.206±0.012 | 0.234±0.006 | 0.240±0.003 | 0.190±0.002 | N/A | 0.220±0.010 | 0.233±0.001 | **0.256±0.003** |
| M-ImageNet | 0.576±0.016 | 0.554±0.004 | 0.582±0.005 | 0.437±0.015 | 0.564±0.002 | 0.467±0.007 | 0.561±0.004 | **0.586±0.008** |
| CIFAR | 0.681±0.010 | 0.681±0.008 | 0.692±0.023 | 0.662±0.027 | 0.681±0.016 | 0.640±0.011 | 0.695±0.035 | **0.697±0.029** |

From the **Table** 5, we can see that when there is no noise, the overall performance does not differ significantly across different methods. The reason could be that since the meta-learning backbone remains the same for all the methods, the meta-model performance upper bound can be similar for different scheduling algorithms, without the presence of other confounding factors (e.g., noise, task distribution skewness). In the practical application scenarios with noisy data, BASS-guided meta-models tend to perform well in presence of task noise and skewness compared with baselines, as presented by our experiments in the main body.

### A.3  Parameter Study for Exploration Coefficient

As in **Eq.** 7 and **Eq.** 9, BASS involves an exploration coefficient $\alpha$ to balance the exploitation-exploration and the two exploration objectives. Here, we conduct the parameter study for the exploration coefficient $\alpha$, and include the results with no exploration (i.e., removing $f_2$).

Table 6: Comparison among different $\alpha$ values [dataset (shot) ; final results $\pm$ standard deviation].

| Algo. \ Data | Drug (1) | Drug (5) | CIFAR100 (1) | CIFAR100 (5) |
|---|---|---|---|---|
| No Exploration | 0.234±0.003 | 0.239±0.012 | 0.256±0.027 | 0.537±0.012 |
| $\alpha = 0.1$ | 0.231±0.005 | 0.233±0.013 | 0.264±0.051 | 0.522±0.024 |
| $\alpha = 0.3$ | 0.228±0.013 | 0.231±0.008 | 0.268±0.047 | 0.528±0.014 |
| $\alpha = 0.5$ | 0.236±0.004 | **0.245±0.006** | **0.272±0.025** | **0.553±0.008** |
| $\alpha = 0.7$ | **0.242±0.012** | 0.227±0.006 | 0.241±0.005 | 0.543±0.021 |
| $\alpha = 1.0$ | 0.236±0.002 | 0.235±0.013 | 0.266±0.006 | 0.537±0.005 |

From the results in **Table** 6, we see that the exploration module can indeed improve the performance of BASS compared with the performance with no exploration. This also fits our initial argument that the greedy algorithm alone can lead to sub-optimal performances of meta-model. By properly choosing the $\alpha$ value, we will be able to achieve a good balance between exploitation and exploration, as well as between the two exploration objectives. Here, setting $\alpha \in [0.5, 0.7]$ will be good enough to achieve satisfactory performances. Meanwhile, we also note that even with no exploration, our BASS still achieves good performances by directly learning the correlation between the adapted meta-parameter and the generalization score, and refining the scheduling strategy based on the status of the meta-model.

### A.4  Running Time Comparison

In Figure 5, we include the running time comparison with baselines. We can see that BASS can achieve significant improvement in terms of the running time, and can take as little as 50% of ATS's running time. The intuition is that our proposed BASS only needs one round of the optimization process to update the meta-model and BASS. On the other hand, from Algorithm 1 of the ATS paper [51], we see that ATS requires two optimization rounds for each meta-training iteration to (1) update the scheduler with the temporal meta-model, and (2) update the actual meta-model respectively. Based on the figure on the RHS, we also see that BASS can achieve a relatively good balance between computational cost and performance.

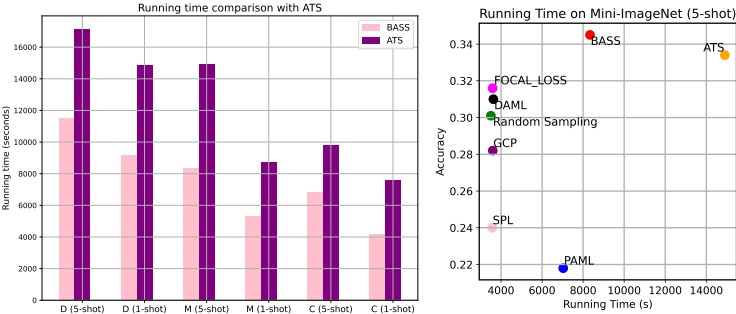

Figure 5: Running time results (including training both the scheduler and the meta-model). "D", "M" and "C" refer to the "Drug", "Mini-ImageNet", "CIFAR-100" data sets respectively. BASS can take as little as approximately 50% of ATS's running time. On the RHS, we have the scatter plot in terms of running time vs. performance on the Mini-ImageNet dataset.

### A.5  Performances with Different Task Skewness Settings

In Table 7, we include the experiments with different levels of skewness. Here, we see that with less skewness levels (the skewness level reduces from Setting 1 to Setting 3), the accuracy of BASS as well as the baselines will continue to improve, while BASS still maintains decent performances.

| Skewness Setting \ Algo. | Uniform | ATS | **BASS** |
|---|---|---|---|
| Skewness Setting 1 | 0.375±0.009 | 0.382±0.007 | 0.408±0.008 |
| Skewness Setting 2 | 0.429±0.012 | 0.448±0.006 | 0.460±0.013 |
| Skewness Setting 3 | 0.497±0.008 | 0.502±0.010 | 0.539±0.009 |

Table 7: Results for different skewness levels on CIFAR-100 data set (5-shot). (1) Setting 1 is the original setting in paper Subsec. 5.2. (2) For Setting 2, we assign 5 tasks with 8% sampling probability, 5 tasks with 3%, and the rest of the tasks equally share the 45% probability. (3) For Setting 3, we assign 5 tasks with 5%, 5 tasks with 2%, while the rest of the tasks equally share the 65% probability.

## A.6   Performances with Different Batch Size

With Table 8, we include additional experiments with different batch sizes $B$, in comparison with the ATS and the uniform sampling approach. Here, we see that with larger $B$ values, the accuracy of BASS as well as the baselines will generally improve.

| $B$ (batch size) \ Algo. | Uniform | ATS | **BASS** |
|---|---|---|---|
| 1 | 0.459±0.009 | 0.449±0.010 | 0.472±0.012 |
| 2 | 0.526±0.011 | 0.515±0.015 | 0.553±0.008 |
| 3 | 0.570±0.012 | 0.563±0.007 | 0.588±0.010 |
| 5 | 0.581±0.005 | 0.571±0.007 | 0.586±0.009 |

Table 8: Results for different $B$ values (batch sizes) on CIFAR-100 data set (5-shot).

## A.7   Performances with Different Embedding Approaches of Arm Contexts

In Table 9, we include additional experiments with different levels of average pooling, such that after the average pooling, the dimensionality of the pooled vector representation will fall into $\{20, 100, 500\}$. We see that overly small dimensionality of the average-pooled vector representation (e.g., 20) can lead to sub-optimal performance of the BASS framework. In addition, we see that setting the dimensionality to 50 can generally lead to good enough performance.

| Dimensionality | 20 | 50 | 100 | 500 |
|---|---|---|---|---|
| Accuracy | 0.541±0.008 | 0.553±0.008 | 0.558±0.006 | 0.555±0.010 |

Table 9: With CIFAR-100 (5-shot), different dimensionality of the average-pooled vector representation (Remark 3) of the meta-parameters.

Here, we also include additional experimental results using MLP to map the original context into the lower dimensional space instead of using our proposed average pooling (Remark 3). Results are shown in Table 10. Here, we use the one-layer MLP with the ReLU activation to embed the original meta-parameters to the low-dimensional vector representations. We can see that the MLP-based method can indeed lead to some performance improvement. But in general, the performance difference between MLP-based embedding and the average-pooling vector representation is subtle. We also note that the MLP-based mapping approach is more time consuming compared with the average pooling approach, since we also need to train the additional embedding layer, which has a considerable number of trainable parameters.

| Dimensionality | Original avg-pooled (50) | 50 | 100 | 200 |
|---|---|---|---|---|
| Accuracy | 0.553±0.008 | 0.558±0.013 | 0.560±0.012 | 0.553±0.015 |

Table 10: With CIFAR-100 (5-shot), different dimensionality of the one-layer MLP(with ReLU)-embedded vector representation of the meta-parameters. "original avg-pooled (50)" refers to the average-pooled vector representation (Remark 3) with dimensionality of 50.

## A.8 Additional Experiments on the "DomainNet" data set

In Table 11, we include additional experiments on the new "DomainNet" data set [37]. Within the "real" domain, we filter 100 classes that have at least 600 images. In this way, with each class being a task with 600 images, we will have a total of 100 tasks. Compared with image data sets in our paper (Mini-ImageNet and CIFAR-100), we increase the image resolution of "DomainNet" by resizing its images to $128 \times 128$ pixels. Following the settings in our paper, we divide tasks into 64 : 16 : 20 portions that correspond to the training set, validation set and the test set respectively. For the few-shot settings, we formulate the problem to be 5-shot, 5-way / 7-way with uniform sampling and ATS as baselines. With a higher image resolution of the "DomainNet" data set, BASS can still maintain the good performance compared with the baselines.

| Setting \ Algo. | Uniform | ATS | **BASS** |
|---|---|---|---|
| 5-way | 0.475±0.002 | 0.483±0.006 | 0.511±0.012 |
| 7-way | 0.411±0.005 | 0.372±0.009 | 0.435±0.008 |

Table 11: Results for the "DomainNet" data set (noise level 0.5, 5-shot settings).

## B Appendix: Additional Discussion on the Necessity of Assumption 5.1

We would like to mention that in order to finish the convergence and generalization analysis for the neural Contextual Bandit works (e.g., [54, 2, 9]), the separateness assumption of the arm context is the minimum requirement of the data set. This is because the training data needs to be non-degenerate (i.e., every pairs of samples are distinct) to ensure that the neural network can consistently converge, as indicated by Assumption 2.1 in [3]. Therefore, our Assumption 5.1 regarding the arm separateness aims to ensure that the BASS is able to adequately learn the underlying reward mapping function with sufficient information. Comparing with the existing works, in the convergence analysis works on meta-learning [47, 48], they measure the arm separateness in terms of the minimum eigenvalue $\lambda_0$ (with $\lambda_0 > 0$) of the Neural Tangent Kernel (NTK) [23] matrix, which is comparable with our Euclidean separateness $\rho$. For existing neural bandit works, Assumption 5.1 in [9] is similar to our separateness assumption. Meanwhile, Assumption 4.2 in [54] and Assumption 3.4 from [53] also imply that no two arms are the same in terms of the minimum NTK matrix eigenvalue $\lambda_0 > 0$.

## C Appendix: Limitation

One potential limitation of BASS is that its improvement over baselines may not be significant when dealing with noise-free settings and non-skewed task distributions (**Table** 5). Meanwhile, the non-adaptive FOCAL-LOSS [30] tends to achieve a similar performance comparing with the adaptive method ATS [51], while enjoying an advantage in terms of the computational cost. In practical terms, although BASS can generally achieve the decent performance and enjoys a smaller computational cost than ATS, the practitioner still needs to consider whether their task distribution is noisy or skewed in order to strike a good balance between the computational resource needed and the meta-model performance, as BASS can achieve a more significant advantage over baselines given the noisy or skewed task distribution.

## D Appendix: Theoretical Analysis

In this section, we present the proof for **Theorem** 5.2. Here, instead of directly going for the batch setting where we adopt training task batch $\Omega_k$ for each iteration $k \in [K]$ ($|\Omega_k| = |\Omega_k^*| = B$), we first introduce the results of the single-task setting (Subsec. D.1), i.e., $|\Omega_k| = |\Omega_k^*| = 1$. Then, the results will be extended to the batch settings as in Subsec. D.2. Recall that for the meta-model, we first consider it to be a $L_\mathcal{F}$-layer fully-connected (FC) network (of width $m_\mathcal{F}$ for the theoretical analysis (lines 237-239). In particular, we follow the settings in [3] for the Gaussian initialization of weight matrices. For the weight matrix elements in meta-model's first $(L_\mathcal{F} - 1)$ layers, we draw each of them from the Gaussian distribution $\mathcal{N}(0, 2/m_\mathcal{F})$. Then, for the weight matrix elements of the last layer ($L_\mathcal{F}$-th layer), we draw each of them from the Gaussian distribution $\mathcal{N}(0, 1)$.

## D.1 Single-task settings

For the brevity of notation, we denote the scheduler output $f(\Theta^{(K-1)}[\mathcal{T}_{k,i}]; \theta^{(k-1)}) = f_1(\chi^q_{k,i}; \theta^{(k-1)}_1) + f_2([\nabla_\theta f_1(\chi^s_{k,i}); \nabla_\theta f_1(\chi^q_{k,i})]; \theta^{(k-1)}_2)$, which corresponds to the definition in **Eq.** 7. In this case, $\mathcal{T}(K) = \{\mathcal{T}_1, \ldots, \mathcal{T}_K\}$ refer to the chosen tasks and $\mathcal{T}^*(K) = \{\mathcal{T}^*_1, \ldots, \mathcal{T}^*_K\}$ are the optimal ones. Based on the problem definition, we will have

$$
\begin{aligned}
R_{\text{single}}(K) &= \mathbb{E}_{\mathcal{T} \sim \mathcal{P}(\mathcal{T}), \boldsymbol{x} \sim \mathcal{D}_{\mathcal{T}}} \left[ \mathcal{L}\big(\boldsymbol{x}; \mathcal{I}(\mathcal{T}, \Theta^{(K)})\big) \right] - \mathbb{E}_{\mathcal{T} \sim \mathcal{P}(\mathcal{T}), \boldsymbol{x} \sim \mathcal{D}_{\mathcal{T}}} \left[ \mathcal{L}\big(\boldsymbol{x}; \mathcal{I}(\mathcal{T}, \Theta^{(K),*})\big) \right] \\
&= \mathbb{E}_{\mathcal{T} \sim \mathcal{P}(\mathcal{T}), \boldsymbol{x} \sim \mathcal{D}_{\mathcal{T}}} \left[ \mathcal{L}\big(\boldsymbol{x}; \mathcal{I}(\mathcal{T}, \Theta^{(K-1)}[\mathcal{T}_K])\big) \right] - \mathbb{E}_{\mathcal{T} \sim \mathcal{P}(\mathcal{T}), \boldsymbol{x} \sim \mathcal{D}_{\mathcal{T}}} \left[ \mathcal{L}\big(\boldsymbol{x}; \mathcal{I}(\mathcal{T}, \Theta^{(K-1),*}[\mathcal{T}^*_K])\big) \right] \\
&= h(\Theta^{(K-1),*}[\mathcal{T}^*_K]) - h(\Theta^{(K-1)}[\mathcal{T}_K]) \\
&= h(\Theta^{(K-1),*}[\mathcal{T}^*_K]) - f(\chi^*_K; \tilde{\theta}^{(K-1)}) + f(\chi^*_K; \tilde{\theta}^{(K-1)}) - f(\chi_K; \theta^{(K-1)}) \\
&\qquad + f(\chi_K; \theta^{(K-1)}) - h(\Theta^{(K-1)}[\mathcal{T}_K]) \\
&\leq h(\Theta^{(K-1),*}[\mathcal{T}^*_K]) - f(\chi^*_K; \tilde{\theta}^{(K-1)}) + f(\chi^*_K; \tilde{\theta}^{(K-1)}) - f(\Theta^{(K-1)}[\mathcal{T}^*_K]; \theta^{(K-1)}) \\
&\qquad + f(\chi_K; \theta^{(K-1)}) - h(\Theta^{(K-1)}[\mathcal{T}_K]) \\
&= h(\Theta^{(K-1),*}[\mathcal{T}^*_K]) - f(\Theta^{(K-1),*}[\mathcal{T}^*_K]; \tilde{\theta}^{(K-1)}) + f(\Theta^{(K-1),*}[\mathcal{T}^*_K]; \tilde{\theta}^{(K-1)}) - f(\Theta^{(K-1)}[\mathcal{T}^*_K]; \theta^{(K-1)}) \\
&\qquad + f(\Theta^{(K-1)}[\mathcal{T}_K]; \theta^{(K-1)}) - h(\Theta^{(K-1)}[\mathcal{T}_K]) \\
&\leq |h(\Theta^{(K-1),*}[\mathcal{T}^*_K]) - f(\Theta^{(K-1),*}[\mathcal{T}^*_K]; \tilde{\theta}^{(K-1)})| + \underbrace{|f(\Theta^{(K-1),*}[\mathcal{T}^*_K]; \tilde{\theta}^{(K-1)}) - f(\Theta^{(K-1)}[\mathcal{T}^*_K]; \theta^{(K-1)})|}_{I_0} \\
&\qquad + |f(\Theta^{(K-1)}[\mathcal{T}_K]; \theta^{(K-1)}) - h(\Theta^{(K-1)}[\mathcal{T}_K])|
\end{aligned}
$$

where the first inequality is due to the arm pulling mechanism, i.e., $f(\Theta^{(K-1)}[\mathcal{T}^*_K]; \theta^{(K-1)}) \leq f(\Theta^{(K-1)}[\mathcal{T}_K]; \theta^{(K-1)})$. Here, $f(\cdot; \tilde{\theta}^{(K-1)})$ is defined as the "shadow" bandit model that are trained on optimal tasks $\{\mathcal{T}^*_1, \mathcal{T}^*_2, \ldots, \mathcal{T}^*_{K-1}\}$ and the corresponding meta-model parameters. Here, denote $\chi_K = \Theta^{(K-1)}[\mathcal{T}_K] \in \mathbb{R}^p$ as the arm context given the arm $\mathcal{T}_K$ and the meta-model parameter $\Theta^{(K-1)}$; similarly, we have $\chi^*_K = \Theta^{(K-1),*}[\mathcal{T}^*_K] \in \mathbb{R}^p$ being the arm context given the arm $\mathcal{T}^*_K$ and the meta-model parameter $\Theta^{(K-1),*}$. Thus, for the term $I_0$ on the RHS, we have

$$
\begin{aligned}
I_0 &= |f(\chi^*; \tilde{\theta}^{(K-1)}) - f(\Theta^{(K-1)}[\mathcal{T}^*_K]; \theta^{(K-1)})| \\
&= |f(\Theta^{(K-1),*}[\mathcal{T}^*_K]; \tilde{\theta}^{(K-1)}) - f(\Theta^{(K-1),*}[\mathcal{T}^*_K]; \theta^{(K-1)}) \\
&\qquad\qquad + f(\Theta^{(K-1),*}[\mathcal{T}^*_K]; \theta^{(K-1)}) - f(\Theta^{(K-1)}[\mathcal{T}^*_K]; \theta^{(K-1)})| \\
&\leq |f(\Theta^{(K-1),*}[\mathcal{T}^*_K]; \tilde{\theta}^{(K-1)}) - f(\Theta^{(K-1),*}[\mathcal{T}^*_K]; \theta^{(K-1)})| \\
&\qquad\qquad + |f(\Theta^{(K-1),*}[\mathcal{T}^*_K]; \theta^{(K-1)}) - f(\Theta^{(K-1)}[\mathcal{T}^*_K]; \theta^{(K-1)})|.
\end{aligned}
$$

Then, inserting the inequality will lead to

$$
\begin{aligned}
R(K) \leq &\underbrace{|h(\Theta^{(K-1)}[\mathcal{T}_K]) - f(\chi_K; \theta^{(K-1)})|}_{I_1} + \underbrace{|f(\chi^*_K; \tilde{\theta}^{(K-1)}) - h(\Theta^{(K-1),*}[\mathcal{T}^*_K])|}_{I_2} \\
&+ \underbrace{|f(\Theta^{(K-1),*}[\mathcal{T}^*_K]; \tilde{\theta}^{(K-1)}) - f(\Theta^{(K-1),*}[\mathcal{T}^*_K]; \theta^{(K-1)})|}_{I_3} \\
&+ \underbrace{|f(\Theta^{(K-1),*}[\mathcal{T}^*_K]; \theta^{(K-1)}) - f(\Theta^{(K-1)}[\mathcal{T}^*_K]; \theta^{(K-1)})|}_{I_4}.
\end{aligned}
$$

Here, the terms $I_1, I_2$ refer to the approximation error for the two bandit models (our possessed model $f(\cdot; \theta^{(K-1)})$ and the pseudo model $f(\cdot; \tilde{\theta}^{(K-1)})$). Then, the third term $I_3$ bounds the output difference when given the same input $\Theta^{(K-1),*}[\mathcal{T}_K]$ to two separate bandit models, and the final

term $I_4$ refers to the difference of the meta-model parameters when adapted to the same task with two individual sets of parameters. Here, the terms $I_1, I_2$ can be bounded by **Lemma** D.1, **Corollary** D.2. Then, the point is to bound the difference term $I_4$ when given different inputs to the bandit model.

### D.1.1 Bounding error terms and assembling the regret bound

[**Bounding term** $I_3$] For error term $I_3$, it focuses on bounding the output difference between two bandit models $f(\cdot; \tilde{\boldsymbol{\theta}}^{(K-1)}), f(\cdot; \boldsymbol{\theta}^{(K-1)})$ given the same input $\boldsymbol{\Theta}^{(K-1),*}[\mathcal{T}_K^*]$, and we have

$$I_3 = |f(\boldsymbol{\Theta}^{(K-1),*}[\mathcal{T}_K^*]; \tilde{\boldsymbol{\theta}}^{(K-1)}) - f(\boldsymbol{\Theta}^{(K-1),*}[\mathcal{T}_K^*]; \boldsymbol{\theta}^{(K-1)})| = |f(\boldsymbol{\chi}_K^*; \tilde{\boldsymbol{\theta}}^{(K-1)}) - f(\boldsymbol{\chi}_K^*; \boldsymbol{\theta}^{(K-1)})|$$

$$\leq \underbrace{|f_1(\boldsymbol{\chi}_K^*; \tilde{\boldsymbol{\theta}}_1^{(K-1)}) - f_1(\boldsymbol{\chi}_K^*; \boldsymbol{\theta}_1^{(K-1)})|}_{I_{3.1}}$$

$$+ \underbrace{|f_2\Big([\nabla_{\tilde{\boldsymbol{\theta}}} f_1(\boldsymbol{\chi}_K^{s,*}); \nabla_{\tilde{\boldsymbol{\theta}}} f_1(\boldsymbol{\chi}_K^{q,*})]; \tilde{\boldsymbol{\theta}}_2^{(K-1)}\Big) - f_2\Big([\nabla_{\boldsymbol{\theta}} f_1(\boldsymbol{\chi}_K^{s,*}); \nabla_{\boldsymbol{\theta}} f_1(\boldsymbol{\chi}_K^{q,*})]; \boldsymbol{\theta}_2^{(K-1)}\Big)|}_{I_{3.2}}.$$

With the defined $\xi_L$, applying **Lemma** D.11 as well as **Corollary** D.12, we will have

$$I_{3.1} \leq \left(1 + \mathcal{O}(\frac{KL^3 \log^{5/6}(m)}{\rho^{1/3} m^{1/6}})\right) \cdot \mathcal{O}(\frac{K^3 L}{\rho \sqrt{m}} \log(m)) + \mathcal{O}\left(\frac{K^4 L^2 \log^{11/6}(m)}{\rho^{4/3} m^{1/6}}\right)$$

Then, for term $I_{3.2}$, we have

$$I_{3.2} = |f_2\Big([\nabla_{\tilde{\boldsymbol{\theta}}} f_1(\boldsymbol{\chi}_K^{s,*}); \nabla_{\tilde{\boldsymbol{\theta}}} f_1(\boldsymbol{\chi}_K^{q,*})]; \tilde{\boldsymbol{\theta}}_2^{(K-1)}\Big) - f_2\Big([\nabla_{\boldsymbol{\theta}} f_1(\boldsymbol{\chi}_K^{s,*}); \nabla_{\boldsymbol{\theta}} f_1(\boldsymbol{\chi}_K^{q,*})]; \boldsymbol{\theta}_2^{(K-1)}\Big)|$$

$$\leq |f_2\Big([\nabla_{\tilde{\boldsymbol{\theta}}} f_1(\boldsymbol{\chi}_K^{s,*}); \nabla_{\tilde{\boldsymbol{\theta}}} f_1(\boldsymbol{\chi}_K^{q,*})]; \tilde{\boldsymbol{\theta}}_2^{(K-1)}\Big) - f_2\Big([\nabla_{\tilde{\boldsymbol{\theta}}} f_1(\boldsymbol{\chi}_K^{s,*}); \nabla_{\tilde{\boldsymbol{\theta}}} f_1(\boldsymbol{\chi}_K^{q,*})]; \boldsymbol{\theta}_2^{(K-1)}\Big)|$$

$$+ |f_2\Big([\nabla_{\tilde{\boldsymbol{\theta}}} f_1(\boldsymbol{\chi}_K^{s,*}); \nabla_{\tilde{\boldsymbol{\theta}}} f_1(\boldsymbol{\chi}_K^{q,*})]; \boldsymbol{\theta}_2^{(K-1)}\Big) - f_2\Big([\nabla_{\boldsymbol{\theta}} f_1(\boldsymbol{\chi}_K^{s,*}); \nabla_{\boldsymbol{\theta}} f_1(\boldsymbol{\chi}_K^{q,*})]; \boldsymbol{\theta}_2^{(K-1)}\Big)|.$$

Here, for the first term on the RHS, we apply **Lemma** D.11 as well as **Corollary** D.12 to bound.

Then, for the second term, with Gaussian initialization of weight matrices, for the over-parameterized FC network $f$ with Lipschitz-smooth activation functions (e.g., Sigmoid), we can have $|f(\boldsymbol{x}) - f(\boldsymbol{x}')|, \|\nabla f(\boldsymbol{x}) - \nabla f(\boldsymbol{x}')\| \leq \xi \cdot \|\boldsymbol{x} - \boldsymbol{x}'\|$ due to its Lipschitz continuity / smoothness property [47, 18]. Meanwhile, we also have the Lipschitz continuity property for over-parameterized FC network $f'$ with ReLU activation [3], such that $|f'(\boldsymbol{x}) - f'(\boldsymbol{x}')| \leq \xi' \cdot \|\boldsymbol{x} - \boldsymbol{x}'\|$. By the Gaussian initialization of BASS's weight matrices and the properties of over-parameterized neural networks [3, 47, 18], we have $\xi_L = \max\{\xi, \xi'\} \leq \mathcal{O}(c_\xi^L)$ being the Lipschitz constant for our $f_1, f_2$, where $c_\xi > 1$ is a small constant. Applying the conclusion above, we will have

$$\Big|f_2\Big([\nabla_{\tilde{\boldsymbol{\theta}}} f_1(\boldsymbol{\chi}_K^{s,*}); \nabla_{\tilde{\boldsymbol{\theta}}} f_1(\boldsymbol{\chi}_K^{q,*})]; \boldsymbol{\theta}_2^{(K-1)}\Big) - f_2\Big([\nabla_{\boldsymbol{\theta}} f_1(\boldsymbol{\chi}_K^{s,*}); \nabla_{\boldsymbol{\theta}} f_1(\boldsymbol{\chi}_K^{q,*})]; \boldsymbol{\theta}_2^{(K-1)}\Big)\Big|$$

$$\leq \xi_L \cdot \big\|[\nabla_{\tilde{\boldsymbol{\theta}}} f_1(\boldsymbol{\chi}_K^{s,*}); \nabla_{\tilde{\boldsymbol{\theta}}} f_1(\boldsymbol{\chi}_K^{q,*})] - \xi_L \cdot [\nabla_{\boldsymbol{\theta}} f_1(\boldsymbol{\chi}_K^{s,*}); \nabla_{\boldsymbol{\theta}} f_1(\boldsymbol{\chi}_K^{q,*})]\big\|$$

$$\leq \xi_L \cdot \big\|\nabla_{\tilde{\boldsymbol{\theta}}} f_1(\boldsymbol{\chi}_K^{s,*}) - \nabla_{\boldsymbol{\theta}} f_1(\boldsymbol{\chi}_K^{s,*})\big\| + \xi_L \cdot \big\|\nabla_{\boldsymbol{\theta}} f_1(\boldsymbol{\chi}_K^{q,*}) - \nabla_{\tilde{\boldsymbol{\theta}}} f_1(\boldsymbol{\chi}_K^{q,*})\big\|$$

$$\leq \xi_L \cdot \frac{KL^4 \log^{5/6}(m)}{\rho^{1/3} m^{1/6}}$$

where the last inequality is by Theorem 5 in [3] and the proof of **Lemma** D.11. With the above results, it will give us

$$I_3 \leq \left(1 + \mathcal{O}(\frac{KL^3 \log^{5/6}(m)}{\rho^{1/3} m^{1/6}})\right)\mathcal{O}(\frac{K^3 L}{\rho \sqrt{m}} \log(m)) + \mathcal{O}\left(\frac{K^4 L^2 \log^{11/6}(m)}{\rho^{4/3} m^{1/6}}\right) + \frac{\xi_L K L^4 \log^{5/6}(m)}{\rho^{1/3} m^{1/6}}$$

**[Bounding term $I_4$]** On the other hand, applying the analogous procedure for term $I_4$, denoting $\boldsymbol{\chi}_K^* = \boldsymbol{\Theta}^{(K-1),*}[\mathcal{T}_K^*]$ and $\bar{\boldsymbol{\chi}}_K^* = \boldsymbol{\Theta}^{(K-1)}[\mathcal{T}_K^*]$ for the brevity of notation, we can have

$$I_4 = |f(\boldsymbol{\Theta}^{(K-1),*}[\mathcal{T}_K^*]; \boldsymbol{\theta}^{(K-1)}) - f(\boldsymbol{\Theta}^{(K-1)}[\mathcal{T}_K^*]; \boldsymbol{\theta}^{(K-1)})|$$

$$\leq \underbrace{\xi_L \cdot \|\boldsymbol{\Theta}^{(K-1),*}[\mathcal{T}_K^*] - \boldsymbol{\Theta}^{(K-1)}[\mathcal{T}_K^*]\|_2}_{I_{4.1}}$$

$$+ \underbrace{|f_2\Big([\nabla_{\boldsymbol{\theta}}f_1(\boldsymbol{\chi}_K^{s,*}); \ \nabla_{\boldsymbol{\theta}}f_1(\boldsymbol{\chi}_K^{q,*})]; \boldsymbol{\theta}_2^{(K-1)}\Big) - f_2\Big([\nabla_{\boldsymbol{\theta}}f_1(\bar{\boldsymbol{\chi}}_K^{s,*}); \ \nabla_{\boldsymbol{\theta}}f_1(\bar{\boldsymbol{\chi}}_K^{q,*})]; \boldsymbol{\theta}_2^{(K-1)}\Big)|}_{I_{4.2}}.$$

where $\boldsymbol{\chi}_K^{s,*}, \boldsymbol{\chi}_K^{q,*}$ respectively represents the support set and query set for task $\mathcal{T}_K^*$ and the meta-parameters $\boldsymbol{\Theta}^{(K-1),*}$. Similar notation also applies to $\bar{\boldsymbol{\chi}}_K^* = \boldsymbol{\Theta}^{(K-1)}[\mathcal{T}_K^*]$. And the inequality is due to the fact that ReLU networks are naturally Lipschitz continuous w.r.t. some coefficient $\xi_L$ when they are wide enough [3], as we have discussed above.

**[Bounding term $I_{4.1}$]** Based on the meta-optimization procedure (inner-loop optimization + outer-loop optimization), we have

$$I_{4.1} = \xi_L \cdot \|\boldsymbol{\Theta}^{(K-1),*}[\mathcal{T}_K^*] - \boldsymbol{\Theta}^{(K-1)}[\mathcal{T}_K^*]\|_2$$

$$= \xi_L \cdot \|\Big(\boldsymbol{\Theta}^{(K-1),*} - \eta_2 \cdot \nabla_{\boldsymbol{\Theta}}\mathcal{L}(D_K^{q,*}; \boldsymbol{\Theta}_K^{(J),*})\Big) - \Big(\boldsymbol{\Theta}^{(K-1)} - \eta_2 \cdot \nabla_{\boldsymbol{\Theta}}\mathcal{L}(D_K^{q,*}; \boldsymbol{\Theta}_K^{(J)})\Big)\|_2$$

where $\boldsymbol{\Theta}_K^{(J),*}$ is the task-specific parameter of $\mathcal{T}_K^*$ after adapting on $\boldsymbol{\Theta}^{(K-1),*}$ with inner-loop optimization, and the $\boldsymbol{\Theta}_K^{(J)}$ is the similar parameter after adapting on $\boldsymbol{\Theta}^{(K-1)}$. Here, we simplify the formula by representing the gradient derivation (inner-loop + outer-loop) with the mapping $H : \mathcal{T} \times \boldsymbol{\Theta} \mapsto \mathbb{R}^p$, which leads to

$$\|\boldsymbol{\Theta}^{(K-1),*}[\mathcal{T}_K^*] - \boldsymbol{\Theta}^{(K-1)}[\mathcal{T}_K^*]\|_2$$

$$= \|\Big(\boldsymbol{\Theta}^{(K-1),*} - \eta_2 \cdot \nabla_{\boldsymbol{\Theta}}\mathcal{L}(D^q; \boldsymbol{\Theta}_K^{(J),*})\Big) - \Big(\boldsymbol{\Theta}^{(K-1)} - \eta_2 \cdot \nabla_{\boldsymbol{\Theta}}\mathcal{L}(D^q; \boldsymbol{\Theta}_K^{(J)})\Big)\|_2$$

$$= \|(\boldsymbol{\Theta}^{(K-1),*} - \boldsymbol{\Theta}^{(K-1)}) - \eta_2 \cdot \Big(H(\mathcal{T}_K^*, \boldsymbol{\Theta}^{(K-1),*}) - H(\mathcal{T}_K^*, \boldsymbol{\Theta}^{(K-1)})\Big)\|_2$$

$$= \|(\boldsymbol{\Theta}^{(K-2),*} - \boldsymbol{\Theta}^{(K-2)}) - \eta_2 \cdot \Big(H(\mathcal{T}_K^*, \boldsymbol{\Theta}^{(K-1),*}) - H(\mathcal{T}_K^*, \boldsymbol{\Theta}^{(K-1)})\Big)$$

$$- \eta_2 \cdot \Big(H(\mathcal{T}_{K-1}^*, \boldsymbol{\Theta}^{(K-2),*}) - H(\mathcal{T}_{K-1}^*, \boldsymbol{\Theta}^{(K-2)})\Big)\|_2$$

$$\leq \sum_{k \in [K]} \eta_2 \cdot \big\|H(\mathcal{T}_k^*, \boldsymbol{\Theta}^{(k-1),*}) - H(\mathcal{T}_k^*, \boldsymbol{\Theta}^{(k-1)})\big\|_2$$

Recall that the past arms, including the actual chosen arms $\{\mathcal{T}_1, \mathcal{T}_2, \ldots, \mathcal{T}_K\}$ as well as the optimal ones $\{\mathcal{T}_1^*, \mathcal{T}_2^*, \ldots, \mathcal{T}_K^*\}$ are all from the candidate pool where each candidate arm is drawn i.i.d. from the task distribution $\mathcal{P}(\mathcal{T})$. Therefore, denoting the bound as $\|H(\mathcal{T}_K^*, \boldsymbol{\Theta}^{(K-1),*}) - H(\mathcal{T}_K^*, \boldsymbol{\Theta}^{(K-1)})\|_2 \leq S_1(K)$, we can have the upper bound as $I_{4.1} \leq \eta_2 \cdot \xi_L K \cdot S_1(K)$.

Then, for the term $S_1(K)$, by definition we have $\|H(\mathcal{T}_K^*, \boldsymbol{\Theta}^{(K-1),*}) - H(\mathcal{T}_K^*, \boldsymbol{\Theta}^{(K-1)})\| \leq S_1(K)$, applying mean-reduction for the sample loss will further leads to

$$\|H(\mathcal{T}_K^*, \boldsymbol{\Theta}^{(K-1),*}) - H(\mathcal{T}_K^*, \boldsymbol{\Theta}^{(K-1)})\| = \|\nabla_{\boldsymbol{\Theta}}\mathcal{L}(D_K^{q,*}; \boldsymbol{\Theta}_K^{(J),*}) - \nabla_{\boldsymbol{\Theta}}\mathcal{L}(D_K^{q,*}; \boldsymbol{\Theta}_K^{(J)})\|_2$$

$$= \|\frac{1}{|D_K^{q,*}|}\sum_{\boldsymbol{x} \in D_K^{q,*}} \nabla_{\boldsymbol{\Theta}}\mathcal{L}(\boldsymbol{x}; \boldsymbol{\Theta}_K^{(J),*}) - \frac{1}{|D_K^{q,*}|}\sum_{\boldsymbol{x} \in D_K^{q,*}} \nabla_{\boldsymbol{\Theta}}\mathcal{L}(\boldsymbol{x}; \boldsymbol{\Theta}_K^{(J)})\|_2$$

$$\leq \frac{1}{|D_K^{q,*}|}\sum_{\boldsymbol{x} \in D_K^{q,*}} \|\nabla_{\boldsymbol{\Theta}}\mathcal{L}(\boldsymbol{x}; \boldsymbol{\Theta}_K^{(J),*}) - \nabla_{\boldsymbol{\Theta}}\mathcal{L}(\boldsymbol{x}; \boldsymbol{\Theta}_K^{(J)})\|_2.$$

This inequality essentially bound the gradient difference when given the same input task $\mathcal{T}_K^*$ w.r.t. different sets of model parameters. Based on the conclusion from Lemma 9 of [47] and Lemma

B.3 of [11], we have $\|\nabla_{\boldsymbol{\Theta}_l} f(x; \boldsymbol{\Theta}_K)\|_F, \|\nabla_{\boldsymbol{\Theta}_l}\mathcal{L}(x; \boldsymbol{\Theta}_K)\|_F \leq \mathcal{O}(\sqrt{m_{\mathcal{F}}}), \forall l \in [L_{\mathcal{F}}]$ for any set of parameters within the sphere $\boldsymbol{\Theta}_K \in \mathcal{B}(\boldsymbol{\Theta}_0, \omega)$ where $\boldsymbol{\Theta}_0$ is the center and $\omega$ is the corresponding radius (which is a small value). With a total of $L_{\mathcal{F}}$ layers for the meta-model and each layer of $m_{\mathcal{F}}$ hidden units, this will give us $\|\nabla_{\boldsymbol{\Theta}}\mathcal{L}(\boldsymbol{x}; \boldsymbol{\Theta}_K^{(J),*})\|_2, \|\nabla_{\boldsymbol{\Theta}}\mathcal{L}(\boldsymbol{x}; \boldsymbol{\Theta}_K^{(J)})\|_2 \leq \mathcal{O}(\sqrt{m_{\mathcal{F}} L_{\mathcal{F}}})$ (**Lemma** D.15). And this makes $S_1(K) \leq \mathcal{O}(\sqrt{m_{\mathcal{F}} L_{\mathcal{F}}})$. Since we have $\eta_1, \eta_2 \leq \mathcal{O}(\frac{1}{m_{\mathcal{F}}})$, summarizing the results above, the upper bound can then be derived.

**[Bounding term $I_{4.2}$]** Next, we proceed to bound $I_{4.2}$, which will be

$$
\begin{aligned}
I_{4.2} &= |f_2\bigg([\nabla_{\boldsymbol{\theta}} f_1(\boldsymbol{\chi}_K^{s,*}); \nabla_{\boldsymbol{\theta}} f_1(\boldsymbol{\chi}_K^{q,*})]; \boldsymbol{\theta}_2^{(K-1)}\bigg) - f_2\bigg([\nabla_{\boldsymbol{\theta}} f_1(\bar{\boldsymbol{\chi}}_K^{s,*}); \nabla_{\boldsymbol{\theta}} f_1(\bar{\boldsymbol{\chi}}_K^{q,*})]; \boldsymbol{\theta}_2^{(K-1)}\bigg)| \\
&\leq \xi_L \cdot \big\|[\nabla_{\boldsymbol{\theta}} f_1(\boldsymbol{\chi}_K^{s,*}); \nabla_{\boldsymbol{\theta}} f_1(\boldsymbol{\chi}_K^{q,*})] - [\nabla_{\boldsymbol{\theta}} f_1(\bar{\boldsymbol{\chi}}_K^{s,*}); \nabla_{\boldsymbol{\theta}} f_1(\bar{\boldsymbol{\chi}}_K^{q,*})]\big\|_2 \\
&\leq \xi_L \cdot \big\|\nabla_{\boldsymbol{\theta}} f_1(\boldsymbol{\chi}_K^{s,*}) - \nabla_{\boldsymbol{\theta}} f_1(\bar{\boldsymbol{\chi}}_K^{s,*})\big\|_2 + \xi_L \cdot \big\|\nabla_{\boldsymbol{\theta}} f_1(\boldsymbol{\chi}_K^{q,*}) - \nabla_{\boldsymbol{\theta}} f_1(\bar{\boldsymbol{\chi}}_K^{q,*})\big\|_2 \\
&\leq \xi_L^2 \cdot \big\|\boldsymbol{\chi}_K^{s,*} - \bar{\boldsymbol{\chi}}_K^{s,*}\big\|_2 + \xi_L^2 \cdot \big\|\boldsymbol{\chi}_K^{q,*} - \bar{\boldsymbol{\chi}}_K^{q,*}\big\|_2
\end{aligned}
$$

where the inequalities are due to the Lipschitz continuity / smoothness properties of over-parameterized FC networks as we discussed above. Here, we notice that the second term on the RHS can be bounded by directly applying the proving procedure of term $I_{4.1}$. Then, for the first term on the RHS, we can following a similar procedure as for $I_{4.1}$, by

$$
\begin{aligned}
\big\|\boldsymbol{\chi}_K^{s,*} - \bar{\boldsymbol{\chi}}_K^{s,*}\big\|_2 &= \big\|\mathcal{I}(\mathcal{T}_k^*, \boldsymbol{\Theta}^{(k-1),*}) - \mathcal{I}(\mathcal{T}_k^*, \boldsymbol{\Theta}^{(k-1)})\big\|_2 \\
&= \|\bigg(\boldsymbol{\Theta}^{(K-1),*} - \eta_1 \cdot \sum_{j\in[J]} \nabla_{\boldsymbol{\Theta}}\mathcal{L}(D_K^{s,*}; \boldsymbol{\Theta}_K^{(j),*})\bigg) - \bigg(\boldsymbol{\Theta}^{(K-1)} - \eta_1 \cdot \sum_{j\in[J]} \nabla_{\boldsymbol{\Theta}}\mathcal{L}(D_K^{s,*}; \boldsymbol{\Theta}_K^{(j)})\bigg)\|_2 \\
&= \|(\boldsymbol{\Theta}^{(K-2),*} - \boldsymbol{\Theta}^{(K-2)})) - (\eta_1 \cdot \sum_{j\in[J]} \nabla_{\boldsymbol{\Theta}}\mathcal{L}(D_{K-1}^{s,*}; \boldsymbol{\Theta}_{K-1}^{(j)}) - \eta_1 \cdot \sum_{j\in[J]} \nabla_{\boldsymbol{\Theta}}\mathcal{L}(D_{K-1}^{s,*}; \boldsymbol{\Theta}_{K-1}^{(j),*}) \\
&\qquad - (\eta_1 \cdot \sum_{j\in[J]} \nabla_{\boldsymbol{\Theta}}\mathcal{L}(D_K^{s,*}; \boldsymbol{\Theta}_K^{(j)}) - \eta_1 \cdot \sum_{j\in[J]} \nabla_{\boldsymbol{\Theta}}\mathcal{L}(D_K^{s,*}; \boldsymbol{\Theta}_K^{(j),*})\|_2 \\
&\leq \eta_1 \cdot \sum_{k\in[K]} \|\sum_{j\in[J]} \nabla_{\boldsymbol{\Theta}}\mathcal{L}(D_k^{s,*}; \boldsymbol{\Theta}_k^{(j)}) - \eta_1 \cdot \sum_{j\in[J]} \nabla_{\boldsymbol{\Theta}}\mathcal{L}(D_k^{s,*}; \boldsymbol{\Theta}_k^{(j),*})\|_2 \\
&\leq \mathcal{O}(\eta_1 \cdot KJ\sqrt{m_{\mathcal{F}} L_{\mathcal{F}}})
\end{aligned}
$$

where the last inequality is due to **Lemma** D.15 and by iterating through $K$ meta-training iterations. Summing up the results above, we will have $I_{4.2} \leq \mathcal{O}(\eta_2\xi_L^2 \cdot K\sqrt{m_{\mathcal{F}} L_{\mathcal{F}}}) + \mathcal{O}(\eta_1\xi_L^2 \cdot KJ\sqrt{m_{\mathcal{F}} L_{\mathcal{F}}})$.

**[Summing up the results]** Then, combining all the results, we would have

$$
R_{\text{single}}(K) \leq \mathcal{O}(\frac{1}{\sqrt{K}}) \cdot \bigg(\sqrt{2\xi_1} + \frac{3L}{\sqrt{2}} + (1+2\gamma_1)\sqrt{2\log(\frac{K}{\delta})}\bigg) + \mathcal{O}(\frac{\xi_L^2 KJ\sqrt{L_{\mathcal{F}}}}{\sqrt{m_{\mathcal{F}}}}) + \gamma_m
$$

where

$$
\gamma_1 = 2 + \mathcal{O}\left(\frac{K^3 L}{\rho\sqrt{m}}\log m\right) + \mathcal{O}\left(\frac{L^2 K^4}{\rho^{4/3}m^{1/6}}\log^{11/6}(m)\right)
$$

$$
\gamma_m = \left(1 + \mathcal{O}(\frac{KL^3\log^{5/6}(m)}{\rho^{1/3}m^{1/6}})\right)\mathcal{O}(\frac{K^3 L}{\rho\sqrt{m}}\log(m)) + \mathcal{O}\left(\frac{K^4 L^2\log^{11/6}(m)}{\rho^{4/3}m^{1/6}}\right) + \frac{\xi_L KL^4\log^{5/6}(m)}{\rho^{1/3}m^{1/6}}
$$

Note that the majority of the terms above can be cancelled to $\mathcal{O}(1)$ with proper networks width $m$ indicated in **Theorem** 5.2. With increasingly large network width $m$, these terms will also become diminutive enough to achieve our regret bound in the main body.

## D.2 Extending the result to the batch settings (Proof of Theorem 5.2)

With the results and conclusions from Subsection D.1, we proceed to provide the proof of **Theorem** 5.2 under the batch settings. Recall that in our original problem formulation and Algorithm 1, we are expected to select a batch of $B$ arms in each meta-training iteration, denoted by $\{\Omega_k\}_{k\in[K]}$. Note that each of the candidate arms from $\Omega_{\text{task}}^{(k)}$ are drawn i.i.d. from the task distribution $\mathcal{P}(\mathcal{T})$. Meantime,

we will have the corresponding optimal arm batches, denoted by $\{\Omega_k^*\}_{k\in[K]}$, which minimizes the loss objective in **Eq.** 5. Recall that we update the meta-model parameters with

$$\boldsymbol{\Theta}^{(k)} = \boldsymbol{\Theta}^{(k-1)} - \eta_2 \cdot \nabla_{\boldsymbol{\Theta}}\left(\frac{1}{|\Omega_k|}\sum_{\mathcal{T}_{k,i}\in\Omega_k}\mathcal{L}(D_{k,i}^q;\boldsymbol{\Theta}_{k,i}^{(J)})\right) = \boldsymbol{\Theta}^{(k-1)} - \frac{\eta_2}{|\Omega_k|}\sum_{\mathcal{T}_{k,i}\in\Omega_k}\nabla_{\boldsymbol{\Theta}}\left(\mathcal{L}(D_{k,i}^q;\boldsymbol{\Theta}_{k,i}^{(J)})\right)$$

where $\boldsymbol{\Theta}_{k,i}^{(J)}$ is the task-specific parameter for $\mathcal{T}_{k,i}$ after the inner-loop optimization for $J$ steps.

Analogously, for the notation brevity and the sake of analysis, we denote $f(\boldsymbol{\Theta}^{(K-1)}[\Omega_K];\boldsymbol{\theta}^{(k-1)}) = f_1(\boldsymbol{\chi}_k^q;\boldsymbol{\theta}_1^{(k-1)}) + f_2\left([\nabla_{\boldsymbol{\theta}}f_1(\boldsymbol{\chi}_k^s);\nabla_{\boldsymbol{\theta}}f_1(\boldsymbol{\chi}_k^q)];\boldsymbol{\theta}_2^{(k-1)}\right)$ where we have $\boldsymbol{\chi}_k^q := \boldsymbol{\Theta}^{(K-1)}[\Omega_K]$ being the meta-parameters adapted to batch of tasks $\Omega_K$, and the batch-specific parameter is defined as $\boldsymbol{\chi}_k^s := \frac{1}{|\Omega_K|}\sum_{\mathcal{T}_{k,\hat{i}}\in\Omega_K}[\boldsymbol{\Theta}_{k,\hat{i}}^{(J)}]$. Then, the regret under the batch setting can be denoted by

$$R(K) = R_{\text{batch}}(K) = \mathbb{E}_{\mathcal{T}\sim\mathcal{P}(\mathcal{T}),\boldsymbol{x}\sim\mathcal{D}_{\mathcal{T}}}\left[\mathcal{L}\big(\boldsymbol{x};\mathcal{I}(\mathcal{T},\boldsymbol{\Theta}^{(K)})\big)\right] - \mathbb{E}_{\mathcal{T}\sim\mathcal{P}(\mathcal{T}),\boldsymbol{x}\sim\mathcal{D}_{\mathcal{T}}}\left[\mathcal{L}\big(\boldsymbol{x};\mathcal{I}(\mathcal{T},\boldsymbol{\Theta}^{(K),*})\big)\right]$$

$$= \mathbb{E}_{\mathcal{T}\sim\mathcal{P}(\mathcal{T}),\boldsymbol{x}\sim\mathcal{D}_{\mathcal{T}}}\left[\mathcal{L}\big(\boldsymbol{x};\mathcal{I}(\mathcal{T},\boldsymbol{\Theta}^{(K-1)}[\Omega_K])\big)\right] - \mathbb{E}_{\mathcal{T}\sim\mathcal{P}(\mathcal{T}),\boldsymbol{x}\sim\mathcal{D}_{\mathcal{T}}}\left[\mathcal{L}\big(\boldsymbol{x};\mathcal{I}(\mathcal{T},\boldsymbol{\Theta}^{(K-1),*}[\Omega_K^*])\big)\right]$$

$$= h(\boldsymbol{\Theta}^{(K-1),*}[\Omega_K^*]) - h(\boldsymbol{\Theta}^{(K-1)}[\Omega_K])$$

$$= h(\boldsymbol{\Theta}^{(K-1),*}[\Omega_K^*]) - f(\boldsymbol{\Theta}^{(K-1),*}[\Omega_K^*];\tilde{\boldsymbol{\theta}}^{(K-1)}) + f(\boldsymbol{\Theta}^{(K-1),*}[\Omega_K^*];\tilde{\boldsymbol{\theta}}^{(K-1)}) - f(\boldsymbol{\Theta}^{(K-1)}[\Omega_K];\boldsymbol{\theta}^{(K-1)})$$

$$+ f(\boldsymbol{\Theta}^{(K-1)}[\Omega_K];\boldsymbol{\theta}^{(K-1)}) - h(\boldsymbol{\Theta}^{(K-1)}[\Omega_K]),$$

and after applying properties of the arm pulling mechanism, it is equivalent to

$$R(K) \le h(\boldsymbol{\Theta}^{(K-1),*}[\Omega_K^*]) - f(\boldsymbol{\Theta}^{(K-1),*}[\Omega_K^*];\tilde{\boldsymbol{\theta}}^{(K-1)})$$

$$+ f(\boldsymbol{\Theta}^{(K-1),*}[\Omega_K^*];\tilde{\boldsymbol{\theta}}^{(K-1)}) - \hat{f}(\boldsymbol{\Theta}^{(K-1),*}[\Omega_K^*];\tilde{\boldsymbol{\theta}}^{(K-1)})$$

$$+ \hat{f}(\boldsymbol{\Theta}^{(K-1),*}[\Omega_K^*];\tilde{\boldsymbol{\theta}}^{(K-1)}) - \hat{f}(\boldsymbol{\Theta}^{(K-1)}[\Omega_K^*];\boldsymbol{\theta}^{(K-1)})$$

$$+ \hat{f}(\boldsymbol{\Theta}^{(K-1)}[\Omega_K];\boldsymbol{\theta}^{(K-1)}) - f(\boldsymbol{\Theta}^{(K-1)}[\Omega_K];\boldsymbol{\theta}^{(K-1)}) + f(\boldsymbol{\Theta}^{(K-1)}[\Omega_K];\boldsymbol{\theta}^{(K-1)}) - h(\boldsymbol{\Theta}^{(K-1)}[\Omega_K])$$

$$\le \underbrace{|h(\boldsymbol{\Theta}^{(K-1),*}[\Omega_K^*]) - f(\boldsymbol{\Theta}^{(K-1),*}[\Omega_K^*];\tilde{\boldsymbol{\theta}}^{(K-1)})|}_{I_5} + \underbrace{|f(\boldsymbol{\Theta}^{(K-1),*}[\Omega_K^*];\tilde{\boldsymbol{\theta}}^{(K-1)}) - \hat{f}(\boldsymbol{\Theta}^{(K-1),*}[\Omega_K^*];\tilde{\boldsymbol{\theta}}^{(K-1)})|}_{I_6}$$

$$+ \underbrace{|\hat{f}(\boldsymbol{\Theta}^{(K-1),*}[\Omega_K^*];\tilde{\boldsymbol{\theta}}^{(K-1)}) - \hat{f}(\boldsymbol{\Theta}^{(K-1)}[\Omega_K^*];\boldsymbol{\theta}^{(K-1)})|}_{I_7} + \underbrace{|\hat{f}(\boldsymbol{\Theta}^{(K-1)}[\Omega_K];\boldsymbol{\theta}^{(K-1)}) - f(\boldsymbol{\Theta}^{(K-1)}[\Omega_K];\boldsymbol{\theta}^{(K-1)})|}_{I_8}$$

$$+ \underbrace{|f(\boldsymbol{\Theta}^{(K-1)}[\Omega_K];\boldsymbol{\theta}^{(K-1)}) - h(\boldsymbol{\Theta}^{(K-1)}[\Omega_K])|}_{I_9}$$

where the average value of estimated benefit scores for individual tasks $\mathcal{T}_{K,i}\in\Omega_K$ is represented as $\hat{f}(\boldsymbol{\Theta}^{(K-1)}[\Omega_K]) = \frac{1}{|\Omega_K|}\cdot\sum_{\mathcal{T}_{K,i}\in\Omega_K}f(\boldsymbol{\Theta}^{(K-1)}[\mathcal{T}_{K,i}]) = \frac{1}{|\Omega_K|}\cdot\sum_{\mathcal{T}_{K,i}\in\Omega_K}f(\boldsymbol{\Theta}^{(k-1)} - \eta_2\cdot\nabla_{\boldsymbol{\Theta}}\mathcal{L}(D_{K,i}^q;\boldsymbol{\Theta}_{K,i}^{(J)});\boldsymbol{\theta}^{(K-1)})$, and the inequality is due to the pulling mechanism of BASS. Here, $I_5, I_9$ individually correspond to $I_1, I_2$ in the single-task setting and can be bounded by **Lemma** D.3, **Corollary** D.4. Term $I_7$ can be upper bounded by $I_3 + I_4$ from the single-task setting above. Then, for the rest terms $I_6, I_8$, we proceed to bound them separately.

### D.2.1 Bounding error terms and assembling the regret bound

We begin with the term $I_8$, and then proceed to $I_6$. For the chosen batch of tasks $\Omega_K$ in the round $K$, we will have $f_1(\boldsymbol{\Theta}^{(K-1)}[\Omega_K]) = f_1\big(\boldsymbol{\Theta}^{(K-1)} - \eta_2\cdot\nabla_{\boldsymbol{\Theta}}\big(\frac{1}{|\Omega_K|}\sum_{\mathcal{T}_{K,i}\in\Omega_K}\mathcal{L}(D_{K,i}^q;\boldsymbol{\Theta}_{K,i}^{(J)})\big);\boldsymbol{\theta}_1^{(K-1)}\big)$, In this case, the average value of estimation sampling probabilities for tasks $\mathcal{T}_{K,i}\in\Omega_K$ is

$$\hat{f}(\boldsymbol{\Theta}^{(K-1)}[\Omega_K]) = \hat{f}_1(\boldsymbol{\Theta}^{(K-1)}[\Omega_K]) + \hat{f}_2(\boldsymbol{\Theta}^{(K-1)}[\Omega_K])$$

$$= \frac{1}{|\Omega_K|}\sum_{\mathcal{T}_{K,i}\in\Omega_K}\left[f_1(\boldsymbol{\Theta}^{(K-1)}[\mathcal{T}_{K,i}];\boldsymbol{\theta}_1^{(K-1)}) + f_2\big([\nabla_{\boldsymbol{\theta}}f_1(\boldsymbol{\chi}_{K,i}^s);\ \nabla_{\boldsymbol{\theta}}f_1(\boldsymbol{\chi}_{K,i}^q)];\boldsymbol{\theta}_2^{(K-1)}\big)\right]$$

**[Bounding the $f_1$ output difference]** Next, let us first proceed to bound the output difference with respect to the exploitation module $f_1$, where we can transform this term into

$$f_1(\mathbf{\Theta}^{(K-1)}[\Omega_K]) - \widehat{f}_1(\mathbf{\Theta}^{(K-1)}[\Omega_K])$$

$$= f_1\bigg(\mathbf{\Theta}_1^{(k-1)} - \eta_2 \cdot \nabla_{\mathbf{\Theta}}\big(\frac{1}{|\Omega_K|}\sum_{\mathcal{T}_{K,i}\in\Omega_K}\mathcal{L}(D_{K,i}^q; \mathbf{\Theta}_{K,i}^{(J)})\big); \boldsymbol{\theta}_1^{(K-1)}\bigg)$$

$$- \frac{1}{|\Omega_K|}\cdot\sum_{\mathcal{T}_{K,j}\in\Omega_K} f_1\bigg(\mathbf{\Theta}^{(k-1)} - \eta_2\cdot\nabla_{\mathbf{\Theta}}\mathcal{L}(D_{K,j}^q; \mathbf{\Theta}_{K,j}^{(J)}); \boldsymbol{\theta}_1^{(K-1)}\bigg)$$

$$= \frac{1}{|\Omega_K|}\cdot\sum_{\mathcal{T}_{K,j}\in\Omega_K}\bigg(f_1\big(\mathbf{\Theta}^{(k-1)} - \eta_2\cdot\nabla_{\mathbf{\Theta}}\big(\frac{1}{|\Omega_K|}\sum_{\mathcal{T}_{K,i}\in\Omega_K}\mathcal{L}(D_{K,i}^q; \mathbf{\Theta}_{K,i}^{(J)})\big); \boldsymbol{\theta}_1^{(K-1)}\big)$$

$$- f_1\big(\mathbf{\Theta}^{(k-1)} - \eta_2\cdot\nabla_{\mathbf{\Theta}}\mathcal{L}(D_{K,j}^q; \mathbf{\Theta}_{K,j}^{(J)}); \boldsymbol{\theta}_1^{(K-1)}\big)\bigg).$$

Then, applying the Lipschitz continuity property will lead to

$$f_1(\mathbf{\Theta}^{(K-1)}[\Omega_K]) - \widehat{f}_1(\mathbf{\Theta}^{(K-1)}[\Omega_K])$$

$$\leq \frac{\eta_2\cdot\xi_L}{|\Omega_K|}\cdot\sum_{\mathcal{T}_{K,i}\in\Omega_K}\|\nabla_{\mathbf{\Theta}}\big(\frac{1}{|\Omega_K|}\sum_{\mathcal{T}_{K,j}\in\Omega_K}\mathcal{L}(D_{K,j}^q; \mathbf{\Theta}_{K,j}^{(J)})\big) - \nabla_{\mathbf{\Theta}}\mathcal{L}(D_{K,i}^q; \mathbf{\Theta}_{K,i}^{(J)})\|_2.$$

Here, by the definition of the outer-loop optimization of first-order meta-learning, we will have an alternative form the inequality, denoted by

$$f_1(\mathbf{\Theta}^{(K-1)}[\Omega_K]) - \widehat{f}_1(\mathbf{\Theta}^{(K-1)}[\Omega_K]) \leq$$

$$\frac{\eta_2\cdot\xi_L}{|\Omega_K|}\cdot\bigg(\sum_{\mathcal{T}_{K,i}\in\Omega_K}\|\frac{1}{|\Omega_K|}\sum_{\mathcal{T}_{K,j}\in\Omega_K}\nabla_{\mathbf{\Theta}}\big(\mathcal{L}(D_{K,j}^q; \mathbf{\Theta}_{K,j}^{(J)})\big) - \nabla_{\mathbf{\Theta}}\mathcal{L}(D_{K,i}^q; \mathbf{\Theta}_{K,i}^{(J)})\|_2\bigg).$$

For the term in the parentheses on the RHS, substituting the backward operation with the $H(\cdot,\cdot)$ mapping function, we have

$$\sum_{\mathcal{T}_{K,i}\in\Omega_K}\|\frac{1}{|\Omega_K|}\sum_{\mathcal{T}_{K,j}\in\Omega_K}\nabla_{\mathbf{\Theta}}\big(\mathcal{L}(D_{K,j}^q; \mathbf{\Theta}_{K,j}^{(J)})\big) - \nabla_{\mathbf{\Theta}}\mathcal{L}(D_{K,i}^q; \mathbf{\Theta}_{K,i}^{(J)})\|_2$$

$$= \sum_{\mathcal{T}_{K,i}\in\Omega_K}\|\frac{1}{|\Omega_K|}\sum_{\mathcal{T}_{K,j}\in\Omega_K}\nabla_{\mathbf{\Theta}}\big(\mathcal{L}(D_{K,j}^q; \mathbf{\Theta}_{K,j}^{(J)})\big) - \frac{1}{|\Omega_K|}\sum_{\mathcal{T}_{K,j}\in\Omega_K}\nabla_{\mathbf{\Theta}}\mathcal{L}(D_{K,i}^q; \mathbf{\Theta}_{K,i}^{(J)})\|_2$$

$$\leq \frac{1}{|\Omega_K|}\sum_{\mathcal{T}_{K,i}\in\Omega_K}\sum_{\mathcal{T}_{K,j}\in\Omega_K}\|\nabla_{\mathbf{\Theta}}\mathcal{L}(D_{K,j}^q; \mathbf{\Theta}_{K,j}^{(J)}) - \nabla_{\mathbf{\Theta}}\mathcal{L}(D_{K,i}^q; \mathbf{\Theta}_{K,i}^{(J)})\|_2$$

$$= \frac{1}{|\Omega_K|}\sum_{\mathcal{T}_{K,i}\in\Omega_K}\sum_{\mathcal{T}_{K,j}\in\Omega_K}\|H(\mathcal{T}_{K,i}, \mathbf{\Theta}^{(K-1)}) - H(\mathcal{T}_{K,j}, \mathbf{\Theta}^{(K-1)})\|_2$$

$$\leq |\Omega_K|\cdot S_1(K).$$

with $|\Omega_K| = B$. Therefore, the $f_1$ part of error term $I_8$ could be bounded by $f_1(\mathbf{\Theta}^{(K-1)}[\Omega_K]) - \widehat{f}_1(\mathbf{\Theta}^{(K-1)}[\Omega_K]) \leq \eta_2\cdot\xi_L\cdot B\cdot S_1(K)$. where the upper bound $S_1(K) \leq \mathcal{O}(\sqrt{m_{\mathcal{F}}L_{\mathcal{F}}})$ can be found in the procedure bounding term $I_{4.1}$.

**[Bounding the $f_2$ output difference]** Then, with $\boldsymbol{\chi}_K = \mathbf{\Theta}^{(K-1)}[\Omega_K]$, we proceed to bound the output difference with respect to the exploration module, which is represented by

$$f_2(\mathbf{\Theta}^{(K-1)}[\Omega_K]) - \widehat{f}_2(\mathbf{\Theta}^{(K-1)}[\Omega_K])$$

$$= f_2\big([\nabla_{\boldsymbol{\theta}}f_1(\boldsymbol{\chi}_K^s); \nabla_{\boldsymbol{\theta}}f_1(\boldsymbol{\chi}_K^q)]; \boldsymbol{\theta}_2^{(K-1)}\big) - \frac{1}{|\Omega_K|}\cdot\sum_{\mathcal{T}_{K,j}\in\Omega_K}f_2\big([\nabla_{\boldsymbol{\theta}}f_1(\boldsymbol{\chi}_{K,j}^s); \nabla_{\boldsymbol{\theta}}f_1(\boldsymbol{\chi}_{K,j}^q)]; \boldsymbol{\theta}_2^{(K-1)}\big)$$

$$= \frac{1}{|\Omega_K|}\cdot\sum_{\mathcal{T}_{K,j}\in\Omega_K}\bigg(f_2\big([\nabla_{\boldsymbol{\theta}}f_1(\boldsymbol{\chi}_K^s); \nabla_{\boldsymbol{\theta}}f_1(\boldsymbol{\chi}_K^q)]; \boldsymbol{\theta}_2^{(K-1)}\big) - f_2\big([\nabla_{\boldsymbol{\theta}}f_1(\boldsymbol{\chi}_{K,j}^s); \nabla_{\boldsymbol{\theta}}f_1(\boldsymbol{\chi}_{K,j}^q)]; \boldsymbol{\theta}_2^{(K-1)}\big)\bigg)$$

By adopting the Lipschitz continuity property of $f_2$, we will have

$$f_2(\mathbf{\Theta}^{(K-1)}[\Omega_K]) - \widehat{f}_2(\mathbf{\Theta}^{(K-1)}[\Omega_K])$$

$$\leq \frac{\xi_L}{|\Omega_K|} \cdot \sum_{\mathcal{T}_{K,j} \in \Omega_K} \left\| [\nabla_{\boldsymbol{\theta}} f_1(\boldsymbol{\chi}_K^s); \nabla_{\boldsymbol{\theta}} f_1(\boldsymbol{\chi}_K^q)] - [\nabla_{\boldsymbol{\theta}} f_1(\boldsymbol{\chi}_{K,j}^s); \nabla_{\boldsymbol{\theta}} f_1(\boldsymbol{\chi}_{K,j}^q)] \right\|_2$$

$$\leq \frac{\xi_L}{|\Omega_K|} \cdot \sum_{\mathcal{T}_{K,j} \in \Omega_K} \left\| \nabla_{\boldsymbol{\theta}} f_1(\boldsymbol{\chi}_K^s) - \nabla_{\boldsymbol{\theta}} f_1(\boldsymbol{\chi}_{K,j}^s) \right\|_2 + \left\| \nabla_{\boldsymbol{\theta}} f_1(\boldsymbol{\chi}_K^q) - \nabla_{\boldsymbol{\theta}} f_1(\boldsymbol{\chi}_{K,j}^q) \right\|_2$$

$$\leq \frac{\xi_L^2}{|\Omega_K|} \cdot \sum_{\mathcal{T}_{K,j} \in \Omega_K} \left\| \boldsymbol{\chi}_K^s - \boldsymbol{\chi}_{K,j}^s \right\|_2 + \left\| \boldsymbol{\chi}_K^q - \boldsymbol{\chi}_{K,j}^q \right\|_2$$

$$= \frac{\xi_L^2 \cdot \eta_2}{|\Omega_K|} \cdot \sum_{\mathcal{T}_{K,j} \in \Omega_K} \left\| \nabla_{\mathbf{\Theta}} \left( \frac{1}{|\Omega_K|} \sum_{\mathcal{T}_{K,i} \in \Omega_K} \mathcal{L}(D_{K,i}^q; \mathbf{\Theta}_{K,i}^{(J)}) \right) - \nabla_{\mathbf{\Theta}} \mathcal{L}(D_{K,j}^q; \mathbf{\Theta}_{K,j}^{(J)}) \right\|_2$$

$$+ \frac{\xi_L^2}{|\Omega_K|} \cdot \sum_{\mathcal{T}_{K,j} \in \Omega_K} \left\| \boldsymbol{\chi}_K^s - \boldsymbol{\chi}_{K,j}^s \right\|_2$$

$$\leq \eta_2 \cdot \xi_L^2 B \cdot S_1(K) + \frac{\xi_L^2 \cdot \eta_1}{|\Omega_K|} \cdot \sum_{\mathcal{T}_{K,j} \in \Omega_K} \left\| \frac{1}{|\Omega_K|} \sum_{\mathcal{T}_{K,i} \in \Omega_K} \mathbf{\Theta}_{K,i}^{(J)} - \frac{1}{|\Omega_K|} \sum_{\mathcal{T}_{K,i} \in \Omega_K} \mathbf{\Theta}_{K,j}^{(J)} \right\|_2$$

where the last inequality is by applying the conclusion when bounding the output difference w.r.t. the exploitation module $f_1$. Then, for the second term on the RHS,

$$\sum_{\mathcal{T}_{K,j} \in \Omega_K} \left\| \frac{1}{|\Omega_K|} \sum_{\mathcal{T}_{K,i} \in \Omega_K} \mathbf{\Theta}_{K,i}^{(J)} - \frac{1}{|\Omega_K|} \sum_{\mathcal{T}_{K,i} \in \Omega_K} \mathbf{\Theta}_{K,j}^{(J)} \right\|_2 \leq \frac{1}{|\Omega_K|} \sum_{\mathcal{T}_{K,j} \in \Omega_K} \sum_{\mathcal{T}_{K,i} \in \Omega_K} \left\| \mathbf{\Theta}_{K,i}^{(J)} - \mathbf{\Theta}_{K,j}^{(J)} \right\|_2$$

$$\leq \frac{1}{|\Omega_K|} \sum_{\mathcal{T}_{K,j} \in \Omega_K} \sum_{\mathcal{T}_{K,i} \in \Omega_K} \left\| (\mathbf{\Theta}^{K-1} - \sum_{\tau \in [\tau]} \nabla_{\mathbf{\Theta}} \mathcal{L}(D_{K,i}^s; \mathbf{\Theta}_{K,i}^{(\tau)})) - (\mathbf{\Theta}^{K-1} - \sum_{\tau \in [J]} \nabla_{\mathbf{\Theta}} \mathcal{L}(D_{K,j}^s; \mathbf{\Theta}_{K,j}^{(\tau)})) \right\|_2$$

$$= \frac{1}{|\Omega_K|} \sum_{\mathcal{T}_{K,j} \in \Omega_K} \sum_{\mathcal{T}_{K,i} \in \Omega_K} \left\| \sum_{\tau \in [J]} \nabla_{\mathbf{\Theta}} \mathcal{L}(D_{K,i}^s; \mathbf{\Theta}_{K,i}^{(\tau)}) - \sum_{\tau \in [J]} \nabla_{\mathbf{\Theta}} \mathcal{L}(D_{K,j}^s; \mathbf{\Theta}_{K,j}^{(\tau)}) \right\|_2$$

$$\leq |\Omega_K| J \cdot \mathcal{O}(\sqrt{m_{\mathcal{F}} L_{\mathcal{F}}})$$

where the last inequality is due to **Lemma** D.15. Summing up all the results above will give us the upper bound for $f_2$ output difference $f_2(\mathbf{\Theta}^{(K-1)}[\Omega_K]) - \widehat{f}_2(\mathbf{\Theta}^{(K-1)}[\Omega_K]) \leq \mathcal{O}(\eta_2 \cdot \xi_L^2 B \cdot \sqrt{m_{\mathcal{F}} L_{\mathcal{F}}} + \eta_1 \cdot BJ \cdot \sqrt{m_{\mathcal{F}} L_{\mathcal{F}}})$.

[**Similar procedure for term** $I_6$] Analogously, we can apply the same derivation for the error term $I_6$, which leads to

$$I_6 = f(\mathbf{\Theta}^{(K-1),*}[\Omega_K^*]; \tilde{\boldsymbol{\theta}}^{(K-1)}) - \widehat{f}(\mathbf{\Theta}^{(K-1),*}[\Omega_K^*]; \tilde{\boldsymbol{\theta}}^{(K-1)})$$

$$= f_1 \left( \mathbf{\Theta}^{(k-1),*} - \eta_2 \nabla_{\mathbf{\Theta}} \left( \frac{1}{|\Omega_K^*|} \sum_{\mathcal{T}_{i*} \in \Omega_K^*} \mathcal{L}(D_{i*}^q; \mathbf{\Theta}_{i*}^{(J)}) \right); \tilde{\boldsymbol{\theta}}_1^{(K-1)} \right)$$

$$- \frac{1}{|\Omega_K^*|} \cdot \sum_{\mathcal{T}_{i*} \in \Omega_K^*} f_1 \left( \mathbf{\Theta}^{(k-1),*} - \eta_2 \nabla_{\mathbf{\Theta}} \mathcal{L}(D_{i*}^q; \mathbf{\Theta}_{i*}^{(J)}); \tilde{\boldsymbol{\theta}}_1^{(K-1)} \right).$$

Following a similar procedure as that of term $I_8$ will give us a similar bound as

$$I_6 = f(\mathbf{\Theta}^{(K-1),*}[\Omega_K^*]; \tilde{\boldsymbol{\theta}}^{(K-1)}) - \widehat{f}(\mathbf{\Theta}^{(K-1),*}[\Omega_K^*]; \tilde{\boldsymbol{\theta}}^{(K-1)})$$

$$\leq \mathcal{O}(\eta_2 \cdot \xi_L^2 B \cdot \sqrt{m_{\mathcal{F}} L_{\mathcal{F}}} + BJ \cdot \sqrt{m_{\mathcal{F}} L_{\mathcal{F}}} \eta_1).$$

where the learning rate $\eta_1, \eta_2 \leq \mathcal{O}(\frac{1}{m_{\mathcal{F}}})$ is a small value. Then, the upper bounds for error terms $I_6, I_8$ are given as desired.

[**Assembling the results**] Then, combining all the results, we would have

$$R(K) \leq \mathcal{O}(\frac{1}{\sqrt{K}}) \cdot \left( \sqrt{2\xi_1} + \frac{3L}{\sqrt{2}} + (1 + 2\gamma_1)\sqrt{2\log(\frac{K}{\delta})} \right) + \mathcal{O}(\frac{\xi_L^2 KBJ\sqrt{L_{\mathcal{F}}}}{\sqrt{m_{\mathcal{F}}}}) + \gamma_m$$

where

$$\gamma_1 = 2 + \mathcal{O}\left(\frac{K^3 L}{\rho\sqrt{m}}\log m\right) + \mathcal{O}\left(\frac{L^2 K^4}{\rho^{4/3}m^{1/6}}\log^{11/6}(m)\right)$$

$$\gamma_m = \left(1 + \mathcal{O}(\frac{KL^3\log^{5/6}(m)}{\rho^{1/3}m^{1/6}})\right)\mathcal{O}(\frac{K^3 L}{\rho\sqrt{m}}\log(m)) + \mathcal{O}\left(\frac{K^4 L^2 \log^{11/6}(m)}{\rho^{4/3}m^{1/6}}\right) + \frac{\xi_L K L^4 \log^{5/6}(m)}{\rho^{1/3}m^{1/6}}$$

Similarly, with proper networks width $m$ as in **Theorem** 5.2, the majority of the terms above can be cancelled to $\mathcal{O}(1)$. With increasingly large network width $m$ under the over-parameterization settings, $\gamma_1, \gamma_m$ will also become diminutive.

### D.3 Performance Guarantee for the Exploitation and Exploration Modules

In this subsection, we would like to give the performance guarantee for the proposed BASS framework, and the corresponding performance bound can be applied to derive an upper bound for the error terms $I_1, I_2$ for the single-task settings and $I_5, I_9$ under the batch settings. Up to meta-training iteration $k \in [K]$ (before updating the meta-parameters and BASS), we denote all the past records received as $\mathcal{P}_{k-1}$. Before presenting the main lemmas, we first introduce the following operator. Inspired by [3], with two arbitrary vectors $\tilde{\chi}, \chi$ such that $\|\tilde{\chi}\|_2 \leq 1, \|\chi\|_2 = 1$, we have the following operator

$$\phi(\tilde{\chi}, \chi) = (\frac{\tilde{\chi}}{\sqrt{2}}, \frac{\chi}{2}, c) \tag{11}$$

as the concatenation of the two vectors $\frac{\tilde{\chi}}{\sqrt{2}}, \frac{\chi}{2}$ and one constant $c$, where $c = \sqrt{\frac{3}{4} - (\frac{\|\tilde{\chi}\|_2}{\sqrt{2}})^2} \geq \frac{1}{2}$. And this operator transforms the transformed vector into unit norm, $\|\phi(\tilde{\chi}, \chi)\|_2 = 1$. The idea of this operator is to make the gradients $\nabla_{\boldsymbol{\theta}} f_1(\cdot; \boldsymbol{\theta}_1)$ of the exploitation model, which is the input of the exploration model $f_2(\cdot; \boldsymbol{\theta}_2)$, comply with the normalization requirement and the separateness assumption (**Assumption** 5.1). For the sake of analysis, we will adopt this operation in the following proof. Note that this operator is just one possible solution, and our results could be easily generalized to other forms of input gradients under the unit-length and separateness assumption. Similar ideas are also applied in previous works [9]. We begin to bound the single-task settings with the following lemma.

**Lemma D.1.** *For the constants $c'_g > 0$, $0 < \rho \leq \mathcal{O}(\frac{1}{L})$ and $\xi_1 \in (0, 1)$, given the past records $\mathcal{P}_{k-1}$, we suppose $m, \eta_1, \eta_2$ satisfy the conditions in **Theorem** 5.2, and randomly draw the parameter $\{\boldsymbol{\theta}_1^{(k)}, \boldsymbol{\theta}_2^{(k)}\} \sim \{\widetilde{\boldsymbol{\theta}}_1^{(\tau)}, \widetilde{\boldsymbol{\theta}}_2^{(\tau)}\}_{\tau \in [k]}$. Consider the past records $\mathcal{P}_{k-1}$ up to round $k$ are generated by a fixed policy when witness the candidate arms $\{\Omega_{task}^{(\tau)}\}_{\tau \in [k]}$. Then, with probability at least $1 - \delta$ given an arm-reward pair $(\mathcal{T}_{k,\hat{i}}, r_{k,\hat{i}})$, we have*

$$\mathbb{E}_{\mathcal{T}_{k,i} \sim \mathcal{P}(\mathcal{T})}\left[|f_2\left(\phi(\frac{[\nabla_{\boldsymbol{\theta}} f_1(\boldsymbol{\chi}_{k,\hat{i}}^s); \nabla_{\boldsymbol{\theta}} f_1(\boldsymbol{\chi}_{k,\hat{i}}^q)]}{c'_g L}, \boldsymbol{\chi}_{k,\hat{i}}^q); \boldsymbol{\theta}_2^{(k-1)}\right) - \left(r_\tau - f_1(\boldsymbol{\chi}_{k,\hat{i}}^q; \boldsymbol{\theta}_1^{(k-1)})\right)| \mid \Omega_{task}^{(k)}, \mathcal{P}_{k-1}\right]$$

$$\leq \frac{1}{\sqrt{k}} \cdot \left(\sqrt{2\xi_1} + \frac{3L}{\sqrt{2}} + (1 + 2\gamma_1)\sqrt{2\log(\frac{k}{\delta})}\right)$$

*where*

$$\gamma_1 = 2 + \mathcal{O}\left(\frac{k^3 L}{\rho\sqrt{m}}\log m\right) + \mathcal{O}\left(\frac{L^2 k^4}{\rho^{4/3}m^{1/6}}\log^{11/6}(m)\right).$$

**Proof.** The proof of this lemma is inspired by Lemma C.1 from [9]. First, we can derive the output upper bound

$$\left|f_2\left(\phi(\frac{[\nabla_{\boldsymbol{\theta}} f_1(\boldsymbol{\chi}_{k,\hat{i}}^s); \nabla_{\boldsymbol{\theta}} f_1(\boldsymbol{\chi}_{k,\hat{i}}^q)]}{c'_g L}, \boldsymbol{\chi}_{k,\hat{i}}^q); \boldsymbol{\theta}_2^{(k-1)}\right) - \left(r_k - f_1(\boldsymbol{\chi}_{k,\hat{i}}^q; \boldsymbol{\theta}_1^{(k-1)})\right)\right|$$

$$\leq \left|f_2\left(\phi(\frac{[\nabla_{\boldsymbol{\theta}} f_1(\boldsymbol{\chi}_{k,\hat{i}}^s); \nabla_{\boldsymbol{\theta}} f_1(\boldsymbol{\chi}_{k,\hat{i}}^q)]}{c'_g L}, \boldsymbol{\chi}_{k,\hat{i}}^q); \boldsymbol{\theta}_2^{(k-1)}\right)\right| + \left|f_1(\boldsymbol{\chi}_{k,\hat{i}}^q; \boldsymbol{\theta}_1^{(k-1)})\right| + 1$$

$$\leq 1 + 2\gamma_1$$

by triangle inequality and applying the generalization result of FC networks (**Lemma** D.5) on $f_1(\cdot; \boldsymbol{\theta}_1), f_2(\cdot; \boldsymbol{\theta}_2)$.

For the brevity of notation, we use $\nabla f_1(\mathcal{T}_{k,\widehat{i}})$ to denote $\phi(\frac{[\nabla_{\boldsymbol{\theta}} f_1(\boldsymbol{\chi}^s_{k,\widehat{i}}); \nabla_{\boldsymbol{\theta}} f_1(\boldsymbol{\chi}^q_{k,\widehat{i}})]}{c'_g L}, \boldsymbol{\chi}^q_{k,\widehat{i}})$ and apply $(\boldsymbol{\chi}_k, r_k)$ as $(\boldsymbol{\chi}^q_{k,\widehat{i}}, r_{k,\widehat{i}})$ for the following proof. Define the difference sequence as

$$V_\tau^{(1)} = \mathbb{E}\left[\left\|f_2\left(\nabla f_1(\mathcal{T}_{\tau,\widehat{i}}); \boldsymbol{\theta}_2^{(\tau-1)}\right) - \left(r_\tau - f_1(\boldsymbol{\chi}_\tau; \boldsymbol{\theta}_1^{(\tau-1)})\right)\right\|\right]$$
$$- \left|f_2\left(\nabla f_1(\mathcal{T}_{\tau,\widehat{i}}); \boldsymbol{\theta}_2^{(\tau-1)}\right) - \left(r_\tau - f_1(\boldsymbol{\chi}_\tau; \boldsymbol{\theta}_1^{(\tau-1)})\right)\right|.$$

Since the past rewards and the received arm-reward pairs $(\boldsymbol{\chi}_\tau, r_\tau)$ are generated by the same reward mapping function, we have the expectation

$$\mathbb{E}[V_\tau^{(1)}|F_\tau] = \mathbb{E}\left[\left\|f_2\left(\nabla f_1(\mathcal{T}_{\tau,\widehat{i}}); \boldsymbol{\theta}_2^{(\tau-1)}\right) - \left(r_\tau - f_1(\boldsymbol{\chi}_\tau; \boldsymbol{\theta}_1^{(\tau-1)})\right)\right\|\right]$$
$$- \mathbb{E}\left[\left\|f_2\left(\nabla f_1(\mathcal{T}_{\tau,\widehat{i}}); \boldsymbol{\theta}_2^{(\tau-1)}\right) - \left(r_\tau - f_1(\boldsymbol{\chi}_\tau; \boldsymbol{\theta}_1^{(\tau-1)})\right)\right\| \Big| F_\tau\right] = 0.$$

where $F_\tau$ denotes the filtration given the past records $\mathcal{P}_\tau$, up to round $\tau \in [k]$. This also gives the fact that $V_\tau^{(1)}$ is a martingale difference sequence. Then, after applying the martingale difference sequence over $[k]$, we have

$$\frac{1}{k}\sum_{\tau\in[k]} V_\tau^{(1)} = \frac{1}{k}\sum_{\tau\in[k]} \mathbb{E}\left[\left\|f_2\left(\nabla f_1(\mathcal{T}_{\tau,\widehat{i}}); \boldsymbol{\theta}_2^{(\tau-1)}\right) - \left(r_\tau - f_1(\boldsymbol{\chi}_\tau; \boldsymbol{\theta}_1^{(\tau-1)})\right)\right\|\right]$$
$$- \frac{1}{k}\sum_{\tau\in[k]}\left|f_2\left(\nabla f_1(\mathcal{T}_{\tau,\widehat{i}}); \boldsymbol{\theta}_2^{(\tau-1)}\right) - \left(r_\tau - f_1(\boldsymbol{\chi}_\tau; \boldsymbol{\theta}_1^{(\tau-1)})\right)\right|.$$

Then, by applying the Azuma-Hoeffding inequality, it leads to

$$\mathbb{P}\left[\frac{1}{k}\sum_{\tau\in[k]} V_\tau^{(1)} - \frac{1}{k}\sum_{\tau\in[k]} \mathbb{E}[V_\tau^{(1)}] \geq (1+2\gamma_1)\sqrt{\frac{2\log(1/\delta)}{k}}\right] \leq \delta$$

Since the expectation of $V_\tau^{(1)}$ is zero, with the probability at least $1 - \delta$ and an existing set of parameters $\boldsymbol{\theta}_2$ s.t. $\|\boldsymbol{\theta}_2 - \boldsymbol{\theta}_2^{(0)}\| \leq \mathcal{O}\left(\frac{k^3}{\rho\sqrt{m}}\log m\right)$, the above inequality implies

$$\frac{1}{k}\sum_{\tau\in[k]} V_\tau^{(1)} \leq (1+2\gamma_1)\sqrt{\frac{2\log(1/\delta)}{k}} \implies$$

$$\mathbb{E}_{\mathcal{T}_{k,i}\sim\mathcal{P}(\mathcal{T})}\mathbb{E}_{\{\boldsymbol{\theta}_1^{(k-1)},\boldsymbol{\theta}_2^{(k-1)}\}}\left[\left\|f_2\left(\nabla f_1(\mathcal{T}_{\tau,\widehat{i}}); \boldsymbol{\theta}_2^{(k-1)}\right) - \left(r_\tau - f_1(\boldsymbol{\chi}_\tau; \boldsymbol{\theta}_1^{(k-1)})\right)\right\|\right]$$
$$= \frac{1}{k}\sum_{\tau\in[k]} \mathbb{E}\left[\left\|f_2\left(\nabla f_1(\mathcal{T}_{\tau,\widehat{i}}); \boldsymbol{\theta}_2^{(\tau-1)}\right) - \left(r_k - f_1(\boldsymbol{\chi}_\tau; \boldsymbol{\theta}_1^{(\tau-1)})\right)\right\|\right]$$
$$\leq \frac{1}{k}\sum_{\tau\in[k]} \left|f_2\left(\nabla f_1(\mathcal{T}_{\tau,\widehat{i}}); \boldsymbol{\theta}_2^{(\tau-1)}\right) - \left(r_\tau - f_1(\boldsymbol{\chi}_\tau; \boldsymbol{\theta}_1^{(\tau-1)})\right)\right| + (1+2\gamma_1)\sqrt{\frac{2\log(1/\delta)}{k}}$$
$$\underset{(i)}{\leq} \frac{1}{k}\sum_{\tau\in[k]} \left|f_2\left(\nabla f_1(\mathcal{T}_{\tau,\widehat{i}}); \boldsymbol{\theta}_2\right) - \left(r_\tau - f_1(\boldsymbol{\chi}_\tau; \boldsymbol{\theta}_1^{(\tau-1)})\right)\right| + \frac{3L}{\sqrt{2k}} + (1+2\gamma_1)\sqrt{\frac{2\log(1/\delta)}{k}}$$
$$\leq \frac{1}{\sqrt{k}}\sqrt{\sum_{\tau\in[k]} \left|f_2\left(\nabla f_1(\mathcal{T}_{\tau,\widehat{i}}); \boldsymbol{\theta}_2\right) - \left(r_\tau - f_1(\boldsymbol{\chi}_\tau; \boldsymbol{\theta}_1^{(\tau-1)})\right)\right|^2} + \frac{3L}{\sqrt{2k}} + (1+2\gamma_1)\sqrt{\frac{2\log(1/\delta)}{k}}$$
$$\underset{(ii)}{\leq} \sqrt{\frac{2\xi_1}{k}} + \frac{3L}{\sqrt{2k}} + (1+2\gamma_1)\sqrt{\frac{2\log(1/\delta)}{k}}.$$

where the first equality is due to the sampling of candidate tasks and the model parameters. Here, the upper bound (i) is derived by applying the conclusions of **Lemma** D.6 and **Lemma** D.10, and the inequality (ii) is derived by adopting **Lemma** D.6 while defining the empirical loss to be $\frac{1}{2}\sum_{\tau\in[k]}\left|f_2\left(\nabla f_1(\mathcal{T}_{\tau,\widehat{i}});\boldsymbol{\theta}_2\right)-\left(r_\tau-f_1(\boldsymbol{\chi}_\tau;\boldsymbol{\theta}_1^{(\tau-1)})\right)\right|^2\leq\xi_1$. Finally, applying the union bound would give the aforementioned results.

$\square$

Here, analogous to the trained parameters, we consider the shadow parameters as $\{\boldsymbol{\theta}_1^{(k),*},\boldsymbol{\theta}_2^{(k),*}\}\sim\{\widetilde{\boldsymbol{\theta}}_1^{(\tau),*},\widetilde{\boldsymbol{\theta}}_2^{(\tau),*}\}_{\tau\in[k]}$. Similarly, each pair $\{\widetilde{\boldsymbol{\theta}}_1^{(\tau),*},\widetilde{\boldsymbol{\theta}}_2^{(\tau),*}\}$ is separately trained on past received rewards of the optimal arm(s) $\{r_{\tau',i^*}\}_{\tau'\in[\tau],\mathcal{T}_{\tau',i^*}\in\Omega_k^*}$ and past exploration scores of the optimal arm(s) $\{e_{\tau',i^*}\}_{\tau'\in[\tau],\mathcal{T}_{\tau',i^*}\in\Omega_k^*}$ with $J_{\boldsymbol{\theta}}$-iteration GD, starting from the random initialization $\{\boldsymbol{\theta}_1^{(0)},\boldsymbol{\theta}_2^{(0)}\}$.

**Corollary D.2.** *For the constants $0<\rho\leq\mathcal{O}(1/L)$ and $\xi_1\in(0,1)$, given the past records $\mathcal{P}_{k-1}$, we suppose $m,\eta_1,J$ satisfy the conditions in **Theorem** 5.2, and randomly draw the parameters $\{\boldsymbol{\theta}_1^{(k),*},\boldsymbol{\theta}_2^{(k),*}\}\sim\{\widetilde{\boldsymbol{\theta}}_1^{(\tau),*},\widetilde{\boldsymbol{\theta}}_2^{(\tau),*}\}_{\tau\in[k]}$. For the optimal arm $\mathcal{T}_{k,i^*}\in\Omega_{task}^k$, consider its union set with the the collection of past optimal arms $\mathcal{P}_{k-1}^*\cup\{\mathcal{T}_{k,i^*},r_{k,i^*}\}$ are generated by a fixed policy when witness the candidate arms $\{\Omega_{task}^{(\tau)}\}_{\tau\in[k]}$, with $\mathcal{P}_{k-1}^*$ being the collection chosen by this policy. Then, with probability at least $1-\delta$, we have*

$$\mathbb{E}_{\mathcal{T}_{k,i}\sim\mathcal{P}(\mathcal{T})}\left[|f_2\left(\phi(\frac{[\nabla_{\boldsymbol{\theta}}f_1(\boldsymbol{\chi}_k^{s,*});\ \nabla_{\boldsymbol{\theta}}f_1(\boldsymbol{\chi}_k^{q,*})]}{c_g'L},\boldsymbol{\chi}_k^{q,*});\boldsymbol{\theta}_2^{(k-1),*}\right)-\left(r_\tau-f_1(\boldsymbol{\chi}_k^{q,*};\boldsymbol{\theta}_1^{(k-1),*})\right)|\ |\Omega_{task}^{(k)},\mathcal{P}_{k-1}^*\right]$$

$$\leq\frac{1}{\sqrt{k}}\cdot\left(\sqrt{2\xi_1}+\frac{3L}{\sqrt{2}}+(1+\gamma_1)\sqrt{2\log(\frac{k}{\delta})}\right)+\Gamma_k$$

*where $r_{\tau,i^*}$ is the corresponding reward generated by the mapping function given an arm $\boldsymbol{\chi}_{\tau,i^*}$, and*

$$\Gamma_k=\left(1+\mathcal{O}(\frac{kL^3\log^{5/6}(m)}{\rho^{1/3}m^{1/6}})\right)\cdot\mathcal{O}(\frac{k^4L}{\rho\sqrt{m}}\log(m))+\mathcal{O}\left(\frac{k^5L^2\log^{11/6}(m)}{\rho^{4/3}m^{1/6}}\right).$$

**Proof.** This corollary is the direct application of Lemma D.1 by following a similar proof procedure. First, suppose the shadow models $f_1(\cdot;\boldsymbol{\theta}_2),f_2(\cdot;\boldsymbol{\theta}_2)$ are trained on the alternative trajectory $\mathcal{P}_{k-1}^*$. Analogous to the proof of Lemma D.1, we can define the following martingale difference sequence with regard to the previous records $\mathcal{P}_{k-1}^*$ up to round $\tau\in[t]$ as

$$V_\tau^{(1),*}=\mathbb{E}\left[\left|f_2\left(\nabla f_1(\mathcal{T}_{\tau,i^*});\boldsymbol{\theta}_2^{(\tau-1),*}\right)-\left(r_\tau^*-f_1(\boldsymbol{\chi}_\tau^*;\boldsymbol{\theta}_1^{(\tau-1),*})\right)\right|\right]$$
$$-\left|f_2\left(\nabla f_1(\mathcal{T}_{\tau,i^*});\boldsymbol{\theta}_2^{(\tau-1),*}\right)-\left(r_\tau^*-f_1(\boldsymbol{\chi}_\tau^*;\boldsymbol{\theta}_1^{(\tau-1),*})\right)\right|.$$

Since the records in set $\mathcal{P}_{k-1}^*$ are sharing the same reward mapping function, we have the expectation

$$\mathbb{E}[V_\tau^{(1),*}|F_\tau^*]=\mathbb{E}\left[\left|f_2\left(\nabla f_1(\mathcal{T}_{\tau,i^*});\boldsymbol{\theta}_2^{(\tau-1),*}\right)-\left(r_\tau^*-f_1(\boldsymbol{\chi}_\tau^*;\boldsymbol{\theta}_1^{(\tau-1),*})\right)\right|\right]$$
$$-\mathbb{E}\left[\left|f_2\left(\nabla f_1(\mathcal{T}_{\tau,i^*});\boldsymbol{\theta}_2^{(\tau-1),*}\right)-\left(r_\tau^*-f_1(\boldsymbol{\chi}_\tau^*;\boldsymbol{\theta}_1^{(\tau-1),*})\right)\right|\ |F_\tau^*\right]=0.$$

where $F_\tau^*$ denotes the filtration given the past records $\mathcal{P}_{k-1}^*$. The mean value of $V_\tau^{(1),*}$ across different time steps will be

$$\frac{1}{k}\sum_{\tau\in[k]}V_\tau^{(1),*}=\frac{1}{k}\sum_{\tau\in[k]}\mathbb{E}\left[\left|f_2\left(\nabla f_1(\mathcal{T}_{\tau,i^*});\boldsymbol{\theta}_2^{(\tau-1),*}\right)-\left(r_\tau^*-f_1(\boldsymbol{\chi}_\tau^*;\boldsymbol{\theta}_1^{(\tau-1),*})\right)\right|\right]$$
$$-\frac{1}{k}\sum_{\tau\in[k]}\left|f_2\left(\nabla f_1(\mathcal{T}_{\tau,i^*});\boldsymbol{\theta}_2^{(\tau-1),*}\right)-\left(r_\tau^*-f_1(\boldsymbol{\chi}_\tau^*;\boldsymbol{\theta}_1^{(\tau-1),*})\right)\right|.$$

with the expectation of zero. Afterwards, applying the Azuma-Hoeffding inequality, with a constant $\delta \in (0,1)$, it leads to

$$\mathbb{P}\left[\frac{1}{k}\sum_{\tau\in[k]}V_\tau^{(1),*} - \frac{1}{k}\sum_{\tau\in[k]}\mathbb{E}[V_\tau^{(1),*}] \geq (1+2\gamma_1)\sqrt{\frac{2\log(1/\delta)}{k}}\right] \leq \delta$$

To bound the output difference between the shadow model $f_1(\cdot;\boldsymbol{\theta}_1^{(k-1),*}), f_2(\cdot;\boldsymbol{\theta}_2^{(k-1),*})$ and the model we trained based on received records $f_1(\cdot;\boldsymbol{\theta}_1^{(k-1)}), f_2(\cdot;\boldsymbol{\theta}_2^{(k-1)})$, we apply the conclusion from **Lemma** D.11, which leads to that given arbitrary input vectors $\boldsymbol{x}, \boldsymbol{x}'$, we have

$$|f_1(\boldsymbol{x};\boldsymbol{\theta}_1^{(k-1),*}) - f_1(\boldsymbol{x};\boldsymbol{\theta}_1^{(k-1)})|, \ |f_2(\boldsymbol{x}';\boldsymbol{\theta}_2^{(k-1),*}) - f_2(\boldsymbol{x}';\boldsymbol{\theta}_2^{(k-1)})| \leq$$
$$\left(1 + \mathcal{O}(\frac{kL^3\log^{5/6}(m)}{\rho^{1/3}m^{1/6}})\right) \cdot \mathcal{O}(\frac{k^3L}{\rho\sqrt{m}}\log(m)) + \mathcal{O}\left(\frac{k^4L^2\log^{11/6}(m)}{\rho^{4/3}m^{1/6}}\right).$$

Finally, combining all the results will finish the proof.

$\square$

We will also be able to have the performance guarantee under the batch settings. Recall that given a batch of chosen tasks $\Omega_k \subset \Omega_{\text{task}}^{(k)}, k \in [K]$, we have the meta-parameters adapted to this batch of tasks being $\boldsymbol{\Theta}^{(K-1)}[\Omega_K]$, which we consider as the input for the $f_1(\cdot;\boldsymbol{\theta}_1)$ model, where the tasks within each collection are sampled from the task distribution. Thus, chosen task batches from different iterations are also independent from each other. Intuitively, we can also define the corresponding reward for arm batch $\Omega_k$ as $r_k = h(\boldsymbol{\Theta}^{(k-1)}[\Omega_k])$. Then, we bound the batch settings with the following lemma and corollary.

**Lemma D.3.** *For the constants $c_g' > 0$, $\rho \in (0, \mathcal{O}(\frac{1}{L}))$ and $\xi_1 \in (0,1)$, given the past records $\mathcal{P}_{k-1}$, we suppose $m, \eta_1, \eta_2$ satisfy the conditions in* **Theorem** *5.2, and randomly draw the parameter $\{\boldsymbol{\theta}_1^{(k)}, \boldsymbol{\theta}_2^{(k)}\} \sim \{\widetilde{\boldsymbol{\theta}}_1^{(\tau)}, \widetilde{\boldsymbol{\theta}}_2^{(\tau)}\}_{\tau\in[k]}$. Consider the past records $\mathcal{P}_{k-1}$ up to round $k$ are generated by a fixed policy when witness the candidate arms $\{\Omega_{\text{task}}^{(\tau)}\}_{\tau\in[k]}$. Then, with probability at least $1-\delta$ given the pair of chosen arm batch and the reward $(\Omega_k, r_k)$ in round $k$, we have*

$$\mathbb{E}_{\mathcal{T}_{k,i}\sim\mathcal{P}(\mathcal{T})}\left[|f_2\left(\phi(\frac{[\nabla_{\boldsymbol{\theta}}f_1(\boldsymbol{\chi}_k^s); \ \nabla_{\boldsymbol{\theta}}f_1(\boldsymbol{\chi}_k^q)]}{c_g'L}, \boldsymbol{\chi}_k^q); \boldsymbol{\theta}_2^{(k-1)}\right) - \left(r_\tau - f_1(\boldsymbol{\chi}_k^q;\boldsymbol{\theta}_1^{(k-1)})\right)| \ |\Omega_{\text{task}}^{(k)}, \mathcal{P}_{k-1}\right]$$
$$\leq \frac{1}{\sqrt{k}} \cdot \left(\sqrt{2\xi_1} + \frac{3L}{\sqrt{2}} + (1+2\gamma_1)\sqrt{2\log(\frac{k}{\delta})}\right)$$

*where*

$$\gamma_1 = 2 + \mathcal{O}\left(\frac{k^3L}{\rho\sqrt{m}}\log m\right) + \mathcal{O}\left(\frac{L^2k^4}{\rho^{4/3}m^{1/6}}\log^{11/6}(m)\right).$$

**Proof.** The proof of this lemma is analogous to the proof of **Lemma** D.1. First, we can derive the output upper bound

$$\left|f_2\left(\phi(\frac{[\nabla_{\boldsymbol{\theta}}f_1(\boldsymbol{\chi}_k^s); \ \nabla_{\boldsymbol{\theta}}f_1(\boldsymbol{\chi}_k^q)]}{c_g'L}, \boldsymbol{\chi}_k^q); \boldsymbol{\theta}_2^{(k-1)}\right) - \left(r_k - f_1(\boldsymbol{\chi}_k^q;\boldsymbol{\theta}_1^{(k-1)})\right)\right|$$
$$\leq \left|f_2\left(\phi(\frac{[\nabla_{\boldsymbol{\theta}}f_1(\boldsymbol{\chi}_k^s); \ \nabla_{\boldsymbol{\theta}}f_1(\boldsymbol{\chi}_k^q)]}{c_g'L}, \boldsymbol{\chi}_k^q); \boldsymbol{\theta}_2^{(k-1)}\right)\right| + \left|f_1(\boldsymbol{\chi}_k^q;\boldsymbol{\theta}_1^{(k-1)})\right| + 1$$
$$\leq 1 + 2\gamma_1$$

by triangle inequality and applying the generalization result of FC networks (**Lemma** D.5) on $f_1(\cdot;\boldsymbol{\theta}_1), f_2(\cdot;\boldsymbol{\theta}_2)$, where $c_g' > 0$ is a positive number to scale the concatenated gradient vector.

For the brevity of notation, we use $\nabla f_1(\Omega_k)$ to denote $\phi(\frac{[\nabla_{\boldsymbol{\theta}}f_1(\boldsymbol{\chi}_k^s); \ \nabla_{\boldsymbol{\theta}}f_1(\boldsymbol{\chi}_k^q)]}{c_g'L}, \boldsymbol{\chi}_k^q)$ and apply $(\boldsymbol{\chi}_k, r_k)$ as $(\boldsymbol{\chi}_k^q, r_k)$ for the following proof. Define the difference sequence as

$$V_\tau^{(2)} = \mathbb{E}\left[\left|f_2\left(\nabla f_1(\Omega_\tau); \boldsymbol{\theta}_2^{(\tau-1)}\right) - \left(r_\tau - f_1(\boldsymbol{\chi}_\tau;\boldsymbol{\theta}_1^{(\tau-1)})\right)\right|\right]$$
$$- \left|f_2\left(\nabla f_1(\Omega_\tau); \boldsymbol{\theta}_2^{(\tau-1)}\right) - \left(r_\tau - f_1(\boldsymbol{\chi}_\tau;\boldsymbol{\theta}_1^{(\tau-1)})\right)\right|.$$

Since the past rewards and the received arm batch-reward pairs $(\chi_\tau, r_\tau)$ are generated by the same reward mapping function, we have the expectation

$$
\mathbb{E}[V_\tau^{(2)}|F_\tau] = \mathbb{E}\left[\left\|f_2\left(\nabla f_1(\Omega_\tau); \boldsymbol{\theta}_2^{(\tau-1)}\right) - \left(r_\tau - f_1(\chi_\tau; \boldsymbol{\theta}_1^{(\tau-1)})\right)\right\|\right]
$$
$$
- \mathbb{E}\left[\left\|f_2\left(\nabla f_1(\Omega_\tau); \boldsymbol{\theta}_2^{(\tau-1)}\right) - \left(r_\tau - f_1(\chi_\tau; \boldsymbol{\theta}_1^{(\tau-1)})\right)\right\|\Big| F_\tau\right] = 0.
$$

where $F_\tau$ denotes the filtration given the past records $\mathcal{P}_\tau$, up to round $\tau \in [k]$. This also gives the fact that $V_\tau^{(2)}$ is a martingale difference sequence. Then, after applying the martingale difference sequence over $[k]$, we have

$$
\frac{1}{k}\sum_{\tau\in[k]} V_\tau^{(2)} = \frac{1}{k}\sum_{\tau\in[k]} \mathbb{E}\left[\left\|f_2\left(\nabla f_1(\Omega_\tau); \boldsymbol{\theta}_2^{(\tau-1)}\right) - \left(r_\tau - f_1(\chi_\tau; \boldsymbol{\theta}_1^{(\tau-1)})\right)\right\|\right]
$$
$$
- \frac{1}{k}\sum_{\tau\in[k]}\left|f_2\left(\nabla f_1(\Omega_\tau); \boldsymbol{\theta}_2^{(\tau-1)}\right) - \left(r_\tau - f_1(\chi_\tau; \boldsymbol{\theta}_1^{(\tau-1)})\right)\right|.
$$

By the Azuma-Hoeffding inequality, it leads to $\mathbb{P}\left[\frac{1}{k}\sum_{\tau\in[k]} V_\tau^{(2)} - \frac{1}{k}\sum_{\tau\in[k]} \mathbb{E}[V_\tau^{(2)}] \geq (1 + 2\gamma_1)\sqrt{\frac{2\log(1/\delta)}{k}}\right] \leq \delta$. As we have discussed, the tasks within each collection are sampled from the task distribution, which makes chosen task batches from different iterations $\Omega_k, k \in [K]$ are also independent from each other. Since the expectation of $V_\tau^{(2)}$ is zero, with the probability at least $1 - \delta$ and an existing set of parameters $\boldsymbol{\theta}_2$ s.t. $\|\boldsymbol{\theta}_2 - \boldsymbol{\theta}_2^{(0)}\| \leq \mathcal{O}\left(\frac{k^3}{\rho\sqrt{m}}\log m\right)$, the above inequality implies

$$
\frac{1}{k}\sum_{\tau\in[k]} V_\tau^{(1)} \leq (1 + 2\gamma_1)\sqrt{\frac{2\log(1/\delta)}{k}} \implies
$$

$$
\mathbb{E}_{\mathcal{T}_{k,i}\sim\mathcal{P}(\mathcal{T})}\mathbb{E}_{\{\boldsymbol{\theta}_1^{(k-1)},\boldsymbol{\theta}_2^{(k-1)}\}}\left[\left\|f_2\left(\nabla f_1(\Omega); \boldsymbol{\theta}_2^{(k-1)}\right) - \left(r - f_1(\chi; \boldsymbol{\theta}_1^{(k-1)})\right)\right\|\right]
$$
$$
= \frac{1}{k}\sum_{\tau\in[k]} \mathbb{E}\left[\left\|f_2\left(\nabla f_1(\Omega_\tau); \boldsymbol{\theta}_2^{(\tau-1)}\right) - \left(r_k - f_1(\chi_\tau; \boldsymbol{\theta}_1^{(\tau-1)})\right)\right\|\right]
$$
$$
\leq \frac{1}{k}\sum_{\tau\in[k]}\left|f_2\left(\nabla f_1(\Omega_\tau); \boldsymbol{\theta}_2^{(\tau-1)}\right) - \left(r_\tau - f_1(\chi_\tau; \boldsymbol{\theta}_1^{(\tau-1)})\right)\right| + (1 + 2\gamma_1)\sqrt{\frac{2\log(1/\delta)}{k}}
$$
$$
\underset{(i)}{\leq} \frac{1}{k}\sum_{\tau\in[k]}\left|f_2\left(\nabla f_1(\Omega_\tau); \boldsymbol{\theta}_2\right) - \left(r_\tau - f_1(\chi_\tau; \boldsymbol{\theta}_1^{(\tau-1)})\right)\right| + \frac{3L}{\sqrt{2k}} + (1 + 2\gamma_1)\sqrt{\frac{2\log(1/\delta)}{k}}
$$
$$
\leq \frac{1}{\sqrt{k}}\sqrt{\sum_{\tau\in[k]}\left|f_2\left(\nabla f_1(\Omega_\tau); \boldsymbol{\theta}_2\right) - \left(r_\tau - f_1(\chi_\tau; \boldsymbol{\theta}_1^{(\tau-1)})\right)\right|^2} + \frac{3L}{\sqrt{2k}} + (1 + 2\gamma_1)\sqrt{\frac{2\log(1/\delta)}{k}}
$$
$$
\underset{(ii)}{\leq} \sqrt{\frac{2\xi_2}{k}} + \frac{3L}{\sqrt{2k}} + (1 + 2\gamma_1)\sqrt{\frac{2\log(1/\delta)}{k}}.
$$

where the first equality is due to the sampling of candidate tasks and the model parameters. Here, the upper bound (i) is derived by applying the conclusions of **Lemma** D.6 and **Lemma** D.10, and the inequality (ii) is derived by adopting **Lemma** D.6 while defining the empirical loss to be $\frac{1}{2}\sum_{\tau\in[k]}\left|f_2\left(\nabla f_1(\Omega_\tau); \boldsymbol{\theta}_2\right) - \left(r_\tau - f_1(\chi_\tau; \boldsymbol{\theta}_1^{(\tau-1)})\right)\right|^2 \leq \xi_2$. Finally, applying the union bound would give the aforementioned results.

$\square$

Analogously, we consider the shadow parameters as $\{\boldsymbol{\theta}_1^{(k),*}, \boldsymbol{\theta}_2^{(k),*}\} \sim \{\widetilde{\boldsymbol{\theta}}_1^{(\tau),*}, \widetilde{\boldsymbol{\theta}}_2^{(\tau),*}\}_{\tau \in [k]}$ where each pair $\{\widetilde{\boldsymbol{\theta}}_1^{(\tau),*}, \widetilde{\boldsymbol{\theta}}_2^{(\tau),*}\}$ is separately trained on past received rewards of the optimal arm(s) $\{r_{\tau',i^*}\}_{\tau' \in [\tau], \mathcal{T}_{\tau',i^*} \in \Omega_k^*}$ and past exploration scores of the optimal arm(s) $\{e_{\tau',i^*}\}_{\tau' \in [\tau], \mathcal{T}_{\tau',i^*} \in \Omega_k^*}$ with $J_{\boldsymbol{\theta}}$-iteration GD starting from the random initialization $\{\boldsymbol{\theta}_1^{(0)}, \boldsymbol{\theta}_2^{(0)}\}$.

**Corollary D.4.** *For the constants $\rho \in (0, \mathcal{O}(\frac{1}{L}))$ and $\xi_1 \in (0, 1)$, given the past records $\mathcal{P}_{k-1}$, we suppose $m, \eta_1, J$ satisfy the conditions in **Theorem 5.2**, and randomly draw the parameters $\{\boldsymbol{\theta}_1^{(k),*}, \boldsymbol{\theta}_2^{(k),*}\} \sim \{\widetilde{\boldsymbol{\theta}}_1^{(\tau),*}, \widetilde{\boldsymbol{\theta}}_2^{(\tau),*}\}_{\tau \in [k]}$. For the optimal arm batch $\Omega_k^* \subset \Omega_{task}^*$, consider its union set with the the collection of past optimal arms $\mathcal{P}_{k-1}^* \cup \{\Omega_k^*, r_k^*\}$ are generated by a fixed policy when witness the candidate arms $\{\Omega_{task}^{(\tau)}\}_{\tau \in [k]}$, with $\mathcal{P}_{k-1}^*$ being the collection chosen by this policy. Then, with probability at least $1 - \delta$, we have*

$$\mathbb{E}_{\mathcal{T}_{k,i} \sim \mathcal{P}(\mathcal{T})}\left[|f_2\left(\phi(\frac{[\nabla_{\boldsymbol{\theta}} f_1(\boldsymbol{\chi}_k^{s,*}); \nabla_{\boldsymbol{\theta}} f_1(\boldsymbol{\chi}_k^{q,*})]}{c_g' L}, \boldsymbol{\chi}_k^{q,*}); \boldsymbol{\theta}_2^{(k-1),*}\right) - \left(r_\tau - f_1(\boldsymbol{\chi}_k^{q,*}; \boldsymbol{\theta}_1^{(k-1),*})\right)| \mid \Omega_{task}^{(k)}, \mathcal{P}_{k-1}^*\right]$$

$$\leq \frac{1}{\sqrt{k}} \cdot \left(\sqrt{2\xi_2} + \frac{3L}{\sqrt{2}} + (1 + \gamma_1)\sqrt{2\log(\frac{k}{\delta})}\right) + \Gamma_k$$

*where $r_{\tau,i^*}$ is the corresponding reward generated by the mapping function given an arm $\boldsymbol{\chi}_{\tau,i^*}$, and*

$$\Gamma_k = \left(1 + \mathcal{O}(\frac{kL^3 \log^{5/6}(m)}{\rho^{1/3} m^{1/6}})\right) \cdot \mathcal{O}(\frac{k^4 L}{\rho\sqrt{m}} \log(m)) + \mathcal{O}\left(\frac{k^5 L^2 \log^{11/6}(m)}{\rho^{4/3} m^{1/6}}\right).$$

This corollary is a directly application of **Lemma D.3** and can be obtained with a similar proof as in **Corollary** D.2.

## D.4 Ancillary Lemmas

Applying $\mathcal{P}_{k-1}$ as the training data, we have the following properties for the over-parameterized FC network $f(\cdot; \boldsymbol{\theta})$ after GD.

**Lemma D.5.** *For the constants $\rho \in (0, \mathcal{O}(\frac{1}{L}))$ and $\xi_1 \in (0, 1)$, given the past records $\mathcal{P}_{k-1}$ up to time step $k$, we suppose $m, \eta_1, J_1$ satisfy the conditions in **Theorem 5.2**. Then, with probability at least $1 - \delta$, given a sample-label pair $(\boldsymbol{x}, r)$, we have*

$$|f(\boldsymbol{x}; \boldsymbol{\theta}^{(k)})| \leq \gamma_1 = 2 + \mathcal{O}\left(\frac{k^3 L}{\rho\sqrt{m}} \log m\right) + \mathcal{O}\left(\frac{L^2 k^4}{\rho^{4/3} m^{1/6}} \log^{11/6}(m)\right).$$

**Proof.** The LHS of the inequality could be written as

$$|f(\boldsymbol{x}; \boldsymbol{\theta})| \leq |f(\boldsymbol{x}; \boldsymbol{\theta}) - f(\boldsymbol{x}; \boldsymbol{\theta}^{(0)}) - \langle \nabla_{\boldsymbol{\theta}^{(0)}} f(\boldsymbol{x}; \boldsymbol{\theta}^{(0)}), \boldsymbol{\theta} - \boldsymbol{\theta}^{(0)} \rangle|$$
$$+ |f(\boldsymbol{x}; \boldsymbol{\theta}^{(0)}) + \langle \nabla_{\boldsymbol{\theta}^{(0)}} f(\boldsymbol{x}; \boldsymbol{\theta}^{(0)}), \boldsymbol{\theta} - \boldsymbol{\theta}^{(0)} \rangle|.$$

Here, we could bound the first term on the RHS with **Lemma D.7**. Applying **Lemma D.8** on the second term, and recalling $\|\boldsymbol{\theta} - \boldsymbol{\theta}^{(0)}\|_2 \leq \omega$, would give

$$|f(\boldsymbol{x}; \boldsymbol{\theta})| \leq 2 + \|\nabla_{\boldsymbol{\theta}^{(0)}} f(\boldsymbol{x}; \boldsymbol{\theta}^{(0)})\|_2 \|\boldsymbol{\theta} - \boldsymbol{\theta}^{(0)}\|_2 +$$
$$\mathcal{O}(\omega^{1/3} L^2 \sqrt{m \log(m)}) \cdot \|\boldsymbol{\theta} - \boldsymbol{\theta}^{(0)}\|_2$$
$$\leq 2 + \mathcal{O}(L) \cdot \|\boldsymbol{\theta} - \boldsymbol{\theta}^{(0)}\|_2 + \mathcal{O}(L^2 \sqrt{m \log(m)})(\|\boldsymbol{\theta} - \boldsymbol{\theta}^{(0)}\|_2)^{\frac{4}{3}}.$$

Then, applying the conclusion of **Lemma D.6** would lead to

$$|f(\boldsymbol{x}; \boldsymbol{\theta})| \leq 2 + \mathcal{O}(L) \cdot \mathcal{O}\left(\frac{k^3}{\rho\sqrt{m}} \log m\right) + \mathcal{O}(L^2 \sqrt{m \log(m)}) \left(\mathcal{O}(\frac{k^3}{\rho\sqrt{m}} \log m)\right)^{\frac{4}{3}}$$
$$= 2 + \mathcal{O}\left(\frac{k^3 L}{\rho\sqrt{m}} \log m\right) + \mathcal{O}\left(\frac{L^2 k^4}{\rho^{4/3} m^{1/6}} \log^{11/6}(m)\right) = \gamma_1.$$

$\square$

**Lemma D.6** (Theorem 1 from [3]). *For any $0 < \xi_1 \leq 1$, $0 < \rho \leq \mathcal{O}(\frac{1}{L})$. Given the past records $\mathcal{P}_{k-1}$, suppose $m, \eta_1, J$ satisfy the conditions in **Theorem** 5.2, then with probability at least $1 - \delta$, we could have*

  *1. $\mathcal{L}(\boldsymbol{\theta}) \leq \xi_1$ after $J$ iterations of GD.*

  *2. For any $j \in [J]$, $\|\boldsymbol{\theta}^{(j)} - \boldsymbol{\theta}^{(0)}\| \leq \mathcal{O}\left(\frac{k^3}{\rho\sqrt{m}} \log m\right).$*

In particular, **Lemma** D.6 above provides the convergence guarantee for $f(\cdot; \boldsymbol{\theta})$ after certain rounds of GD training on the past records $\mathcal{P}_{k-1}$.

**Lemma D.7** (Lemma 4.1 in [11]). *Assume a constant $\omega$ such that $\mathcal{O}(m^{-3/2}L^{-3/2}[\log(TnL^2/\delta)]^{3/2}) \leq \omega \leq \mathcal{O}(L^{-6}[\log m]^{-3/2})$ and $n$ training samples. With randomly initialized $\boldsymbol{\theta}^{(0)}$, for parameters $\boldsymbol{\theta}, \boldsymbol{\theta}'$ satisfying $\|\boldsymbol{\theta} - \boldsymbol{\theta}^{(0)}\|, \|\boldsymbol{\theta} - \boldsymbol{\theta}^{(0)}\| \leq \omega$, we have*

$$|f(\boldsymbol{x}; \boldsymbol{\theta}) - f(\boldsymbol{x}; \boldsymbol{\theta}') - \langle \nabla_{\boldsymbol{\theta}'} f(\boldsymbol{x}; \boldsymbol{\theta}'), \boldsymbol{\theta} - \boldsymbol{\theta}'\rangle| \leq \mathcal{O}(\omega^{1/3}L^2\sqrt{m\log(m)})\|\boldsymbol{\theta} - \boldsymbol{\theta}'\|$$

*with the probability at least $1 - \delta$.*

**Lemma D.8.** *Assume $m, \eta_1, J$ satisfy the conditions in **Theorem** 5.2 and $\boldsymbol{\theta}^{(0)}$ is randomly initialized. Then, with probability at least $1 - \delta$ and given an arm $\|\boldsymbol{x}\|_2 = 1$, we have*

  *1. $|f(\boldsymbol{x}; \boldsymbol{\theta}^{(0)})| \leq 2$,*

  *2. $\|\nabla_{\boldsymbol{\theta}^{(0)}} f(\boldsymbol{x}; \boldsymbol{\theta}^{(0)})\|_2 \leq \mathcal{O}(L)$.*

**Proof.** The conclusion (1) is a direct application of Lemma 7.1 in [3]. Suppose the parameters of the $L$-layer FC network are $\boldsymbol{\theta} = \{\boldsymbol{\theta}_1, \ldots, \boldsymbol{\theta}_L\}$. For conclusion (2), applying Lemma 7.3 in [3], for each layer $\boldsymbol{\theta}_l \in \{\boldsymbol{\theta}_1, \ldots, \boldsymbol{\theta}_L\}$, we have

$$\|\nabla_{\boldsymbol{\theta}_l} f(\boldsymbol{x}; \boldsymbol{\theta}^{(0)})\|_2 = \|(\boldsymbol{\theta}_L \boldsymbol{D}_{L-1} \cdots \boldsymbol{D}_{l+1}\boldsymbol{\theta}_{l+1}) \cdot (\boldsymbol{D}_{l+1}\boldsymbol{\theta}_{l+1} \cdots \boldsymbol{D}_1\boldsymbol{\theta}_1) \cdot \boldsymbol{x}^{\mathsf{T}}\|_2 = \mathcal{O}(\sqrt{L}).$$

where $\boldsymbol{D}$ is the diagonal matrix corresponding to the activation function. Then, we could have the conclusion that

$$\|\nabla_{\boldsymbol{\theta}^{(0)}} f(\boldsymbol{x}; \boldsymbol{\theta}^{(0)})\|_2 = \sqrt{\sum_{l \in [L]} \|\nabla_{\boldsymbol{\theta}_l} f(\boldsymbol{x}; \boldsymbol{\theta}^{(0)})\|_2^2} = \mathcal{O}(L).$$

$\square$

**Lemma D.9** (Theorem 5 in [3]). *Assume $m, \eta_1, J$ satisfy the conditions in **Theorem** 5.2 and $\boldsymbol{\theta}^{(0)}$ being randomly initialized. Then, with probability at least $1 - \delta$, and for all parameter $\boldsymbol{\theta}$ such that $\|\boldsymbol{\theta} - \boldsymbol{\theta}^{(0)}\|_2 \leq \omega$, we have*

$$\|\nabla_{\boldsymbol{\theta}} f(\boldsymbol{x}; \boldsymbol{\theta}) - \nabla_{\boldsymbol{\theta}^{(0)}} f(\boldsymbol{x}; \boldsymbol{\theta}^{(0)})\|_2 \leq \mathcal{O}(\omega^{1/3}L^3\sqrt{\log(m)})$$

**Lemma D.10.** *Assume $m, \eta_1$ satisfy the condition in **Theorem** 5.2. For notation brevity, suppose the training sample-label pairs are $\{\boldsymbol{x}_\tau, r_\tau\}_{\tau \in [k]}$. With the probability at least $1 - \delta$, we have*

$$\sum_{\tau \in [k]} |f(\boldsymbol{x}_\tau; \boldsymbol{\theta}^{(\tau)}) - r_\tau| \leq \sum_{\tau \in [k]} |f(\boldsymbol{x}_\tau; \boldsymbol{\theta}^{(k)}) - r_\tau| + \frac{3L\sqrt{2k}}{2}$$

**Proof.** With the notation from Lemma 4.3 in [11], set $R = \frac{k^3 \log(m)}{\delta}$, $\nu = R^2$, and $\epsilon = \frac{LR}{\sqrt{2\nu k}}$. Then, considering the loss function to be $\mathcal{L}(\boldsymbol{\theta}) := \sum_{\tau \in [k]} |f(\boldsymbol{x}_\tau; \boldsymbol{\theta}) - r_\tau|$ would complete the proof. $\square$

**Lemma D.11.** *Consider a randomly initialized $L$-layer ReLU fully-connected network $f(\cdot; \boldsymbol{\theta}_0)$. For any $0 < \xi_2 \leq 1$, $0 < \rho \leq \mathcal{O}(\frac{1}{L})$. Let there be two sets of training samples $\mathcal{P}_k, \mathcal{P}'_k$ with the unit-length and the $\rho$-separateness assumption, and let $\boldsymbol{\theta}$ be the trained parameter on $\mathcal{P}_k$ while $\boldsymbol{\theta}'$ is the trained parameter on $\mathcal{P}'_k$. Suppose the conditions in **Theorem** 5.2 are satisfied. Then, with probability at least $1 - \delta$, we have*

$$|f(\boldsymbol{x}; \boldsymbol{\theta}) - f(\boldsymbol{x}; \boldsymbol{\theta}')| \leq \left(1 + \mathcal{O}(\frac{kL^3 \log^{5/6}(m)}{\rho^{1/3}m^{1/6}})\right) \cdot \mathcal{O}(\frac{k^3 L}{\rho\sqrt{m}} \log(m)) + \mathcal{O}\left(\frac{k^4 L^2 \log^{11/6}(m)}{\rho^{4/3}m^{1/6}}\right)$$

*when given a new sample $\boldsymbol{x} \in \mathbb{R}^d$.*

**Proof.** First, based on the conclusion from Theorem 1 from [3] and regarding the $t$ samples, the trained the parameters satisfy $\|\boldsymbol{\theta} - \boldsymbol{\theta}_0\|_2, \|\boldsymbol{\theta}' - \boldsymbol{\theta}_0\|_2 \leq \mathcal{O}(\frac{k^3}{\rho\sqrt{m}}\log(m)) = \omega$ where $\boldsymbol{\theta}_0$ is the randomly initialized parameter. Then, we could have

$$\|\nabla_{\boldsymbol{\theta}} f(\boldsymbol{x};\boldsymbol{\theta})\|_2 \leq \|\nabla_{\boldsymbol{\theta}_0} f(\boldsymbol{x};\boldsymbol{\theta}_0)\|_2 + \|\nabla_{\boldsymbol{\theta}} f(\boldsymbol{x};\boldsymbol{\theta}) - \nabla_{\boldsymbol{\theta}_0} f(\boldsymbol{x};\boldsymbol{\theta}_0)\|_2$$
$$\leq \left(1 + \mathcal{O}(\frac{kL^3 \log^{5/6}(m)}{\rho^{1/3}m^{1/6}})\right) \cdot \mathcal{O}(L)$$

w.r.t. the conclusion from Theorem 1 and Theorem 5 of [3]. Then, regarding the Lemma 4.1 from [11], we would have

$$|f(\boldsymbol{x};\boldsymbol{\theta}) - f(\boldsymbol{x};\boldsymbol{\theta}') - \langle \nabla_{\boldsymbol{\theta}'} f(\boldsymbol{x};\boldsymbol{\theta}'), \boldsymbol{\theta} - \boldsymbol{\theta}'\rangle| \leq \mathcal{O}(\omega^{1/3}L^2\sqrt{m\log(m)}) \cdot \|\boldsymbol{\theta} - \boldsymbol{\theta}'\|_2.$$

Therefore, the our target could be reformed as

$$|f(\boldsymbol{x};\boldsymbol{\theta}) - f(\boldsymbol{x};\boldsymbol{\theta}')| \leq \|\nabla_{\boldsymbol{\theta}'} f(\boldsymbol{x};\boldsymbol{\theta}')\|_2 \|\boldsymbol{\theta} - \boldsymbol{\theta}'\|_2 + \mathcal{O}(\omega^{1/3}L^2\sqrt{m\log(m)}) \cdot \|\boldsymbol{\theta} - \boldsymbol{\theta}'\|_2$$
$$\leq \left(1 + \mathcal{O}(\frac{kL^3 \log^{5/6}(m)}{\rho^{1/3}m^{1/6}})\right) \cdot \mathcal{O}(L) \cdot \omega + \mathcal{O}(\omega^{4/3}L^2\sqrt{m\log(m)})$$

Substituting the $\omega$ with its value would complete the proof.

$\square$

**Corollary D.12.** *Following a similar settings as in **Lemma** D.11, consider a randomly initialized $L$-layer fully-connected network $f(\cdot;\boldsymbol{\theta}_0)$ with Sigmoid activation. For any $0 < \xi_2 \leq 1, 0 < \rho \leq \mathcal{O}(\frac{1}{L})$. Let there be two sets of training samples $\mathcal{P}_k, \mathcal{P}'_k$ with the unit-length and the $\rho$-separateness assumption, and let $\boldsymbol{\theta}$ be the trained parameter on $\mathcal{P}_k$ while $\boldsymbol{\theta}'$ is the trained parameter on $\mathcal{P}'_k$. Suppose the conditions in **Theorem** 5.2 are satisfied. Then, with probability at least $1 - \delta$, we have*

$$|f(\boldsymbol{x};\boldsymbol{\theta}) - f(\boldsymbol{x};\boldsymbol{\theta}')| \leq \left(1 + \mathcal{O}(\frac{kL^3 \log^{5/6}(m)}{\rho^{1/3}m^{1/6}})\right) \cdot \mathcal{O}(\frac{k^3 L}{\rho\sqrt{m}}\log(m)) + \mathcal{O}\left(\frac{k^4 L^2 \log^{11/6}(m)}{\rho^{4/3}m^{1/6}}\right)$$

*when given a new sample $\boldsymbol{x} \in \mathbb{R}^d$.*

**Proof.** This corollary is an intuitive extension of **Lemma** D.11. Since the result from Theorem 1 of [3] also applies to Lipschitz-smooth (i.e., Sigmoid) activation functions, combining the proof of **Lemma** D.11 and the result from Lemma 7 in [47] will give the conclusion.

$\square$

### D.5 Regret Bound for Uniform Sampling

**Lemma D.13** (Regret Bound for the Uniform Sampling Approach)**.** *When applying the uniform sampling as in most meta-learning frameworks, we denote the corresponding sampled task series as $\Omega_u(K)$. We will have $R_u(K) = \mathbb{E}_{\mathcal{T}\sim\mathcal{P}(\mathcal{T}),\boldsymbol{x}\sim\mathcal{D}_{\mathcal{T}}}\left[\mathcal{L}(\boldsymbol{x};\mathcal{I}(\mathcal{T}, \boldsymbol{\Theta}_u^{(K)})) - \mathcal{L}(\boldsymbol{x};\mathcal{I}(\mathcal{T}, \boldsymbol{\Theta}^{(K),*}))\right]$. where $\boldsymbol{\Theta}_u^{(K)}$ refer to the meta-parameters trained with uniform sampling. With $\|\boldsymbol{\Theta}_u^{(K)} - \boldsymbol{\Theta}^{(K),*}\|_2 \leq \omega$, we have the regret bound for the uniform sampling as*

$$R_u(K) = \mathbb{E}_{\mathcal{T}\sim\mathcal{P}(\mathcal{T}),\boldsymbol{x}\sim\mathcal{D}_{\mathcal{T}}}\left[\mathcal{L}(\boldsymbol{x};\mathcal{I}(\mathcal{T}, \boldsymbol{\Theta}_u^{(K)})) - \mathcal{L}(\boldsymbol{x};\mathcal{I}(\mathcal{T}, \boldsymbol{\Theta}^{(K),*}))\right]$$
$$\leq \sqrt{m_{\mathcal{F}}L_{\mathcal{F}}} \cdot \omega + \mathcal{O}(\omega^{4/3}L_{\mathcal{F}}^3\sqrt{m_{\mathcal{F}}\log(m_{\mathcal{F}})}) + \mathcal{O}(\sqrt{\frac{L_{\mathcal{F}}}{m_{\mathcal{F}}}})$$
$$\leq \min\left\{\mathcal{O}\left(KL_{\mathcal{F}} + \frac{K^{4/3}L_{\mathcal{F}}^{11/3}\sqrt{\log(m_{\mathcal{F}})}}{m_{\mathcal{F}}^{1/6}} + \sqrt{\frac{L_{\mathcal{F}}}{m_{\mathcal{F}}}}\right), 1\right\}$$

**Proof.** Here, for the simplicity of notation, we denote $\boldsymbol{\Theta} = \mathcal{I}(\mathcal{T}, \boldsymbol{\Theta})$, and neglect the expectation terms. Note that the difference between adapted meta-parameters and the original meta-parameters is

small enough and can be well-bounded. We will then have

$$R_u(K) = \mathbb{E}_{\mathcal{T} \sim \mathcal{P}(\mathcal{T}), \boldsymbol{x} \sim \mathcal{D}_{\mathcal{T}}} \left[ \mathcal{L}(\boldsymbol{x}; \mathcal{I}(\mathcal{T}, \boldsymbol{\Theta}_u^{(K)})) - \mathcal{L}(\boldsymbol{x}; \mathcal{I}(\mathcal{T}, \boldsymbol{\Theta}^{(K),*})) \right]$$

$$= \widetilde{\mathcal{L}}(\widetilde{\boldsymbol{\Theta}}_u^{(K)}) - \widetilde{\mathcal{L}}(\widetilde{\boldsymbol{\Theta}}^{(K),*})$$

where the two sets of meta-parameters are trained with uniformly sampled tasks and the optimal tasks, and $\widetilde{\boldsymbol{\Theta}}$ is used to denote the adapted meta-parameters $\mathcal{I}(\mathcal{T}, \boldsymbol{\Theta})$ for simplicity. With any convex loss function (e.g., $L_2$ loss or cross-entropy loss) under the over-parameterization settings, we will have the generalization loss being almost convex w.r.t. the meta-parameters as in **Lemma** D.14, which leads to

$$\widetilde{\mathcal{L}}(\widetilde{\boldsymbol{\Theta}}_u^{(K)}) - \widetilde{\mathcal{L}}(\widetilde{\boldsymbol{\Theta}}^{(K),*}) \leq \langle \nabla_{\widetilde{\boldsymbol{\Theta}}} \widetilde{\mathcal{L}}(\widetilde{\boldsymbol{\Theta}}_u^{(K)}), \widetilde{\boldsymbol{\Theta}}_u^{(K)} - \widetilde{\boldsymbol{\Theta}}^{(K),*} \rangle + \epsilon$$

$$\leq \|\nabla_{\widetilde{\boldsymbol{\Theta}}} \widetilde{\mathcal{L}}(\widetilde{\boldsymbol{\Theta}}_u^{(K)})\|_2 \|\widetilde{\boldsymbol{\Theta}}_u^{(K)} - \widetilde{\boldsymbol{\Theta}}^{(K),*}\|_2 + \epsilon$$

$$\leq \|\nabla_{\widetilde{\boldsymbol{\Theta}}} \widetilde{\mathcal{L}}(\widetilde{\boldsymbol{\Theta}}_u^{(K)})\|_2 \|\boldsymbol{\Theta}_u^{(K)} - \boldsymbol{\Theta}^{(K),*}\|_2 + \eta_1 \cdot \mathcal{O}(\sqrt{m_{\mathcal{F}} L_{\mathcal{F}}}) + \epsilon$$

$$\overset{(i)}{\leq} \sqrt{m_{\mathcal{F}} L_{\mathcal{F}}} \cdot \omega + \mathcal{O}(\omega^{4/3} L_{\mathcal{F}}^3 \sqrt{m_{\mathcal{F}} \log(m_{\mathcal{F}})}) + \mathcal{O}\left(\sqrt{\frac{L_{\mathcal{F}}}{m_{\mathcal{F}}}}\right)$$

$$\overset{(ii)}{\leq} \mathcal{O}\left( K L_{\mathcal{F}} + \frac{K^{4/3} L_{\mathcal{F}}^{11/3} \sqrt{\log(m_{\mathcal{F}})}}{m_{\mathcal{F}}^{1/6}} + \sqrt{\frac{L_{\mathcal{F}}}{m_{\mathcal{F}}}} \right)$$

$$\overset{(iii)}{\Longrightarrow} \widetilde{\mathcal{L}}(\widetilde{\boldsymbol{\Theta}}_u^{(K)}) - \widetilde{\mathcal{L}}(\widetilde{\boldsymbol{\Theta}}^{(K),*}) \leq \min \left\{ \mathcal{O}\left( K L_{\mathcal{F}} + \frac{K^{4/3} L_{\mathcal{F}}^{11/3} \sqrt{\log(m_{\mathcal{F}})}}{m_{\mathcal{F}}^{1/6}} + \sqrt{\frac{L_{\mathcal{F}}}{m_{\mathcal{F}}}} \right), 1 \right\}$$

where $\epsilon = \mathcal{O}(\omega^{4/3} L_{\mathcal{F}}^3 \sqrt{m_{\mathcal{F}} \log(m_{\mathcal{F}})}) > 0$, and $\|\boldsymbol{\Theta}_u^{(K),*} - \boldsymbol{\Theta}_u^{(K)}\|_2 \leq \omega$. Here, the first inequality is due to **Lemma** D.14 and the convexity of the loss function. The third inequality is due to the upper bound for meta-model gradients (**Lemma** D.15). The (i) is due to **Lemma** D.16 and sufficiently small learning rate $\eta_1 \leq \mathcal{O}(\frac{1}{m_{\mathcal{F}}})$. Based on **Lemma** D.15, we will have $\|\nabla_{\boldsymbol{\Theta}} \mathcal{L}(\boldsymbol{x}; \boldsymbol{\Theta}_K^{(J),*})\|_2, \|\nabla_{\boldsymbol{\Theta}} \mathcal{L}(\boldsymbol{x}; \boldsymbol{\Theta}_K^{(J)})\|_2 \leq \mathcal{O}(\sqrt{m_{\mathcal{F}} L_{\mathcal{F}}})$. Since we have $\eta_1, \eta_2 \leq \mathcal{O}(\frac{1}{m})$, starting from randomly initialized $\boldsymbol{\Theta}^{(0)}$, the parameter shift caused by GD can be upper bounded by $\|\boldsymbol{\Theta}_u^{(K),*} - \boldsymbol{\Theta}_u^{(K)}\|_2 \leq \omega = \mathcal{O}(K \cdot \sqrt{\frac{L_{\mathcal{F}}}{m_{\mathcal{F}}}})$. The implication (iii) is because the loss function $\mathcal{L}(\cdot; \cdot)$ has the value range [0, 1]. $\qquad \square$

Here, we notice that the RHS of the regret bound in **Lemma** D.13 has two terms. Although the second term can be reduced to $\mathcal{O}(1)$ with sufficiently large meta-model width $m_{\mathcal{F}} > \mathcal{O}(\text{Poly}(K, L, \rho^{-1}))$, the first term tends to grow along with more iterations $K$ and the larger meta-model width $m_{\mathcal{F}}$. The reason is that the radius for the parameter shift during meta-training $\omega$ can be as large as $\mathcal{O}(\frac{1}{\sqrt{m_{\mathcal{F}}}})$, which means that it cannot cancel out the effects of gradient norms, which have the order of $\mathcal{O}(\sqrt{m_{\mathcal{F}}})$. In this case, we will not able to include a $m_{\mathcal{F}}$ term to the denominator to scale down the regret with $m_{\mathcal{F}}$, and make the upper bound narrower than 1.

**Lemma D.14.** *Given an arbitrary sample $\boldsymbol{x}$ and its label, let $\widetilde{\mathcal{L}}(\boldsymbol{\Theta}) = \mathcal{L}(\boldsymbol{x}; \boldsymbol{\Theta})$. Suppose $m_{\mathcal{F}}, \eta_1, \eta_2$ satisfy the conditions in Theorem 5.2. With probability at least $1 - \mathcal{O}(K L_{\mathcal{F}}^2) \cdot \exp[-\Omega(m_{\mathcal{F}} \omega^{2/3} L_{\mathcal{F}})]$ over randomness of $\boldsymbol{\Theta}^{(0)}$, for all $k \in [K]$, and $\boldsymbol{\Theta}, \boldsymbol{\Theta}'$ satisfying $\|\boldsymbol{\Theta} - \boldsymbol{\Theta}^{(0)}\|_2 \leq \omega$ and $\|\boldsymbol{\Theta}' - \boldsymbol{\Theta}^{(0)}\|_2 \leq \omega$ with $\omega \leq \mathcal{O}(L_{\mathcal{F}}^{-6} [\log m_{\mathcal{F}}]^{-3/2})$, it holds uniformly that*

$$\widetilde{\mathcal{L}}(\boldsymbol{\Theta}) - \widetilde{\mathcal{L}}(\boldsymbol{\Theta}') \leq \langle \nabla_{\boldsymbol{\Theta}} \widetilde{\mathcal{L}}(\boldsymbol{\Theta}), \boldsymbol{\Theta} - \boldsymbol{\Theta}' \rangle + \epsilon.$$

*with $\epsilon = \mathcal{O}(\omega^{4/3} L_{\mathcal{F}}^3 \sqrt{\log m_{\mathcal{F}}})$ being a small constant.*

**proof.** This proof follows an analogous approach as the proof of Lemma 4.2 in [11]. Let $\nabla_{\mathcal{F}} \widetilde{\mathcal{L}}(\boldsymbol{\Theta}')$ be the derivative of $\widetilde{\mathcal{L}}$ with respective to $\mathcal{F}(\boldsymbol{x}; \boldsymbol{\Theta})$. Then, it holds that $|\nabla_{\mathcal{F}} \widetilde{\mathcal{L}}(\boldsymbol{\Theta}')| \leq \mathcal{O}(1)$ based on

**Lemma** D.15. Then, by convexity of $\widetilde{\mathcal{L}}$, we have

$$\widetilde{\mathcal{L}}(\boldsymbol{\Theta}') - \widetilde{\mathcal{L}}(\boldsymbol{\Theta})$$

$$\overset{(i)}{\geq} \nabla_{\mathcal{F}}\widetilde{\mathcal{L}}(\boldsymbol{\Theta}) \cdot (\mathcal{F}(\boldsymbol{x};\boldsymbol{\Theta}') - \mathcal{F}(\boldsymbol{x};\boldsymbol{\Theta}))$$

$$\overset{(ii)}{\geq} \nabla_{\mathcal{F}}\widetilde{\mathcal{L}}(\boldsymbol{\Theta}') \cdot \langle \nabla_{\boldsymbol{\Theta}}\mathcal{F}(\boldsymbol{x};\boldsymbol{\Theta}), \boldsymbol{\Theta}' - \boldsymbol{\Theta}\rangle$$

$$\quad - |\nabla_{\mathcal{F}}\widetilde{\mathcal{L}}(\boldsymbol{\Theta}')| \cdot |\mathcal{F}(\boldsymbol{x};\boldsymbol{\Theta}') - \mathcal{F}(\boldsymbol{x};\boldsymbol{\Theta}) - \langle \nabla\mathcal{F}(\boldsymbol{x};\boldsymbol{\Theta}), \boldsymbol{\Theta}' - \boldsymbol{\Theta}\rangle|$$

$$\geq \langle \nabla_{\boldsymbol{\Theta}}\widetilde{\mathcal{L}}(\boldsymbol{\Theta}), \boldsymbol{\Theta}' - \boldsymbol{\Theta}\rangle - |\nabla_{\mathcal{F}}\widetilde{\mathcal{L}}(\boldsymbol{\Theta}')| \cdot |\mathcal{F}(\boldsymbol{x};\boldsymbol{\Theta}') - \mathcal{F}(\boldsymbol{x};\boldsymbol{\Theta}) - \langle \nabla\mathcal{F}(\boldsymbol{x};\boldsymbol{\Theta}), \boldsymbol{\Theta}' - \boldsymbol{\Theta}\rangle|$$

$$\overset{(iii)}{\geq} \langle \nabla_{\boldsymbol{\Theta}}\widetilde{\mathcal{L}}(\boldsymbol{\Theta}), \boldsymbol{\Theta}' - \boldsymbol{\Theta}\rangle - \mathcal{O}(\omega^{4/3}L_{\mathcal{F}}^3\sqrt{m_{\mathcal{F}}\log(m_{\mathcal{F}})})$$

$$\geq \langle \nabla_{\boldsymbol{\Theta}}\widetilde{\mathcal{L}}(\boldsymbol{\Theta}), \boldsymbol{\Theta}' - \boldsymbol{\Theta}\rangle - \epsilon$$

where (i) is due to the convexity of the loss function $\mathcal{L}$, (ii) is an application of triangle inequality, and (iii) is the application of and **Lemma** D.16. Finally, denoting $\epsilon = \mathcal{O}(\omega^{4/3}L_{\mathcal{F}}^3\sqrt{m_{\mathcal{F}}\log m_{\mathcal{F}}})$ will complete the proof.

$\square$

**Lemma D.15.** *Suppose $m_{\mathcal{F}}, \eta_1, \eta_2$ satisfy the conditions in Theorem 5.2. With probability at least $1 - \mathcal{O}(KL_{\mathcal{F}}) \cdot \exp(-\Omega(m_{\mathcal{F}}\omega^{2/3}L_{\mathcal{F}}))$ over the random initialization, $\boldsymbol{\Theta}$ satisfying $\|\boldsymbol{\Theta} - \boldsymbol{\Theta}^{(0)}\|_2 \leq \omega$ with $\omega \leq \mathcal{O}(L_{\mathcal{F}}^{-9/2}[\log m_{\mathcal{F}}]^{-3})$, it holds uniformly that*

$$\|\nabla_{\boldsymbol{\Theta}}\mathcal{F}(\boldsymbol{x};\boldsymbol{\Theta})\|_2 \leq \mathcal{O}(\sqrt{m_{\mathcal{F}}L_{\mathcal{F}}}),$$
$$\|\nabla_{\boldsymbol{\Theta}}\mathcal{L}(\boldsymbol{x};\boldsymbol{\Theta})\|_2 \leq \mathcal{O}(\sqrt{m_{\mathcal{F}}L_{\mathcal{F}}}).$$

**Proof.** This lemma is a direct application of Lemma 9 of [47] and Lemma B.2, B.3 of [11].

$\square$

**Lemma D.16.** *Suppose $m_{\mathcal{F}}, \eta_1, \eta_2$ satisfy the conditions in Theorem 5.2. With probability at least $1 - \mathcal{O}(KL_{\mathcal{F}}) \cdot \exp(-\Omega(m_{\mathcal{F}}\omega^{2/3}L_{\mathcal{F}}))$, for all $t \in [T], i \in [k]$, $\boldsymbol{\Theta}, \boldsymbol{\Theta}'$ (or $\Theta, \Theta'$ ) satisfying $\|\boldsymbol{\Theta} - \boldsymbol{\Theta}^{(0)}\|_2, \|\boldsymbol{\Theta}' - \boldsymbol{\Theta}^{(0)}\|_2 \leq \omega$ with $\omega \leq \mathcal{O}(L_{\mathcal{F}}^{-9/2}[\log m_{\mathcal{F}}]^{-3})$, it holds uniformly that*

$$|\mathcal{F}(\boldsymbol{x};\boldsymbol{\Theta}) - \mathcal{F}(\boldsymbol{x};\boldsymbol{\Theta}') - \langle \nabla_{\boldsymbol{\Theta}'}\mathcal{F}(\boldsymbol{x};\boldsymbol{\Theta}'), \boldsymbol{\Theta} - \boldsymbol{\Theta}'\rangle| \leq \mathcal{O}(w^{1/3}L_{\mathcal{F}}^2\sqrt{m_{\mathcal{F}}\log(m_{\mathcal{F}})})\|\boldsymbol{\Theta} - \boldsymbol{\Theta}'\|_2.$$

**Proof.** The proof for this lemma directly follows the proof of Lemma 4.1 in [11] and Lemma 7 in [47].

$\square$

