# OpenReview forum: "Meta-Learning with Neural Bandit Scheduler"
_NeurIPS.cc/2023/Conference — NeurIPS 2023 poster_

### Official Review · Reviewer_ua81 · 2023-07-02

**Soundness:** 3 good
**Presentation:** 3 good
**Contribution:** 2 fair
**Rating:** 6
**Confidence:** 4

**Summary:**

This paper proposes scheduling tasks for meta-learning using context bandits.

**Strengths:**

1. The idea of applying context bandits to select tasks for meta-learning is novel
2. The experimental results are strong, especially when the task distribution is skewed

**Weaknesses:**

1. The method is computationally inefficient in two aspects: a) The arm context requires computing both the adapted parameters and the meta-parameters for every task. While some of these gradients can be reused, the computational cost can still be significant as the number of training tasks increases. b) The networks $f_1,f_2$ used to compute the benefit score take parameters as inputs, which is huge in practice. Although, in Remark 3, the authors mention applying the average pooling to reduce the cost, it is unclear how it is done in practice. Moreover, $f_2$ requires computing the gradients of $f_1$ for two different parameter sets.
2. There is no training time comparison between different methods.

**Questions:**

1. What’s the definition of $\mathbf{\theta}$ in Equation (7)? Is it $\theta_1$?
2. Why not simply use $\tilde r_{k,i}+\tilde e_{k,i}$ as the benefit score? In that case, there would be no need to learn $f_2$. Moreover, I believe that, with a little more effort, $f_1$ could be removed either.
3. Could you please detail the average pooling process in Remark 3?

**Limitations:**

This work does not involve some potential negative social impacts.

---

> ### Author Rebuttal · Authors · 2023-08-09
>
> We sincerely thank the reviewer for the valuable questions and comments. Here, we will try our best to address the questions and concerns in the form of Q\&A. Since we are unable to submit the improved manuscript based on reviewers' comments, we will describe these modifications on the current manuscript instead.
> **To better answer reviewer's questions, we also include additional experiments in the 1-page PDF file.**
> *If you have any additional questions or comments, please kindly let us know. Thank you!*
>
> **Q1: What’s the definition of $\theta$ in Equation (7)? Is it $\theta_{1}$?**
>
> Yes, the $\theta$ notation here refers to $\theta_{1}$, and this is because the exploration module $f_{2}$ takes the gradients of $f_{1}$ (with respect to $\theta_{1}$) as input.
> We have updated the manuscript for this part to avoid any confusion.
>
> **Q2: Why not simply use $\tilde{r} + \tilde{e}$?**
>
> *As we may not fully understand this question, we will really appreciate it if the reviewer let us know whether our initial response below could adequately address your concerns. **If not, please kindly let us know, and we will try our best to make ourselves clear. Thank you!** *
>
> Recall that for an arm (task), we have $\tilde{r}$ being the output of $f_{1}$, which refers to the estimated arm reward, and it measures the instant benefit (reward) of including this task (arm) into the meta-training.
> Then, $\tilde{e}$ refers to the corresponding exploration score, and it measures the uncertainty of $\tilde{r}$. Note that since $\tilde{e}$ is the output of $f_{2}$, we need the exploration model $f_{2}$ to estimate $\tilde{e}$ on the fly. In this way, $\tilde{r} + \tilde{e}$ will equal to the addition of $f_1$ and $f_2$ outputs.
>
> Next,
> as we mentioned in the paper, we apply $\alpha\cdot \tilde{r} + \tilde{e}$ to balance the exploitation and exploration, where higher $\alpha$ values lead to higher levels of exploitation. Meanwhile, $\alpha$ is also used to balance our two exploration objectives (lines 208-224).
> Please also refer to the experimental results in Table 3 (Appendix) for the effects of $\alpha$. Based on Table 3 (Appendix), we can generally choose $\alpha\in [0.5, 0.7]$, because in this way we can balance the exploitation-exploration as well as balance our two exploration objectives (lines 83-91, Appendix).
> This also supports our claim that $f_{2}$ is necessary for our proposed BASS model.
>
>
> Furthermore, we would like to include additional discussions on our adaptive exploration strategy.
> Our proposed BASS adopts $f_{2}$ to adaptively learn the exploration score, which can be either positive or negative.
> Here, the intuition is that the exploitation model (i.e., $f_{1}$) can "provide the overly high estimation of the arm reward", and applying the upper confidence bound (UCB) exploration strategy can amplify the mistake, as the UCB is non-negative.
> For notation simplicity, let us denote the expected reward of an arm $x$ as $\mathbb{E}[r] = h(x)$, where $h$ is the unknown reward mapping function. The corresponding reward estimation is denoted as $\hat{r} = f_{1}(x)$ where $f_{1}(\cdot)$ is the exploitation model.
> When the estimated reward is lower than the expected reward ($f_{1}(x) < h(x)$), we will apply the "upward" exploration (i.e., positive exploration score) to increase the chance of arm $x$ being explored. On the contrary, if the estimated reward is higher than the expected reward ($f_{1}(x) > h(x)$), we will apply the "downward" exploration  (i.e., negative exploration score)  instead to tackle the excessively high reward estimation.
> In this work, we apply the exploration model $f_{2}$ to adaptively learn the relationship between the network gradients of $f_{1}$ and the reward estimation residual $h(x) - f_{1}(x)$.
> Moreover, we also incorporate the task adaptation difficulty level to the exploration score, for a refined exploration strategy under the meta-learning task scheduling settings.
>
>
> We have also added the above discussion to the Appendix for readers' reference.
>
> **Q3: Details of the average pooling process in Remark 3?**
>
> As we have discussed in the paper, since the dimensionality of original meta-parameters could be high, inspired by CNNs, we apply average pooling to embed them into low-dimensional vectors. For instance, suppose the dimensionality of the original meta-parameters is 10,000 and our average pooling step is 100. Here, the first 100 elements in the original meta-parameters will be averaged, and the mean value will be considered as the first element of the average-pooled representation vector. Applying this process for the rest elements in the original meta-parameters, we can obtain the average-pooled representation vector (of the dimensionality $10,000 / 100 = 100$) for the original meta-parameters.
>
>
> We have also updated the Remark 3 in the manuscript for better presentation and clarity. Meanwhile, please see Table 3 in the uploaded PDF file for reference, where we compare the model performance with different levels of average pooling.
> Here, overly small dimensionality of the average-pooled vector representation (e.g., 20) can lead to sub-optimal performance of the BASS framework. Meanwhile, we see that setting the dimensionality to 50 can generally lead to the good performance, because this very simple yet effective method can preserve the local characteristic of meta-parameters.
>
> **Q4: Running time of BASS?**
>
> In Figure 1 of the attached 1-page PDF file, we include additional running time comparison results, compared with ATS under 6 different settings.
> This is because ATS is generally the second best task scheduling method, and it is also the only method among baselines that can achieve adaptive task scheduling in meta-learning.
> Compared with ATS, we can observe that BASS can achieve significant running time improvements, and take as little as 50\% of the ATS's running time.
>
> We have also updated the manuscript Appendix by adding these results.

---

> > ### Comment · Reviewer_ua81 · 2023-08-11
> >
> > Thank you for the detailed feedback. I'm still confused about Q2. What I originally meant was that $\tilde e_{k, i}$ can always be computed from $f_1$ during the training time. In that case, why do we introduce another function $f_2$ rather than directly using $\tilde e_{k, i}$ as the exploration reward?

---

> > > ### Author Response · Authors · 2023-08-11
> > > **Further clarification on Q2**
> > >
> > > Thank you so much for getting back to us, and we would like to further clarify your concerns on this question.
> > >
> > >
> > > First, we would like to apologize for the possible confusion caused by our previous response. Here, based on Eq. (7) in the main body, the actual outputs of $f_{1}$, $f_{2}$ are $\hat{r}$ and $\hat{e}$ (symbols with the "hat"), respectively. Then, as in Eq. (8) and (9) in main body, we have $\tilde{r}$ and $\tilde{e}$ (symbols with the "tilde") being the unbiased approximation for the reward and exploration score, individually.
> > > The unbiased approximation here will be used to train the bandit scheduler (lines 225-232, main body).
> > >
> > > To achieve adaptive exploration, we need to obtain the exploration score (please also see our previous response on why we need adaptive exploration strategy).
> > > Based on the definition of $\tilde{e}$ in Eq. (9) of the paper main body, it consists of three values, which are unbiased reward approximation $\tilde{r}$, the output of $f_{1}$, and the validation loss $\mathcal{L}\_{k, i}$. While, the output of $f_{1}$ and $\mathcal{L}\_{k, i}$ are calculated during the inference phase (lines 6-10, Algorithm 1 in main body), the unbiased reward approximation $\tilde{r}$ (in Eq. (8)) are still unknown.
> > > In this case, if we do not use $f_{2}$ to learn $\tilde{e}$ but choose to directly calculate $\tilde{e}$, we will need to first calculate the unbiased reward estimations $\tilde{r}$ (in Eq. (8)) for **all the candidate arms** in the candidate pool $\Omega_{\text{task}}^{(k)}$,  whose cardinality can be a significantly large number.
> > > In this case, the computational cost can be prohibitive, since the calculation of the unbiased reward estimations $\tilde{r}$ involves the meta-adaptation ($J$-step Gradient Descent. Eq. (1) in the main body) on all the validation tasks $\mathcal{T}^{\text{valid}}\in \Omega_{k}^{\text{valid}}$.
> > >
> > > Alternatively, we apply the exploration model $f_{2}$ to calculate the exploration score estimations, which significantly reduces the computational cost.
> > > Afterwards, as shown in lines 11-15 of Algorithm 1, for the sake of training the bandit scheduler, we only need to derive the unbiased reward estimations $\tilde{r}$ (in Eq. (8)) for **the chosen arms** $\Omega_{k}$. Since the size of candidate pool $| \Omega_{\text{task}}^{(k)} |$ is considerably larger compared with the number of chosen arms $| \Omega_{k} |$, our applying of the $f_{2}$ can dramatically reduce the running time.
> > > Therefore, $f_{2}$ is necessary to achieve adaptive exploration in task scheduling in meta-learning with high efficiency.
> > >
> > > We understand that our notation may cause confusion to the reviewer and other readers, such as the subtle visual differences between "hat" symbols and "tilde" symbols. Thus, we will update the corresponding notation and the narrative to offer better clarity. **If you have any further concerns, please kindly let us know. Thanks again for pointing out this issue.**

---

> > > > ### Comment · Reviewer_ua81 · 2023-08-12
> > > >
> > > > Thank the authors for addressing my questions. I've raised my rating from 4 to 6

---

> > > > > ### Author Response · Authors · 2023-08-12
> > > > > **Thank reviewer ua81 for your feedback**
> > > > >
> > > > > We would like to thank you again for the fruitful discussion. We will update the manuscript based on our discussion, as well as include explanation contents for adaptive exploration to the Appendix.

---

### Official Review · Reviewer_zT5d · 2023-07-04

**Soundness:** 3 good
**Presentation:** 3 good
**Contribution:** 3 good
**Rating:** 7
**Confidence:** 3

**Summary:**

This paper considers the problem of task scheduling in meta-learning. Under the gradient-based meta-learning framework, the authors propose BASS, which uses contextual bandits parameterized by neural networks. The tasks in each batch (arms) are selected in an optimistic manner, with the reward estimated using a validation set and its uncertainty estimated using errors in the neural network predictions. A theoretical analysis of the regret is done in the overparameterized neural network setting. Experiments are done on Mini-ImageNet, CIFAR100, and Drug datasets, showing competitiveness with SOTA and outperforming them when there is task imbalance.

**Strengths:**

1. The authors provide detailed mathematical motivation for their algorithmic choices.
2. The use of contextual bandits for meta-learning is novel, as far as I know.
3. The authors provide theoretical analysis of their algorithm.
4. Experiments are done on a range of datasets, with ablations.
5. BASS is able to lead to statistically significant improvements when there is task imbalance and is competitive otherwise.

**Weaknesses:**

1. There is no discussion of the computational burden of BASS. It would strengthen the work to provide wall clock times or plots of the performance as a function of time.
2. It is not clear how BASS would scale, as it takes the parameters of one neural network as input into another neural network.

**Questions:**

1. How does the computation time of BASS compare to the other algorithms?
2. How would BASS scale with the size of the classifier, e.g. if we were to meta-learn an initialization for 100 layers?

**Limitations:**

The paper proposes a new meta-learning algorithm, so I think the authors are ok in not including a broader impact section. The authors do adequately discuss limitations in the appendix.

---

> ### Author Rebuttal · Authors · 2023-08-09
>
> We sincerely thank the reviewer for the valuable questions and comments. Here, we will try our best to address the questions and concerns in the form of Q\&A. Since we are unable to submit the improved manuscript based on reviewers' comments, we will describe these modifications on the current manuscript instead.
> **To better answer reviewer's questions, we also include additional experiments in the 1-page PDF file.**
> *If you have any additional questions or comments, please kindly let us know. Thank you!*
>
>
> **Q1: Running time of BASS?**
>
> Thank you for the feedback.
> We have included additional running time comparison results, compared with ATS [1] on 6 different settings.
> This is because ATS is generally the second-best task scheduling method, and it is also the only baseline method that can achieve adaptive task scheduling in meta-learning.
> Please refer to the running time results shown in Figure 1 of the PDF file.
> From the figure, we can observe that BASS can achieve significant running time improvements, and take as little as 50\% of the ATS's running time.
>
> We have also updated the manuscript Appendix by adding these results.
>
>
>
>
>
> **Q2: How would BASS scale with the size of the classifier?**
>
>
> In general, it can be challenging to train large meta-models.
> In this case, we propose the approximation method by applying the average pooling to reduce the bandit scheduler input in practice (Remark 3).
> This method is simple, but it can preserve the characteristics of meta-parameters, and we can adjust the converted dimensionality based on the complexity of meta-models.
>
>
> Please see our additional experiment result in Table 3 of the uploaded PDF file for reference.
> Here, we see that setting the dimensionality to 50 can generally lead to the good performance, and the BASS framework can also deal with higher dimensional average-pooled vector representations (e.g., dimensionality $= 500$).
> Meanwhile, for the experiments in the paper, the effectiveness of BASS is tested on two different types of meta-models (Fully-connected networks for the Drug data set, and CNNs for the Mini-ImageNet and CIFAR-100 data sets).
> Meanwhile, compared with the state-of-the-art baseline ATS, we can achieve effective task scheduling with relatively small computational cost. Please also see the results of running time for reference (Figure 1 in the PDF file).
>
> **REFERENCE**
>
> [1] Huaxiu Yao, Yu Wang, Ying Wei, Peilin Zhao, Mehrdad Mahdavi, Defu Lian, and Chelsea Finn.
> Meta-learning with an adaptive task scheduler. Advances in Neural Information Processing Systems,
> 34:7497–7509, 2021.

---

> > ### Comment · Reviewer_zT5d · 2023-08-12
> > **Thanks to the authors for the rebuttal**
> >
> > After reading it and the other reviews, I have decided to raise my score from 6 to 7. I would encourage the authors to include a comprehensive comparison of computation time for all baselines in the final paper, as it seems most reviewers had this question.

---

> > > ### Author Response · Authors · 2023-08-12
> > > **Thank reviewer zT5d for the discussion**
> > >
> > > We sincerely thank the reviewer for your constructive comments and suggestions. We will definitely include a more detailed running time comparison with the baselines to the paper Appendix, along with other improved / newly added contents.

---

### Official Review · Reviewer_Z8mk · 2023-07-05

**Soundness:** 3 good
**Presentation:** 3 good
**Contribution:** 2 fair
**Rating:** 7
**Confidence:** 5

**Summary:**

The paper presents a task scheduling approach in meta-learning under a contextual bandit framework. The proposed methodology, named BASS, treats each meta-learning task as an arm, prioritizing the selection of these arms according to exploration and exploitation scores. These scores are computed by a trainable neural network with the input being the meta-model's status. The authors provide theoretical proof demonstrating the convergence of the regret bound. Empirical results indicate that BASS outperforms baselines in meta-learning tasks, particularly those involving noisy or skewed datasets.

**Strengths:**

**S1. Clarity**

The authors present their work with high clarity. The notations, a critical aspect in avoiding confusion in this complex subject, are well-defined. Also, the authors have strategically placed numerous remarks in the main section, which significantly assist readers in comprehending the principal concepts.

**S2. Advantages from the use of bandits**

The application of the contextual bandit in place of the previously used greedy update offers multiple strengths.
-	It eliminates the need for unstable bi-level optimization
-	In naturally manages the balance between exploration and exploitation. The way the extrinsic and intrinsic rewards are formulated is both intriguing and intuitive. Notably, the exploration factor, which takes into account both task uncertainty and difficulty, stands out as a significant advantage of this work.
-	As a consequence of the bandit application, the regret bound can be theoretically limited. The authors have put in considerable effort the substantiate their theoretical analysis in the appendix.

**Weaknesses:**

**W1. Experimental results**

While the proposed method demonstrates significant enhancements over baseline models for noisy or skewed training tasks, it appears to yield marginal or negligible improvement on standard datasets, as acknowledged by the authors. Incorporating additional data regarding computation cost could provide valuable insight, especially when comparing the effect of noise or skewness on different datasets and assessing the increased computation demand posed by BASS.

**Minor comment**

It is recommended to incorporate the 'Related Works' section into the main body of the manuscript instead of including it in the appendix. Given that all submissions adhere to the same page limit, placing this section in the appendix can disrupt the contextualization of the work within the existing literature.

**Acknowledgment Following Rebuttal**

The author's rebuttal successfully addressed concerns about the experiment. The inclusion of additional results on average-pooling is particularly noteworthy.

**Questions:**

**Q1.** The explanation provided from Line 179 to Line 185 is somewhat unclear. Could the authors provide further clarification on this matter?

**Q2.** The notations used in Line 108 and Line 326 appear to be inconsistent. Could these be typos?

**Q3.** It's great that BASS employs all parameters of the meta-model to infer its status. However, BASS uses average pooling to reduce millions of parameters to about 50. Can this average-pooled feature truly encompass all necessary information about the status of the meta-model?

**Q4.** In relation to Q3, the use of average-pooling seems to be due to the high dimensionality of the input. Have the authors considered employing other encoding techniques, such as an MLP with smaller hidden layers?

**Limitations:**

The authors have adequately addressed their limitations regarding the computational cost and the marginal improvement for standard dataset in the appendix.

---

> ### Author Rebuttal · Authors · 2023-08-09
>
> We sincerely thank the reviewer for the valuable questions and comments. Here, we will try our best to address the questions and concerns in the form of Q\&A. Since we are unable to submit the improved manuscript based on reviewers' comments, we will describe these modifications on the current manuscript instead.
> **To better answer reviewer's questions, we also include additional experiments in the 1-page PDF file.**
> *If you have any additional questions or comments, please kindly let us know. Thank you!*
>
>
> **Q1: Could the authors provide further clarification on Line 179 to Line 185?**
>
> Recall that for task (arm) $\mathcal{T}\_{k, i}$, the exploration module $f_{2}$ takes two gradient vector $\nabla_{\theta} f_{1}(\chi_{k, i}^{s})$, $\nabla_{\theta} f_{1}(\chi_{k, i}^{q})$ as the input.
> And this paragraph (lines 176-185, main body) aims to show our intuition of this design.
>
> Here, we show two cases as the examples:
> (1) The variance of the corresponding data distribution $\mathcal{D}\_{\mathcal{T}\_{k, i}}$ is high. In this case, the support set $D\_{k, i}^{s}$ and the query set $D\_{k, i}^{q}$ can be considerably different, which makes the corresponding arm contexts $\chi\_{k, i}^{s}$, $\chi\_{k, i}^{q}$ divergent. As a result, the gradient vectors $\nabla_{\theta} f_{1}(\chi_{k, i}^{s})$, $\nabla_{\theta} f_{1}(\chi_{k, i}^{q})$ will likely be distinct from each other.
> (2) Alternatively, suppose the support set $D_{k, i}^{s}$ and the query set $D_{k, i}^{q}$ are not significantly distinct (which means that $\chi_{k, i}^{s}$ does not significantly differ from $\chi_{k, i}^{q}$). Then, if these two gradient vectors still tend to change dramatically when adapting to task $\mathcal{T}\_{k, i}$, we can consider that the exploitation model $f_{1}$ is not well adapted to this task $\mathcal{T}\_{k, i}$.
>
> For the above two cases, we give two scenarios where more exploration is needed for the task $\mathcal{T}\_{k, i}$, and this is to help $f_{1}$ better learn the reward for this task.
> In this case, the information from these two gradient vectors can help us make better exploration decisions, since they can encode information regarding the dynamics of the meta-model parameters and the exploitation model $f_{1}$ parameters. And we apply the exploration model $f_{2}$ to learn from these two gradient vectors.
>
> We have also updated this part in the manuscript for better clarify and presentation.
>
>
>
>
> **Q2: The notations used in Line 108 and Line 326 appear to be inconsistent. Could these be typos?**
>
>
> Thank you so much for spotting this typo, and it should be $\mathcal{P}(\mathcal{T})$. We have updated the manuscript, and will also double-check the manuscript for other potential typos / mistakes.
>
>
>
>
>
>
>
> **Q3: BASS uses average pooling to reduce millions of parameters to about 50. Can this average-pooled feature truly encompass all necessary information about the status of the meta-model?**
>
>
> Here, we include additional experiments with different levels of average pooling, such that after the average pooling, the input dimensionality will fall into $\\{20, 50, 100, 500\\}$.
> Please see Table 3 in the uploaded PDF file for the experiment results.
> Here, overly small dimensionality of the average-pooled vector representation (e.g., 20) can lead to the sub-optimal performance of the BASS framework. Meanwhile, we see that setting the dimensionality to 50 can generally lead to good performance, which means the average pooling method can effectively preserve the characteristic of meta-parameters.
>
>
>
> **Q4: Have the authors considered employing other encoding techniques, such as an MLP with smaller hidden layers?**
>
>
> Thank you for your suggestion, and we include additional experimental results using MLP to map the original context into the lower dimensional space instead of using our proposed average pooling (Remark 3).
> Please see Table 4 in the uploaded PDF file for experiment results.
> Here, we use the one-layer MLP with the ReLU activation to embed the original meta-parameters to the low-dimensional vector representations. We can see that the MLP-based method can indeed lead to some performance improvement. But in general, the performance difference between MLP-based embedding and the average-pooling vector representation is subtle.
>
>
> Meanwhile, we also note that the MLP-based mapping approach is considerably more time consuming compared with the average pooling approach, since we also need to train the additional embedding layer, which has a large number of trainable parameters.
> Alternatively, the computation cost of the average pooling is trivial.
>
> We have also included these experiment results in the paper Appendix.
>
>
>
> **Q5: Running time results?**
>
> Please refer to the running time results shown in Figure 1 of the PDF file.
> Here, we include the running time comparison results compared with ATS on 6 different settings.
> This is because ATS is generally the second-best task scheduling method, and it is also the only baseline method that can achieve adaptive task scheduling in meta-learning.
> From the figure, we can observe that BASS can achieve significant running time improvements, and take as little as 50\% of the ATS's running time.
>
>
> **Q6: Move "Related Works" section to the main body?**
>
>
> Thank you for the suggestion, and we agree with the reviewer that the "Related Work" should be moved to the main body for the sake of consistency. Therefore, we have updated the manuscript and moved this section to the main body.

---

> > ### Comment · Reviewer_Z8mk · 2023-08-10
> > **Response to the Rebuttal**
> >
> > I'd like to thank the authors for addressing my concerns and for conducting additional experiments. It's noteworthy how effectively the vanilla average-pooling compresses the status of the meta-model. Based on this, I have adjusted my rating accordingly from 6 to 7.

---

> > > ### Author Response · Authors · 2023-08-11
> > > **Thank reviewer Z8mk for the discussion**
> > >
> > > We would like to sincerely thank the reviewer again for the discussion, and we will definitely update the new / improved contents to the paper.

---

### Official Review · Reviewer_nu6A · 2023-07-06

**Soundness:** 3 good
**Presentation:** 3 good
**Contribution:** 3 good
**Rating:** 6
**Confidence:** 2

**Summary:**

This paper proposed a task scheduling framework BASS based on the status of meta-model based on contextual bandits setting. BASS addressed the performance bottleneck of meta-models by balancing exploitation and exploration and handled the data scarcity in the early stages of meta-training iterations with planning future meta-learning iteration strategies. The experimental and theoretical analysis showed the effectiveness of BASS.

**Strengths:**

1 [Writing & Presentation] This paper is well-written and easy to follow.

2 [Motivation] The previous work about task schedulers aimed to improve meta-training strategies based on the various pre-defined criteria and assumptions, ignoring the global knowledge. It may result in a sub-optimal meta-model affected by noise perturbation or skewed task distributions. Motivated by this limitation, the authors proposed a novel framework to solve the meta-learning task scheduling problem.

3 [Contribution] Different from the existing methods that exploit the current/local knowledge for greedy scheduling, the proposed method leveraged a novel method to adaptively learn the relationship between the meta-model parameters and meta-model generalization ability.

4 [Experiments] The experiments demonstrated the effectiveness of BASS compared with seven strong basslines on three real datasets.
The experimental results showed superiority in terms of accuracy and efficiency. Besides, BASS can explore the ‘tail’ task and enjoy a good performance in the ensemble inference setting, which further enhanced the generalization ability of meta-learning models.

**Weaknesses:**

1 [Algorithm] In Algorithm 1. How about the detailed setting of the initialization? Intuitively, the meta-model parameters would rely on the various data distributions.

2 [ Theory] In Section 4, the $k$-iteration regret is the assumption of Theorem 4.2. However, the detailed setting of regret was not given. For example, the explanation of independence of regret bounds is not well illustrated.

3 [Experiment] In Section 5.1, the experimental analysis was on three datasets, the Drug dataset is a textual dataset, and both ImageNet and CIFAR are visual/image datasets. Since the proposed method focused on the generalization ability, it would be better to verify the performance on various categories of datasets.

4 [Experiment] In Table.1, the performance for 1-shot of BASS increased slightly. Does it mean the meta-model is invalid? If so, what is the main point of BASS leading to this result?

**Questions:**

1 [Refer to Weakness 1] How about the assumption on various datasets? Does the assumption independent of each other? Will the results vary with different initialization?

2 [Refer to Weakness 1] What are the constraints or the policy of batch size B? How will the performance vary with different settings of B?

3 [Refer to Weakness 2] How about the bound of regret? How about the distribution of arms? Whether the performance change with different settings/assumptions of regret/arms?

4 [Refer to Weakness 3] How about the performance on various kinds of datasets? For example, image dataset, textual dataset, vision dataset, or dataset represented by deep features. Furthermore, the data dimension, data sparsity, even the image resolution may also affect the performance.

**Limitations:**

The target problem is interesting and worth studying. However, the detailed explanation of theory analysis needs to be clarified, and more experiments should be conducted to verify the superiority of the proposed method.

---

> ### Author Rebuttal · Authors · 2023-08-09
>
> We sincerely thank the reviewer for the valuable questions and comments. Here, we will try our best to address the questions and concerns in the form of Q\&A. Since we are unable to submit the improved manuscript based on reviewers' comments, we will describe these modifications on the current manuscript instead.
> **To better answer reviewer's questions, we also include additional experiments in the 1-page PDF file.**
> *If you have any additional questions or comments, please kindly let us know. Thank you!*
>
> **Q1: How about the assumption on various datasets? Does the assumption independent of each other? Will the results vary with different initialization?**
>
> Analogous to other bandit-based works (e.g., Neural-UCB), in this paper, our regret bound considers the worst-case scenario. Therefore, as long as the separateness assumption (Assumption 4.1) is satisfied, we do not impose additional assumptions on the data or task distribution.
> Meanwhile, the separateness assumption is applied because we need the training data of the scheduler to be non-degenerate, in order to derive the performance guarantee.
> Please also see lines 244-254 (main body) as well as the Appendix Section C for the discussion on Assumption 4.1.
>
> For the meta-model, we first consider it to be a $L_{\mathcal{F}}$-layer fully-connected (FC) network (of width $m_{\mathcal{F}}$) with Gaussian initialization for the theoretical analysis (lines 237-239). In particular, we follow the settings in [1] for the Gaussian initialization of weight matrices.
> For the weight matrix elements in meta-model's first $(L_{\mathcal{F}}-1)$ layers, we draw each of them from the Gaussian distribution $\mathcal{N}(0, 2/m_{\mathcal{F}})$. Then, for the weight matrix elements of the last layer ($L_{\mathcal{F}}$-th layer), we draw each of them from the Gaussian distribution $\mathcal{N}(0, 1)$.
> Meanwhile, as we mentioned in lines 239-240 of the main body, our results can be generalized to other meta-model architectures (e.g., CNN and ResNet). For those architectures, we also can apply analogous Gaussian initialization procedures. Please see the parameter initialization details for these two architectures in Appendix Section B and Appendix Section C of [1].
>
> To avoid possible confusion from the readers, we have added the above details to the Appendix.
>
>
>
> **Q2: What are the constraints or the policy of batch size B? How will the performance vary with different settings of B?**
>
> For the regret bound, we do not have constraints for $B$. Instead, for the second term of the regret bound (RHS of Eq. 10), we see that the term $B$ is in the numerator. Since we are considering the worst-case scenario, with larger $B$, the bandit-trained meta-parameters ($\mathbf{\Theta}^{(K)}$) will be more likely to deviate from the the optimal ones ($\mathbf{\Theta}^{(K), *}$), which may lead to a larger regret bound.
>
> Here, we include additional experiments with different batch sizes $B$, in comparison with ATS and the uniform sampling approach.
> This is because ATS is generally the second-best method.
> Please see Table 2 in the uploaded PDF file for detailed settings and results.
> Based on the results, we see that with larger $B$ values, the accuracy of BASS as well as the baselines will generally improve, and BASS still maintains the best performance.
>
>
> **Q3: How about the bound of regret? How about the distribution of arms? Whether the performance change with different settings/assumptions of regret/arms?**
>
>
> As we have discussed in the answer to Q1, since the data / task distributions are unknown, our regret bound considers the worst-case scenario. In this case, as long as the conditions in Section 4 are met, our derived regret bound can deal with various arm / task / data distributions.
>
>
>
>
> **Q4: Experiment results for 1-shot settings?**
>
> First, we would like to recall that under the 5-shot settings where information for task adaptation is relatively sufficient, our proposed BASS can achieve considerable improvements over the existing baselines, which shows the effectiveness of our proposed approach.
>
> Then, with the accuracy results and the corresponding standard deviation results, we would like to note that our proposed BASS can achieve statistically comparable (or better) performances, in comparison with baselines. This is because under the 1-shot settings, regardless of the scheduling approaches, the meta-learning models will have relatively insufficient information regarding the task adaptation, which would lead to unsatisfactory performances and make the 1-shot settings more challenging. Meanwhile, our improvements over the uniform sampling approach also support the effectiveness of our task scheduling strategy.
>
> In particular, to deal with the challenging 1-shot settings, we offer a practical solution in our case study (Subsec. 5.3, main body) by utilizing the ensemble inference to improve the meta-model's generalization ability. In this way, our proposed BASS can achieve more significant advantage over the baselines.
>
>
>
> **Q5: How about the performance on other datasets with different specifications?**
>
> Thank you for your suggestion, and we agree that data set specifications (e.g., the image resolution) can considerably affect the model performance. Therefore, we include additional experiments on the "DomainNet" data set [Moment Matching for Multi-Source Domain Adaptation (ICCV 2019)], which has a higher resolution ($128\times128$) for the image data.
>
> Please see our general response for the detailed experiment settings (bullet point 6).
> For experimental results, please see Table 5 in the uploaded PDF file. Here, we see that with a higher image resolution of the "DomainNet" data set, BASS can still maintain the best performance compared with the baselines.
>
>
>
>
>
>
>
> **REFERENCE**
>
> [1] Allen-Zhu, Zeyuan, Yuanzhi Li, and Zhao Song. "A Convergence Theory for Deep Learning via Over-Parameterization." arXiv preprint arXiv:1811.03962 (2018).

---

> > ### Comment · Reviewer_nu6A · 2023-08-20
> >
> > I appreciate the authors' detailed response. My concerns are addressed properly. I raised my score from 5 to 6.

---

> > > ### Author Response · Authors · 2023-08-20
> > > **Thank Reviewer nu6A for the feedback**
> > >
> > > We would like to thank Reviewer nu6A again for the comments and suggestions. We will definitely update the manuscript to include the discussion as well as the additional experiment results in the paper Appendix.

---

### Official Review · Reviewer_XGJX · 2023-07-06

**Soundness:** 3 good
**Presentation:** 3 good
**Contribution:** 2 fair
**Rating:** 6
**Confidence:** 3

**Summary:**

This paper proposes to adaptively sample the meta-training tasks by optimizing the task scheduling strategy based on the status of the meta-model. The proposed method treats task scheduling as a contextual multi-arm bandit problem with a reward function balancing the exploitation and exploration. The authors provide theoretical analysis about the regret bound of the proposed method and conduct experiments on real data sets to show the effectiveness.

**Strengths:**

**DISCLAIMER:** I have not checked the proof thoroughly and cannot verify the correctness of the theorems.

* The presentation of the paper is clean and clear.
* It seems to be novel to formulate the curriculum learning as a contextual bandit in the context of meta-learning.

**Weaknesses:**

* The authors claim the proposed method can deal with the data scarcity problem at the early stage of meta-learning. However, I cannot further elaboration anywhere in the paper nor any experimental results supporting this claim.
* Since the proposed method involves extra computation efforts, the quantitative comparison of computation cost is not included in the experiment results.
* If the proposed method outperforms the existing method, especially when the data is noisy and skewed, the authors should include experiments with different levels of noise $\epsilon$ and different levels of skewness to show the trends of improvements.

**Questions:**

* How is the K-round regret bound in Theorem 4.2 compared with the existing work?
* How is the performance of the proposed method compared with the existing method when the data is noise-free and unskewed?

**Limitations:**

The authors discuss the limitation in the appendix.

---

> ### Author Rebuttal · Authors · 2023-08-09
>
> We sincerely thank the reviewer for the valuable questions and comments. Here, we will try our
> best to address the questions and concerns in the form of Q&A. Since we are unable to submit the
> improved manuscript based on reviewers’ comments, we will describe these modifications on the current
> manuscript instead. **To better answer reviewer’s questions, we also include additional experiments in
> the 1-page PDF file.** *If you have any additional questions or comments, please kindly let us know.
> Thank you!*
>
> **Q1: How is the K-round regret bound in Theorem 4.2 compared with the existing work?**
>
> To the best of our knowledge, this is the first work to incorporate bandit ideas for the task scheduling problem in meta-learning, and it is also the first work that introduces the performance guarantee for the task scheduling problem.
>
> Meanwhile, instead of our carefully designed modeling, if we directly apply the existing bandit works to the task scheduling problem, we will (1) introduce additional assumptions for the sake of analysis (e.g., Lin-UCB [1] assumes the reward mapping function $h(\cdot)$ is linear), or (2) introduce additional terms to the regret bound (e.g., Neural-UCB [2] will need an additional $\tilde{d}$ in the regret bound, which is the effective dimension of the related NTK matrix).
>
> In our discussion in the main body (lines 266-269), we also show that the uniform sampling approach can lead to $\mathcal{O}(1)$ regret under the worst-case scenario, which is significantly worse compared with our derived regret bound (Eq. (10)).
> Please also refer to our discussion on the regret bound (lines 261-276 of the main body).
>
> **Q2: How is the performance of the proposed method compared with the existing method when the data is noise-free and unskewed?**
>
>
> Please refer to Table 2 (Subsec. B.1.1) in the Appendix for the experimental results when the data is noise-free and unskewed. We show that BASS can still achieve the best performance compared with baselines. Please also see the results from Table 1 (Appendix) for the experimental results with different noise levels.
>
> Meanwhile, we also include the experiments with different levels of skewness. Please see Table 1 in the uploaded PDF file for details. Here, we see that with less skewness levels (the skewness level reduces from Setting 1 to Setting 3), the accuracy of BASS as well as the baselines will continue to increase, and BASS can still maintain the best performance.
>
>
>
>
>
>
>
> **Q3: The claim of the data scarcity problem?**
>
> In the abstract and the conclusion, we mention that "BASS can deal with the data scarcity problem at the early stage of meta-training, and plan for the future meta-training iterations".
> Here, we mean that, due to the insufficient knowledge regarding the task and data distributions in the early stage of meta-training, existing greedy meta-task schedulers may lead sub-optimal meta-models, since they tend to make scheduling decisions solely based on the limited existing knowledge, without performing exploration for potential benefits.
> Alternatively, with the exploration strategy, our method can be less likely to be significantly affected by the insufficient knowledge issue (e.g., the skewed data distribution example).
> That is why we need exploration for task scheduling, especially during an early stage of meta-training when the learner's knowledge is limited.
>
> In the experiments, we have shown BASS can achieve the same accuracy as baselines using less number of training round (Figure 4, main body). This fact can be interpreted as the support for the exploitation-exploration strategy used in our approach, which is able to maximize the long-term benefit.
>
>
> Since this narrative may confuse the reviewer and the future readers,
> we have updated the abstract as well as the conclusion sections for better presentation, by mentioning the importance of exploration at the early stage of meta-training, due to the insufficient knowledge with respect to the task and data distributions.
>
>
>
> **REFERENCE**
>
> [1] Wei Chu, Lihong Li, Lev Reyzin, and Robert Schapire. Contextual bandits with linear payoff
> functions. In AISTATS, pages 208–214, 2011.
>
> [2] Dongruo Zhou, Lihong Li, and Quanquan Gu. Neural contextual bandits with ucb-based exploration,
> 2020.

---

> > ### Comment · Reviewer_XGJX · 2023-08-18
> > **RE:**
> >
> > Thanks for the clarification and added experiments. I raised my rating to 6.

---

> > > ### Author Response · Authors · 2023-08-18
> > > **Thank Reviewer XGJX for the discussion**
> > >
> > > We would like to thank the reviewer XGJX again for the valuable discussion, and we will definitely add the clarification as well as the additional experiment results to the paper Appendix.

---

### Author Rebuttal · Authors · 2023-08-09


We would like to take the chance to thank all the reviewers for your constructive feedback and detailed comments for our work. Your suggestions can definitely make this paper a more solid one.
Here, to better resolve the questions from reviewers, we have included supplementary experiments to provide the additional reference. **Please see the attached PDF file for details.**
Our experiments include:

1. [Figure 1] We include the running time comparison with the adaptive scheduler ATS, because ATS can generally achieve the best performance among the baselines.
    * We can see that BASS can achieve significant improvement in terms of the running time, and can take as little as 50\% of ATS's running time.
    * (Intuition) This improvement is because our proposed BASS only needs one round of the optimization process to update the meta-model and BASS.
On the other hand, from Algorithm 1 of the ATS paper [2], we see that ATS requires two optimization rounds to update the scheduler (lines 8-12, Algorithm 1 in [2]) with the temporal meta-model, and the actual meta-model (lines 13-14, Algorithm 1 in [2]) respectively.
(Please also refer to Figure 2 in our paper main body.)

2. [Table 1] We include the experiments with different levels of skewness.
    * Here, we see that with less skewness levels (the skewness magnitude reduces from Pattern 1 to Pattern 3), the accuracy of BASS as well as the baselines will continue to improve, while BASS still maintains the best performance.


3. [Table 2] We include additional experiments with different batch sizes $B$, in comparison with the ATS and the uniform sampling approach.
    * Here, we see that with larger $B$ values, the accuracy of BASS as well as the baselines will generally improve, and BASS still maintains the best performance.


4. [Table 3] We include additional experiments with different levels of average pooling, such that after the average pooling, the dimensionality of the pooled vector representation will fall into $\\{20, 50, 100, 500\\}$.
    * Here, overly small dimensionality of the average-pooled vector representation (e.g., 20) can lead to sub-optimal performance of the BASS framework. Meanwhile, we see that setting the dimensionality to 50 can generally lead to good enough performance.



5. [Table 4] Meanwhile, we also include additional experimental results using MLP to map the original context into the lower dimensional space instead of using our proposed average pooling (Remark 3).
    * Here, we use the one-layer MLP with the ReLU activation to embed the original meta-parameters to the low-dimensional vector representations. We can see that the MLP-based method can indeed lead to some performance improvement. But in general, the performance difference between MLP-based embedding and the average-pooling vector representation is subtle.
    * We also note that the MLP-based mapping approach is considerably more time consuming compared with the average pooling approach, since we also need to train the additional embedding layer, which has a large number of trainable parameters.




6. [Table 5] We include additional experiments on the new "DomainNet" data set [1].
    * Within the "real" domain, we filter 100 classes that have at least 600 images. In this way, with each class being a task with 600 images, we will have a total of 100 tasks.
Compared with image data sets in our paper (Mini-ImageNet and CIFAR-100), we increase the image resolution of "DomainNet" by resizing its images to 128$\times$128 pixels.
    * Following the settings in our paper, we divide tasks into the portions of 64 : 16 : 20 that correspond to the training set, validation set and the test set respectively. For the few-shot settings, we formulate the problem to be 5-shot, 5-way / 7-way. We include uniform sampling and ATS as baselines, since they generally perform the best among baselines.
    * Here, we see that with a higher image resolution of the "DomainNet" data set, BASS can still maintain the best performance compared with the baselines.


We will also update all these additional experiment results to the paper Appendix.


**REFERENCE**

[1] Xingchao Peng, Qinxun Bai, Xide Xia, Zijun Huang, Kate Saenko, and Bo Wang. Moment matching
for multi-source domain adaptation. In ICCV, pages 1406–1415, 2019.

[2] Huaxiu Yao, Yu Wang, Ying Wei, Peilin Zhao, Mehrdad Mahdavi, Defu Lian, and Chelsea Finn.
Meta-learning with an adaptive task scheduler. Advances in Neural Information Processing Systems,
34:7497–7509, 2021.

---

### Decision · Program_Chairs · 2023-09-21

**Decision:**

Accept (poster)

**Comment:**

Reviewers have reached a consensus and recognise the strengths of these papers as:
* original novel idea
* Clarity of the presentation
* strong empirical results

We this recommend this paper to be accepted and we encourage the authors to revise their draft to reflect the discussions of the rebuttal period.